# The Convergent Representation of Contrastive Vision-Language Models: Geometry, Modality Gap and Shared Space Alignment

**Lingjie Yi** [1]   **Raphael Douady** [2]   **Chao Chen** [1]

## Abstract

Multimodal contrastive learning (MCL) aims to embed data from two modalities in a shared embedding space. However, in practice, image and text representations occupy completely separated regions of the embedding space, a phenomenon called the modality gap. Meanwhile, empirical findings on how the modality gap affects downstream performance remain inconsistent. These observations motivate two key questions: (1) What causes the modality gap? (2) What determines downstream performance? To address these questions, we develop the first theoretical framework for analyzing the geometry of convergent optimal representations (COR) of MCL when training is optimized. We prove that the modality gap emerges when image and text representations collapse into different subspaces, a phenomenon called *dimension collapse*. Our theory further reveals that although the modality gap prevents direct alignment between image and text representations, their projections onto the shared subspace can align. Moreover, we show that shared space alignment is a dominant factor in downstream performance, while the effect of the modality gap is limited. Inspired by these findings, we propose Shared Space Alignment (SSA) to improve MCL pretraining by enhancing alignment in the shared space without optimizing for modality gap reduction. Extensive empirical results validate our theoretical analysis and the proposed method.

## 1. Introduction

Pretrained vision–language models (VLMs) (Radford et al., 2021) have achieved remarkable success in a wide range of tasks, including zero-shot image classification and zero-shot cross-modal retrieval. These models are typically trained with multimodal contrastive learning (MCL) on large-scale image text pairs. Despite strong empirical performance, our theoretical understanding of how VLMs learn representations and how these representations relate to downstream performance remains limited. In this work, we provide a theoretical study of these issues.

Our understanding of **unimodal** contrastive learning has advanced considerably. From a theoretical standpoint, when training is optimized (i.e., the training loss is minimized), the learned representations converge to an optimal configuration. We refer to this configuration as the *convergent optimal representation* (COR). Wang & Isola (2020) reveal that the COR of unsupervised contrastive learning (Chen et al., 2020) follows an uniform distribution on an $h$-dimensional unit hypersphere ($\mathbb{S}^{h-1}$). For supervised contrastive learning (Khosla et al., 2020), the COR forms a regular simplex inscribed in $\mathbb{S}^{h-1}$ (Graf et al., 2021), and a skewed simplex when the data is imbalanced (Yi et al., 2025b). These research on unimodal data demonstrate that examining geometric properties of CORs yields critical insights into how contrastive pretraining affects downstream performance.

This motivates us to study the COR of **multimodal** contrastive learning (MCL). Intuitively, MCL intends to align data from two modalities in a shared embedding space. However, this is not supported by empirical evidence. Instead, image and text representations cluster within disjoint cones on $\mathbb{S}^{h-1}$, forming a geometric phenomenon called the *modality gap*. Previous studies have hypothesized that the modality gap is caused by the cone effect (Liang et al., 2022), the contrastive objective (Fahim et al., 2024), insufficient training (Shi et al., 2023), or information bias (Schrodi et al., 2025). Meanwhile, the impact of the modality gap on downstream performance remains poorly understood. Some prior work (Liang et al., 2022; Schrodi et al., 2025) even show that narrowing the modality gap post-hoc can degrade downstream performance. Previous studies are mostly based on empirical analysis. None has offered a satisfactory theoretical explanation of what causes the modality gap and how it affects downstream performance.

In this paper, we first study the origin of the modality gap. We establish a novel theoretical framework for analyzing

---

[1]Stony Brook University [2]CNRS, University Paris 1 Pantheon-Sorbonne. Correspondence to: Lingjie Yi <chris.yi@stonybrook.edu>.

*Proceedings of the 43rd International Conference on Machine Learning*, Seoul, South Korea. PMLR 306, 2026. Copyright 2026 by the author(s).

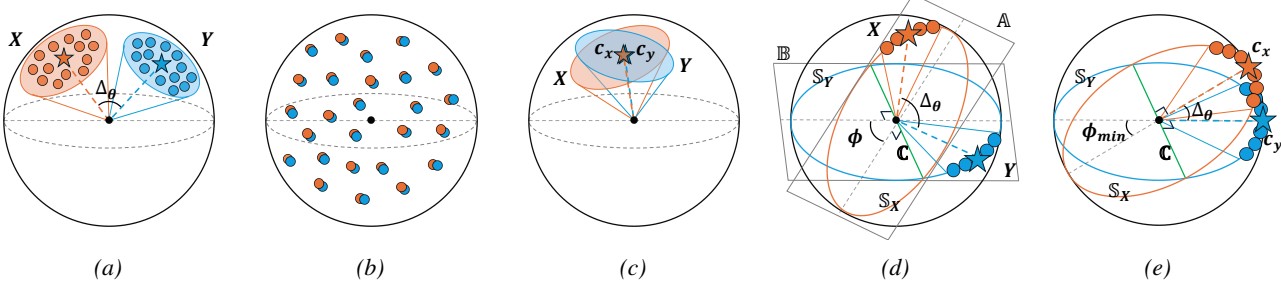

*Figure 1.* The COR of MCL. Orange and blue dots represent $X$ and $Y$. Starts are centers of $X$ and $Y$ (i.e., $c_x, c_y$). $\Delta_\theta$ denotes the modality gap. **(a)**: When a model is initialized, $(X, Y)$ lie within two cones. **(b)**: Without any constraints, $(X, Y)$ converge to a paired uniform distribution and $\Delta_\theta \to 0$. **(c)**: Under cone constraint, $\Delta_\theta \to 0$. **(d)**: $(X, Y)$ collapse into two distinct hyperplanes $\mathbb{A}$ and $\mathbb{B}$. $X \in \mathbb{S}_X$ (orange circle) $= \mathbb{A} \cap \mathbb{S}^{h-1}$ and $Y \in \mathbb{S}_Y$ (blue circle) $= \mathbb{B} \cap \mathbb{S}^{h-1}$. $\phi$ is the angle between $\mathbb{A}$ and $\mathbb{B}$. The green line is the shared space $\mathbb{C} = \mathbb{A} \cap \mathbb{B}$. See Definition 5 for details. **(e)**: Under subspace constraint, when training is optimized, $c_x, c_y \perp \mathbb{C}$ and $\Delta_\theta \to \phi_{\min}$.

the geometry of the COR of MCL. Specifically, we prove (Theorem 1) that, without any distributional constraints, the COR converges to a paired uniform distribution on $\mathbb{S}^{h-1}$ and the modality gap converges to zero (Figure 1b). This shows that **the contrastive objective tends to close the modality gap**. However, representations of each modality are observed to fall into a cone in $\mathbb{S}^{h-1}$ (Figure 1a), a phenomenon called *cone effect*. Theorem 2 shows that even under the cone constraint, the modality gap still converges to zero (Figure 1c). This shows that **the cone effect is not the geometric source of the modality gap**.

The preceding analysis motivates us to ask whether other constraints may underlie the modality gap. Jing et al. (2022) observe that unimodal representations can collapse into a lower-dimensional subspace of the embedding space, a phenomenon called *dimension collapse*. We also observe this in multimodal representations. Theorem 3 reveals that if representations of two modalities collapse into distinct hyperplanes (Figure 1d), the modality gap converges to the minimal angle between these hyperplanes (Figure 1e). This identifies **dimension collapse as the geometric source of the modality gap**. Furthermore, while dimension collapse prevents direct alignment of two modalities in the original space, Theorem 4 shows that their projections onto the shared space align as optimization proceeds. These findings show that **the contrastive objective promotes both the modality gap reduction and shared space alignment**.

Next, we study how the identified geometry affects downstream performance. First, we theoretically reveal that pair matching between $X$ and $Y$ and hence, the downstream performance of a model are **jointly affected by modality gap and shared space alignment**. Then, we empirically identify that **shared space alignment is the dominant factor**, whereas the effect of modality gap is limited. Inspired by this, we propose shared space alignment (SSA) to improve contrastive pretraining without explicitly optimizing for modality gap reduction. We validate our theorems and the proposed method with extensive empirical results.

## 2. Preliminary

Suppose we have a dataset $D = \{(I_i, T_i)\}_{i=1}^N$ of $N$ image-text pairs. The $h$-dimensional unit hypersphere is $\mathbb{S}^{h-1} = \{z \in \mathbb{R}^h : \|z\| = 1\}$. An image encoder $f_I(\cdot)$ and a text encoder $f_T(\cdot)$ map image and text data, after normalization, into a shared embedding space $\mathbb{S}^{h-1}$. We get $X = \{x_i\}_{i=1}^N = \{f_I(I_i)/\|f_I(I_i)\|\}_{i=1}^N \in (\mathbb{S}^{h-1})^N$ and $Y = \{y_i\}_{i=1}^N = \{f_T(T_i)/\|f_T(T_i)\|\}_{i=1}^N \in (\mathbb{S}^{h-1})^N$.

**Multimodal Contrastive Learning (MCL)** aims to embed data from image and text modalities into a shared embedding space. This is achieved by minimizing the MCL loss:

**Definition 1** (Multimodal Contrastive Loss (MCL Loss)). Let $(X, Y) = (x_i, y_i)_{i=1}^N$ be $N$ paired samples in $\mathbb{S}^{h-1}$. $\forall \tau > 0$, the multimodal contrastive loss $\mathcal{L}_{\mathrm{MCL}}(\cdot, \cdot) : (\mathbb{S}^{h-1})^N \times (\mathbb{S}^{h-1})^N \to \mathbb{R}$ is defined as:

$$\mathcal{L}_{\mathrm{MCL}}(X, Y) = \frac{1}{N} \sum_{i=1}^N \mathcal{L}_{\mathrm{MCL}}^i(X, Y),$$

$$\mathcal{L}_{\mathrm{MCL}}^i(X, Y) = \mathcal{L}_{\mathcal{X} \to \mathcal{Y}}(x_i; Y) + \mathcal{L}_{\mathcal{Y} \to \mathcal{X}}(y_i; X). \tag{1}$$

where $\mathcal{L}_{\mathcal{X} \to \mathcal{Y}}$ and $\mathcal{L}_{\mathcal{Y} \to \mathcal{X}}$ are defined as:

$$\mathcal{L}_{\mathcal{X} \to \mathcal{Y}}(x_i; Y) = -\log \frac{\exp(x_i \cdot y_i / \tau)}{\sum_{j=1}^N \exp(x_i \cdot y_j / \tau)},$$

$$\mathcal{L}_{\mathcal{Y} \to \mathcal{X}}(y_i; X) = -\log \frac{\exp(x_i \cdot y_i / \tau)}{\sum_{j=1}^N \exp(x_j \cdot y_i / \tau)}. \tag{2}$$

**Modality Gap** between $X$ and $Y$ can be quantified using different metrics. To measure the global separation between $X$ and $Y$, in this study, we adopt the following metric:

**Definition 2** (Modality Gap). Let $c_x = \frac{\mu_x}{\|\mu_x\|}$ where $\mu_x = \frac{1}{N} \sum_{i=1}^N x_i$ be the center vector of $X$, $c_y = \frac{\mu_y}{\|\mu_y\|}$ where $\mu_y = \frac{1}{N} \sum_{i=1}^N y_i$ be the center vector of $Y$. The modality gap between $X$ and $Y$ is then measured by the angular difference between two center vectors:

$$\Delta_\theta = \cos^{-1}(c_x \cdot c_y). \tag{3}$$

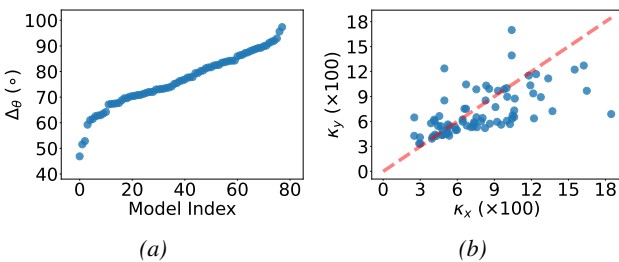

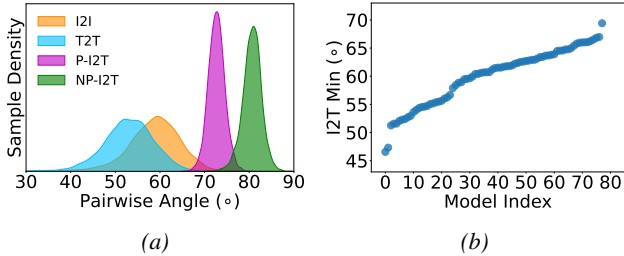

*(a)*      *(b)*            *(a)*      *(b)*

| | |
|---|---|
| *Figure 2.* Results of 78 pretrained VLMs on MSCOCO. **(a)**: Sorted values of the modality gap ($\Delta_\theta$). **(b)**: Estimated values of $\kappa_x$ and $\kappa_y$. The nominal values are scaled down by 100. The dashed line denotes $\kappa_x = \kappa_y$. Results on ImageNet are provided in Sec. D.3. | *Figure 3.* Cone separation on MSCOCO. **(a)**: Density of pairwise angles between image–image (I2I), text–text (T2T), paired (P-I2T) and unpaired image–text (NP-I2T) of CLIP ViT-B/32. **(b)**: Sorted minimal angles of all image-text pairs (I2T-min) of 78 pretrained VLMs. Results on ImageNet are provided in Sec. E.2 |

We compare different modality gap metrics in Sec. D.4. We further study the element-wise structure of two modalities through pairwise relationships between $(x_i, y_i)$. We define two such pairwise relationships as follows:

**Definition 3** (Pair Alignment and Pair Matching). Given $(x_i, y_i)_{i=1}^N$, we say $(x_i, y_i)$ **align** if and only if $x_i = y_i$. And we say $(x_i, y_i)$ are **well-matched** if and only if $\forall j \neq i$, $\|x_i - y_i\| < \|x_i - y_j\|$ and $\|x_i - y_i\| < \|x_j - y_i\|$, otherwise $(x_i, y_i)$ are **mis-matched**.

**Convergent Optimal Representation (COR)** refers to the optimal configuration of the learned representations at the global minimum of $\mathcal{L}_{\mathrm{MCL}}$. It characterizes an idealized regime with unlimited training resources, including **infinitely many training samples** ($N \to \infty$). Theorems 1–4 analyze the COR of MCL in this asymptotic setting. By contrast, Corollaries 1–2 examine the empirical implications for **finite real-world evaluation sets** ($N < \infty$), without imposing any distributional assumptions.

**Evaluation Protocol.** To empirically validate our theory presented in the following sections, we study 78 VLMs. All these models are pretrained with $\mathcal{L}_{\mathrm{MCL}}$, while varying in training data, architecture, activation function, and training configuration. The complete list of selected models is given in Sec. B. We evaluate all VLMs on ImageNet (Deng et al., 2009) and MSCOCO (Lin et al., 2014) benchmarks.

## 3. What Causes the Modality Gap

In this section, we investigate the origin of the modality gap by establishing the first theoretical framework for analyzing the COR of MCL, specifically the COR of the center pair.

### 3.1. Distribution Assumption

The von Mises-Fisher (vMF) distribution (Mardia & Jupp, 2009) is a generalization of the normal distribution in $\mathbb{S}^{h-1}$. Its samples are distributed in a hypercone, parameterized by a center vector $c$ and a concentration degree $\kappa$.

**Definition 4** (vMF Distribution). $\forall c \in \mathbb{S}^{h-1}$ and $\kappa \geq 0$, let $\nu = h/2 - 1$, the probability density of a random $h$-

dimensional unit vector $z \sim \mathrm{vMF}(c, \kappa)$ is given by:

$$f_h(z; c, \kappa) = \frac{\kappa^\nu}{(2\pi)^{\nu+1} I_\nu(\kappa)} e^{\kappa c^\top z}, \qquad (4)$$

where $I_\nu$ is the modified Bessel function of the first kind.

In Theorems 1-4, we assume that $(x_i, y_i)_{i=1}^N$ are i.i.d. samples from a joint distribution $p_{xy}$ whose marginal distributions are $p_x$ and $p_y$. Inspired by prior work modeling images (Du et al., 2024) and texts (Xu & Durrett, 2018) with the vMF distribution, in Theorems 2-4, we further assume that $p_x = \mathrm{vMF}(c_x, \kappa_x)$ and $p_y = \mathrm{vMF}(c_y, \kappa_y)$ to enable closed-form geometric analysis. The empirical justification of this assumption is provided in Sec. C.1.

(Liang et al., 2022) show that, at initialization, $X$ and $Y$ reside within two hypercones (see Figure 1a). During training, the distribution of $(X, Y)$ evolves as $c_x$, $c_y$, $\kappa_x$, and $\kappa_y$ change. Under this parameterization, any modality gap metric, including ones discussed in Sec. D.4, can be expressed as a function of $(c_x, c_y, \kappa_x, \kappa_y)$. Therefore, our qualitative conclusions are not tied to a specific choice of gap metric.

### 3.2. COR Without Constraints

We begin our analysis with an idealized setting where the encoders, $f_I$ and $f_T$, are expressive enough to realize any representation distributions without any constraints. Theorem 1 shows that when the limit of $\mathcal{L}_{\mathrm{MCL}}$ attains its minimum, the representations of each paired sample align ($x_i = y_i$), while the representations of all samples collectively converge to a uniform distribution over $\mathbb{S}^{h-1}$ (see Figure 1b).

**Theorem 1** (Informal). *Let $(x_i, y_i)_{i=1}^N$ be i.i.d. samples from a joint distribution $p_{xy}$ on $\mathbb{S}^{h-1}$ with marginal distributions $p_x$ and $p_y$. It holds that: $\lim_{N \to \infty} \mathcal{L}_{\mathrm{MCL}} - 2\log(N)$ is minimized if and only if the following conditions hold:*

*(A1)* $\forall i \in [N]$, $x_i = y_i$ $(\Rightarrow \Delta_\theta = \cos^{-1}(c_x \cdot c_y) = 0)$.

*(A2)* $p_x = \sigma_{h-1}$ and $p_y = \sigma_{h-1}$.

Here, $\sigma_{h-1}$ denotes the uniform probability measure on $\mathbb{S}^{h-1}$. The details and proof of Theorem 1 are provided

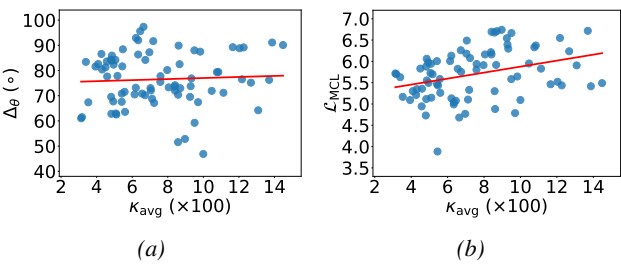

*Figure 4.* Results of 78 pretrained VLMs on MSCOCO. **(a)**: $\kappa_{\mathrm{avg}}$ vs. $\Delta_\theta$. **(b)**: $\kappa_{\mathrm{avg}}$ vs. $\mathcal{L}_{\mathrm{MCL}}$. The nominal values of $\kappa_{\mathrm{avg}}$ are scaled down by 100. The solid line denotes regression fit. Results on ImageNet are provided in Sec. F.2.

| Dataset | $\kappa_{\mathrm{avg}}$ vs. $\Delta_\theta$ | $\kappa_{\mathrm{avg}}$ vs. $\mathcal{L}_{\mathrm{MCL}}$ |
|---|---|---|
| MSCOCO | $0.047(X)$ | $0.247(\checkmark)$ |
| ImageNet | $0.140(X)$ | $0.164(\checkmark)$ |

*Table 1.* Results of 78 pretrained VLMs. Kendall's $\tau$ rank correlation between two metrics. $\checkmark$ denotes statistical significance in permutation test ($p < 0.05$).

As shown in Figure 3a, intra-modal angles are consistently smaller than inter-modal angles. This indicates that $X$ and $Y$ are concentrated in two separated hypercones. We further examine the minimal angle of all image-text pairs (I2T-min) of all 78 pretrained VLMs. The large values in Figure 3b further show that $X$ and $Y$ remain strictly separated in two non-overlapping angular regions across all evaluated models. Additional details and results are provided in Sec. E.

We next consider the case where the encoders, $f_I$ and $f_T$, embed $(X, Y)$ into two distinct hypercones spanning all dimensions of $\mathbb{S}^{h-1}$, i.e., $(X, Y)$ are subject to the *cone constraint*. As suggested by Remark 1, $\kappa_x, \kappa_y > 0$. We first examine the COR of center pair $(c_x, c_y)$ and the associated loss, $\mathcal{L}^{\mathrm{c}}_{MCL} = \mathcal{L}_{\mathcal{X} \to \mathcal{Y}}(c_x; Y) + \mathcal{L}_{\mathcal{Y} \to \mathcal{X}}(c_y; X)$. Theorem 2 shows that, under the cone constraint, when the limit of $\mathcal{L}^{\mathrm{c}}_{\mathrm{MCL}}$ attains its minimum, the modality gap still converges to **zero** ($\Delta_\theta \to 0$), **regardless of the initial cone locations** $(c_x, c_y)$ **or cone sizes** $(\kappa_x, \kappa_y)$ (see Figure 1c)

**Theorem 2** (Informal). *Let $(x_i, y_i)_{i=1}^{N}$ be i.i.d. samples from a joint distribution $p_{xy}$ on $\mathbb{S}^{h-1}$ with marginal distributions $p_x = \mathrm{vMF}(c_x, \kappa_x)$ and $p_y = \mathrm{vMF}(c_y, \kappa_y)$. Suppose $\exists i = c$ such that $x_c = c_x$, $y_c = c_y$. For any fixed $\kappa_x, \kappa_y > 0$, it holds that: $\lim_{N \to \infty} \mathcal{L}^{\mathrm{c}}_{\mathrm{MCL}} - 2\log(N)$ is minimized if and only if the following condition holds:*

*(A3)* $\Delta_\theta = \cos^{-1}(c_x \cdot c_y) = 0$.

The details and proof of Theorem 2 are provided in Sec. K.2. As the distribution of $(X, Y)$ is symmetric, non-center pairs $(x_i, y_i)_{i \neq c}$ do not affect the configuration of $(c_x, c_y)$, as confirmed by Theorem 4. Hence, we conclude that:

> **Conclusion 2:** Cone geometry is not the key structural source of the modality gap.

### 3.5. Evaluation of Cone Constrained Convergence

To examine the effect of cone geometry empirically, we first analyze the relationship between $\Delta_\theta$ and the cone concentration across all 78 pretrained VLMs. The cone concentration of a model is measured by $\hat{\kappa}_{\mathrm{avg}} = (\hat{\kappa}_x + \hat{\kappa}_y)/2$. Figure 4a and Tab. 1 show no statistically significant relationship between $\hat{\kappa}_{\mathrm{avg}}$ and $\Delta_\theta$. (Schrodi et al., 2025) further shows that even when a model is initialized with $c_x = c_y$, a modality gap still emerges during training. These results support our conclusion that cone geometry alone cannot explain the modality gap. Meanwhile, although cone geometry prevents

in Sec. K.1. Condition (A1) implies that the modality gap converges to **zero** ($\Delta_\theta \to 0$). This theorem concludes:

> **Conclusion 1:** The contrastive objective tends to close, rather than preserve, the modality gap.

**Remark 1.** In Theorem 1, if $p_x = \mathrm{vMF}(c_x, \kappa_x)$ and $p_y = \mathrm{vMF}(c_y, \kappa_y)$, Condition (A2) is equivalent to $\kappa_x = \kappa_y = 0$.

### 3.3. Evaluation of Unconstrained Convergence

To examine whether pretrained VLMs realize the unconstrained optimum characterized in Theorem 1, we first evaluate Condition (A1) by computing $\Delta_\theta$ for all 78 pretrained VLMs. Evaluation details are provided in Sec. D.1. As shown in Figure 2a, $\Delta_\theta$ ranges from $47°$ to $97°$, indicating that modality gaps of these evaluated models do not converge to the theoretical optimum of $0°$. Results for alternative gap metrics are reported in Sec. D.4.

Next, we evaluate Condition (A2). If this condition holds, then, as noted in Remark 1, concentration parameters $\kappa_x$ and $\kappa_y$ should converge to zero for each model. To examine this, we estimate $\kappa_x$ and $\kappa_y$ for all 78 pretrained VLMs. Estimation details are provided in Sec. D.2. As shown in Figure 2b, $\hat{\kappa}_x$ ranges from 251 to 1846 and $\hat{\kappa}_y$ ranges from 334 to 1699. These results indicate that the image and text representations of these models are not uniformly distributed on $\mathbb{S}^{h-1}$. Instead, they are highly concentrated within small regions, or hypercones, on $\mathbb{S}^{h-1}$.

These results show that all 78 pretrained VLMs fails to attain the theoretical unconstrained optimum. This suggests that certrain constraints—arising from the training data, model architecture, or training configuration—restrict the learned representations and prevent convergence to the optimum.

### 3.4. Constraint 1: Cone Constraint

To investigate potential representational constraints, we take the CLIP ViT-B/32 model trained on OpenAI's WebImage-Text dataset (Radford et al., 2021) as a representative example. We first compute pairwise angles of intra-modal paris (I2I and T2T) and inter-modal paris (P-I2T and NP-I2T).

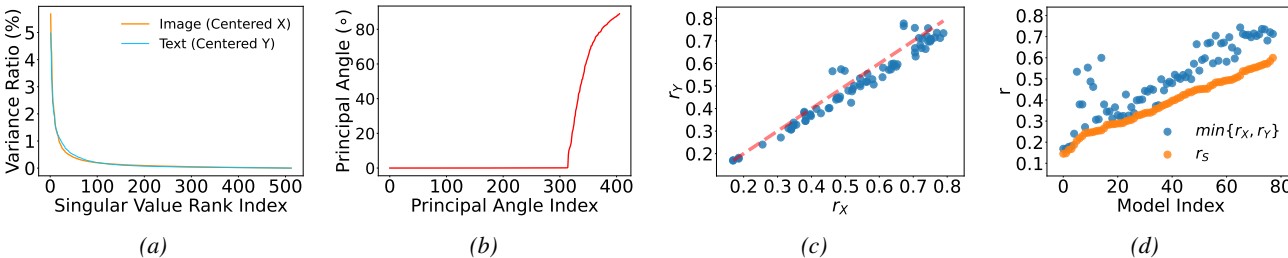

*Figure 5.* Dimension Collapse on MSCOCO. **(a)**: Singular values of centered $(X, Y)$ of CLIP ViT-B/32. **(b)**: Principal angles between two collapsed subspaces of CLIP ViT-B/32. **(c)**: The effective rank ratio, $r_X$ vs. $r_Y$, of 78 pretrained VLMs. The dashed line denotes $r_X = r_Y$. **(d)**: Sorted overlapping rank ratio $r_S$ vs. $\min\{r_X, r_Y\}$ of 78 pretrained VLMs. Results on ImageNet are provided in Sec. E.2.

$\kappa$ from converging to $0$ (Sec. 3.3). Furthermore, Figure 4b and Tab. 1 reveal a statistically significant positive correlation between $\hat{\kappa}_{\mathrm{avg}}$ and $\mathcal{L}_{\mathrm{MCL}}$. It suggests that **the contrastive objective promotes intra-modal uniformity**, even though it cannot fully recover the uniform distribution.

### 3.6. Constraint 2: Subspaces Constraint

Results in Sec. 3.5 motivate the investigation of additional constraints. Empirically, we observe that $X$ and $Y$ collapse into two partially overlapping subspaces of $\mathbb{S}^{h-1}$. To illustrate this phenomenon, we again use the CLIP ViT-B/32 model as a representative example. We compute the singular values $\sigma_i^{X/Y}$ of $X$ and $Y$. The near-zero $\sigma_i^{X/Y}$ in Figure 5a provides clear evidence that $X$ and $Y$ collapse into lower-dimensional subspaces of $\mathbb{S}^{h-1}$. We denote their subspace dimensions by $d_X$ and $d_Y$. We further compute the principal angles $\gamma_i$ between the collapsed subspaces. The near-zero $\gamma_i$ in Figure 5b indicates that the two subspaces partially overlap. We denote the dimensionality of the shared subspace by $d_S$. We then evaluate all 78 VLMs. In Figure 5c, we report the effective rank ratios of $X$ and $Y$, defined as $r_{X/Y} = d_{X/Y}/h$. In Figure 5d, we show the overlapping rank ratio, defined as $r_S = d_S/h$. These results show that subspace collapse is present across all evaluated models. Details of the subspace geometry are provided in Sec. G.

We next consider the case where the encoders, $f_I$ and $f_T$, embed $(X, Y)$ into two partially overlapping subspaces of $\mathbb{S}^{h-1}$ (see Figure 1d), i.e., $(X, Y)$ are subject to the subspace constraint. To enable closed-form geometric analysis, we specialize to a simplified setting that both subspaces are two hyperplanes. We define this geometric setting as:

**Definition 5.** Let $\mathbb{A}$ and $\mathbb{B}$ be two distinct $(h-1)$ dimensional linear subspaces in $\mathbb{R}^h$ (i.e., hyperplanes through the origin). Denote their normal vectors as $n_A$ and $n_B$ and their projection matrices as $P_A$ and $P_B$. Let $\mathbb{C} = \mathbb{A} \cap \mathbb{B}$ be the $(h-2)$ dimensional shared space whose projection matrix is $P_C$. Define $\phi = \cos^{-1}\left(\frac{n_A \cdot n_B}{\|n_A\| \cdot \|n_B\|}\right)$ as the angle between $\mathbb{A}$ and $\mathbb{B}$, restricted to $0 < \phi_{\min} \leq \phi < \frac{\pi}{2}$. Then $\mathbb{S}_X = \mathbb{S}^{h-1} \cap \mathbb{A}$ and $\mathbb{S}_Y = \mathbb{S}^{h-1} \cap \mathbb{B}$ are the subspaces where $X$ and $Y$ collapse, given by:

$$\mathbb{S}_X = \left\{x \in \mathbb{R}^h : \|x\| = 1, n_A \cdot x = 0\right\},$$
$$\mathbb{S}_Y = \left\{y \in \mathbb{R}^h : \|y\| = 1, n_B \cdot y = 0\right\}. \tag{5}$$

Theorem 3 shows that, under the subspace constraint, when the limit of $\mathcal{L}_{\mathrm{MCL}}^c$ attains its minimum, $c_x, c_y$ are orthogonal to $\mathbb{C}$, and the modality gap converges to the **minimal angle between $\mathbb{A}$ and $\mathbb{B}$** ($\Delta_\theta \to \phi_{\min}$) (see Figure 1e).

**Theorem 3** (Informal). *Let $(x_i, y_i)_{i=1}^N$ be i.i.d. samples from a joint distribution $p_{xy}$ on $\mathbb{S}^{h-1}$ with marginal distributions $p_x = \mathrm{vMF}(c_x, \kappa_x)$ and $p_y = \mathrm{vMF}(c_y, \kappa_y)$. $x_i \in \mathbb{S}_X$, $y_i \in \mathbb{S}_Y$ and $x_i, y_i \notin \mathbb{C}$. Suppose $\exists i = c$ such that $x_c = c_x$, $y_c = c_y$. For any fixed $\kappa_x, \kappa_y > 0$, it holds that: $\lim_{N \to \infty} \mathcal{L}_{\mathrm{MCL}}^c - 2\log(N)$ is minimized if and only if the following conditions hold:*

*(A4) $c_x \perp \mathbb{C}$ and $c_y \perp \mathbb{C}$ ($\Rightarrow \Delta_\theta = \phi$).*

*(A5) $\Delta_\theta = \cos^{-1}(c_x \cdot c_y) = \phi_{\min}$.*

The details and proof are provided in Sec. K.3. Condition (A4) shows the optimal configuration of $(c_x, c_y)$ for a given $\phi$. Condition (A5) shows that the loss decreases monotonically as $\phi$ decreases to $\phi_{\min}$. This theorem suggests that:

> **Conclusion 3:** Subspace geometry (dimension collapse) is the structural source of the modality gap.

## 4. What Affects Downstream Performance

In this section, we examine what influences downstream performance by analyzing the COR of all sample pairs.

### 4.1. COR Under the Subspaces Constraint

Downstream performance (e.g., of zero-shot image classification) is determined by the fraction of well-matched pairs (see Definition 3) in the evaluation set. Given an image (e.g., $x_i$), a model retrieves the class name (e.g., $y_j$) that exhibits the highest similarity to it. The prediction is correct if $y_j = y_i$ (well-matched), and incorrect otherwise. Notably, even when $(x_i, y_i)$ are well-matched, the subspace constraint implies that $x_i \neq y_i$, meaning $(x_i, y_i)$ do not align. Understanding what affects downstream performance requires understanding what affects pair matching. We achieve this by analyzing the COR of each sample pair.

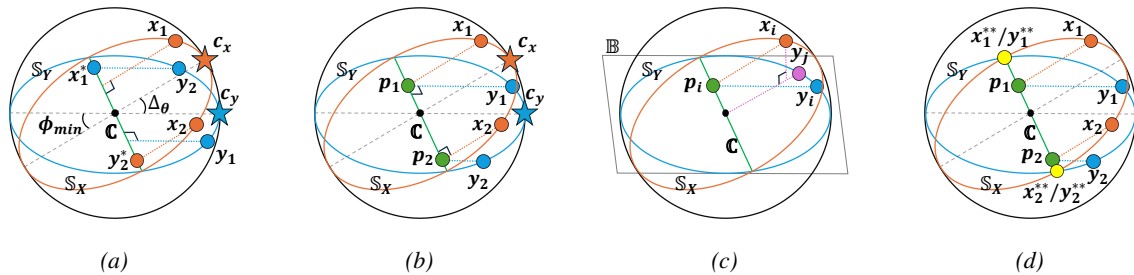

*Figure 6.* Pair matching in MCL. Notations follow Figure 1. **(a)**: $(c_x, c_y)$ achieve the COR when $\lim_{N\to\infty} \mathcal{L}^c_{\mathrm{MCL}} - 2\log(N)$ is minimized. **(b)**: The projections of $(x_i, y_i)_{i\neq c}$ onto $\mathbb{C}$ align to $p_i$ (green points), i.e., $x_i^* = y_i^* = p_i$, when $\lim_{N\to\infty} \mathcal{L}^{i\neq c}_{\mathrm{MCL}} - 2\log(N)$ is minimized. **(c)**: When $x_i^* = y_i^*$, $P_B x_i \nparallel y_i$. If $\exists y_j$ in an evaluation set, i.e., $y_j = P_B x_i / \|P_B x_i\|$ (purple dot), then $x_i \cdot y_j > x_i \cdot y_i$ or $\|x_i - y_j\| < \|x_i - y_i\|$ and therefore, $(x_i, y_i)_{i\neq c}$ are mis-matched. **(d)**: $(x_i^{**}, y_i^{**})$ (yellow dots) align.

We denote the projections and normalized projections of $(X, Y)$ onto the shared space $\mathbb{C}$ as $(X^*, Y^*)$ and $(X^{**}, Y^{**})$, which are given by:

$$(x_i^*, y_i^*) = (P_C x_i, P_C y_i), \quad (x_i^{**}, y_i^{**}) = \left(\frac{x_i^*}{\|x_i^*\|}, \frac{y_i^*}{\|y_i^*\|}\right). \tag{6}$$

Theorem 4 reveals that, under the subspace constraint, when the limit of $\mathcal{L}^{i\neq c}_{\mathrm{MCL}}$ attains its minimum, **the projections** of any non-center pair $(x_i, y_i)_{i\neq c}$ **onto the shared space** $\mathbb{C}$ **align** ($x_i^* = y_i^*$) (Figure 6a vs. Figure 6b).

**Theorem 4** (Informal). *Let $(x_i, y_i)_{i=1}^N$ be i.i.d. samples from a joint distribution $p_{xy}$ on $\mathbb{S}^{h-1}$ with marginal distributions $p_x = \mathrm{vMF}(c_x, \kappa_x)$ and $p_y = \mathrm{vMF}(c_y, \kappa_y)$. $x_i \in \mathbb{S}_X$, $y_i \in \mathbb{S}_Y$ and $x_i, y_i \notin \mathbb{C}$. Then $\forall i \in [N]$, it holds that: $\lim_{N\to\infty} \mathcal{L}^{i\neq c}_{\mathrm{MCL}} - 2\log(N)$ is minimized if and only if the following conditions hold:*

*(A6)* $P_C x_i = P_C y_i$.

*(A7)* $\Delta_\theta = \cos^{-1}(c_x \cdot c_y) = \phi_{\min}$.

The details and proof of Theorem 4 are provided in Sec. K.4. Condition (A6) characterizes the optimal configuration of $(x_i, y_i)_{i\neq c}$ for any given $\phi$. Condition (A7) establishes that the loss decreases monotonically as $\phi$ decreases to $\phi_{\min}$, consistent with Condition (A5). Since all paired samples are non-center pairs almost surely ('centers' form a zero measure set in $\mathbb{S}_X$ or $\mathbb{S}_Y$), we conclude that:

> **Conclusion 4:** The contrastive objective promotes modality gap reduction and shared space alignment.

### 4.2. What Affects Pair Matching

For any sample pair $(x_i, y_i)$ in a finite evaluation set, Theorem 4 implies that if a model is not optimized, their projections may not align and $(x_i, y_i)$ may occupy arbitrary positions in $\mathbb{S}^{h-1}$. Consequently, $(x_i, y_i)$ **can be mis-matched**. One might ask: Are $(x_i, y_i)$ well matched if a model is optimized such that their projections align in the shared space? Corollary 1 shows that this intuition is incorrect:

even at optimality, $(x_i, y_i)$ **can still be mis-matched** as long as **the modality gap persists** (Figure 6c).

**Corollary 1.** $\forall i \in [N], i \neq c$, *if $x_i \in \mathbb{S}_X, y_i \in \mathbb{S}_Y$, $x_i, y_i \notin \mathbb{C}$, $P_C x_i = P_C y_i$ and $\phi > 0$, it holds that:*

*(A8)* $(x_i, y_i)_{i\neq c}$ *can be mis-matched.*

The proof of Corollary 1 is provided in Sec. K.4.3. Furthermore, Corollary 2 reveals that, under subspace constraint, if a model is optimized such that the projections of $(X, Y)$ align in the shared space, **the normalized projections** of $(X, Y)$ also align ($x_i^{**} = y_i^{**}$) (Figure 6d). It suggests that, at optimality, transforming $(X, Y)$ into $(X^{**}, Y^{**})$ post-hoc can eliminate the modality gap and the corresponding negative impact on downstream performance.

**Corollary 2.** $\forall i \in [N], x_i \in \mathbb{S}_X^{h_x}$ *and $y_i \in \mathbb{S}_Y^{h_y}$, if $P_C x_i = P_C y_i$, then the following holds:*

*(A9)* $\frac{P_C x_i}{\|P_C x_i\|} = \frac{P_C y_i}{\|P_C y_i\|}$

The proof of Corollary 2 is provided in Sec. K.4.3. Importantly, both Corollary 1 and Corollary 2 are distribution-free results, and thus apply to arbitrary distributions of $(X, Y)$. These analysis draw the following conclusion:

> **Conclusion 5:** Theoretically, pair matching is decided by (1) **shared space alignment** and (2) **modality gap**.

### 4.3. Evaluation of Subspace Constrained Convergence

Theorem 3 and Theorem 4 derive the COR of MCL under a simplified setting where dimensions of both collapsed subspaces, $\mathbb{S}_X$ and $\mathbb{S}_Y$, are $h - 1$, i.e., $d_X = d_Y = h - 1$. However, in practice, $d_X$ and $d_Y$ are not fixed (Figure 5c). Now, we validate our theory in real pretrained VLMs.

Conditions (A5) and (A7) imply that $\Delta_\theta$ converges to the minimal angle $\phi_{\min}$ between the two collapsed hyperplanes (Figure 6a). In practice, $\phi_{\min}$ is determined by the relative geometry of $\mathbb{S}_X$ and $\mathbb{S}_Y$, including their dimensions, overlap, and principal angles. Thus, $\Delta_\theta$ is jointly shaped by $\mathcal{L}_{\mathrm{MCL}}$ and the learned subspace geometry. This makes the

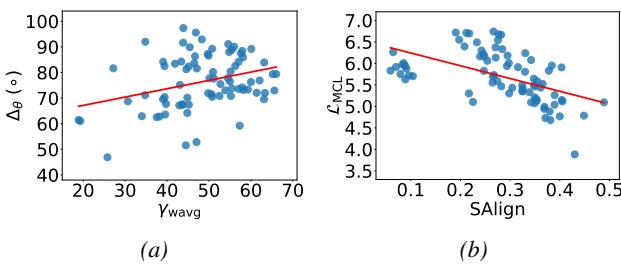

*(a)*      *(b)*

*Figure 7.* Results of 78 pretrained VLMs on MSCOCO. **(a)**: $\gamma_{\text{wavg}}$ vs. $\Delta_\theta$. **(b)**: SAlign vs. $\mathcal{L}_{\text{MCL}}$. The solid line denotes regression fit. Results on ImageNet are provided in Sec. H.2.

| Dataset | $\gamma_{\text{wavg}}$ vs. $\Delta_\theta$ | SAlign vs. $\mathcal{L}_{\text{MCL}}$ |
|---|---|---|
| MSCOCO | 0.178(✓) | −0.465(✓) |
| ImageNet | 0.241(✓) | −0.496(✓) |

*Table 2.* Results of 78 pretrained VLMs. Kendall's $\tau$ rank correlation between two metrics. ✓ denotes statistical significance in permutation test ($p < 0.05$).

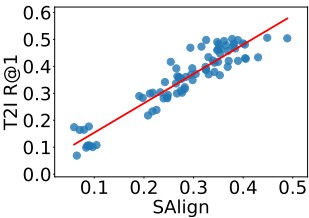

*Figure 8.* Results of 78 pretrained VLMs on MSCOCO. SAlign vs. I2T R@1. The solid line denotes regression fit. Results on ImageNet are provided in Sec. H.2.

| Metric | SAlign vs. Perf. |
|---|---|
| MSCOCO | 0.733 (✓) |
| ImageNet | 0.689 (✓) |

*Table 3.* Results of 78 pretrained VLMs. Caption follows Tab. 2.

Condition (A6) implies that $(X, Y)$ are optimized to align in the shared space. To examine this, we analyze the relationship between $\mathcal{L}_{\text{MCL}}$ and shared space alignment (SAlign) across all 78 VLMs. We first define a metric to measure SAlign between $X$ and $Y$ of a model as follows:

$$\text{SAlign}(X, Y) = \frac{1}{N} \sum_{i=1}^{N} \mathbb{1}_{\{(x_i^*, y_i^*) \text{ are well-matched}\}}. \quad (7)$$

Evaluation details are provided in Sec. H.1. As shown in Figure 7b and Tab. 2, SAlign is significantly negatively correlated with $\mathcal{L}_{\text{MCL}}$. This is consistent with our theoreti-

direct relationship between $\Delta_\theta$ and $\mathcal{L}_{\text{MCL}}$ intrinsically confounded. Instead, we examine the relationship between $\Delta_\theta$ and $\phi_{\min}$ in 78 VLMs. As $\phi_{\min}$ is not directly observable in practice, we use the weighted average of none-zero principal angles, $\gamma_{\text{wavg}}$, as its proxy. Evaluation details are provided in Sec. H.1. The results in Figure 7a and Tab. 2 show that $\Delta_\theta$ is significantly positively correlated with $\gamma_{\text{wavg}}$. This is consistent with the predicted connection between $\Delta_\theta$ and its geometric lower bound. These findings support our theoretical claim that **subspace geometry is the structural source of the modality gap**.

| Evaluated Pairs | $\Delta_\theta$ | I → T | | T → I | |
|---|---|---|---|---|---|
| | | R@1 | R@5 | R@1 | R@5 |
| $(X, Y)$ | 72.98° | 48.40 | 73.72 | 29.89 | 54.04 |
| $(X^*, Y^*)$ | 54.39° | 39.26 | 65.46 | 27.26 | 51.80 |
| $(X^{**}, Y^{**})$ | 54.22° | 36.70 | 65.92 | 29.54 | 53.97 |

*Table 4.* Performance of zero-shot cross-modal retrieval of ViT-B/32 with post-hoc transformation on MSCOCO.

cal claim that **the contrastive objective promotes shared space alignment**.

### 4.4. Evaluation of Alignment vs. Performance

In Sec. 4.2, we theoretically show that downstream performance is determined by both shared space alignment and modality gap. In this and the following sections, we empirically examine our theory in real pretrained VLMs.

We first examine the relationship between SAlign and downstream performance across all 78 VLMs. For ImageNet, performance is measured by Top-1 accuracy in zero-shot image classification. For MSCOCO, performance is measured by Top-1 recall in text to image retrieval (T2I R@1). Evaluation details are provided in Sec. H.1. As shown in Figure 8 and Tab. 3, SAlign exhibits a strongly positive correlation with downstream performance. These results are consistent with our theoretical claim that **shared space alignment plays a vital role in downstream performance**.

### 4.5. Evaluation of Modality Gap vs. Performance

Corollary 1 reveals a theoretical mechanism by which the modality gap can degrade downstream performance: if $\exists y_j$ (i.e., $y_j = P_B x_i / \|P_B x_i\|$) in an evaluation set such that $x_i \cdot y_j > x_i \cdot y_i$ (Figure 6c), $(x_i, y_i)$ are mis-matched. However, this mechanism depends on the presence of such a $y_j$ and therefore may not occur in every finite evaluation set. We now examine the effect of the modality gap empirically.

To isolate the effect of the modality gap on downstream performance, we vary the gap while keeping shared space alignment fixed. As explained in Sec. 4.3, $(X, Y)$ and $(X^*, Y^*)$ have the same SAlign, allowing us to directly compare their downstream performance. We use the CLIP ViT-B/32 model as a representative example and evaluate zero-shot cross-modal retrieval performance on MSCOCO. Evaluation details are provided in Sec. I. The results in Tab. 4 show that reducing $\Delta_\theta$ alone, while preserving SAlign, does not improve downstream performance. This suggests that the **modality gap plays a limited role** in downstream performance. Additional evaluations are provided in Sec. I.3.

Corollary 2 shows that, at optimality, transforming $(X, Y)$ into $(X^{**}, Y^{**})$ post-hoc can eliminate the modality gap, and improve downstream performance. However, the theo-

| Model | CIFAR-10 | | CIFAR-100 | | ImageNet-1K | |
|---|---|---|---|---|---|---|
| | Top-1 | Top-5 | Top-1 | Top-5 | Top-1 | Top-5 |
| CLIP | 61.6 | 95.6 | 28.1 | 55.9 | 31.4 | 58.7 |
| SharedCLIP | 56.9 | 95.2 | 26.4 | 54.7 | 32.1 | 59.7 |
| IMSep | 61.6 | 96.2 | 31 | 60.1 | 32 | 59 |
| AlignCLIP | 69.4 | 97.8 | 36.5 | 66.3 | 32.8 | 60.6 |
| CLIP + SSA | **71.3** | **97.8** | **37.8** | **67.1** | **33.3** | **61.2** |

*Table 5.* Performance on zero-shot image classification with SSA on various datasets.

| Model | I → T | | | T → I | | |
|---|---|---|---|---|---|---|
| | R@1 | R@5 | R@10 | R@1 | R@5 | R@10 |
| CLIP | 31.4 | 57.0 | 68.6 | 20.5 | 44.1 | 55.9 |
| SharedCLIP | 33.6 | 59.6 | 70.8 | 21.8 | **45.4** | **57.3** |
| IMSep | 33.7 | **60.8** | **71.5** | 21.5 | 45.1 | 56.9 |
| AlignCLIP | 34.0 | 59.7 | **71.5** | **22.3** | 45.0 | 56.9 |
| CLIP + SSA | **34.5** | 60.7 | **71.5** | **22.8** | **45.4** | 57.1 |

*Table 6.* Performance on zero-shot cross-modal retrieval on MSCOCO with SSA.

retical optimum, i.e., the COR is fully achieved, is hardly satisfied by real pretrained VLMs. The results in Tab. 4 show that although this post-hoc transformation reduces $\Delta_\theta$, it does not improve downstream performance either. Evaluation details and additional results of this experiment are provided in Sec. I. The analysis in Sec. 4.4 and Sec. 4.5 concludes that:

> **Conclusion 6:** Empirically, **shared space alignment** plays a dominant role in downstream performance.

## 5. How to Improve Pretraining

In this section, we explore how to improve contrastive pretraining by enhancing shared space alignment without explicitly optimizing for modality gap reduction.

### 5.1. Enhancing Shared Space Alignment

In Sec. 4, we show that downstream performance is primarily associated with shared space alignment, whereas the effect of the modality gap is limited. Moreover, as discussed in Sec. 4.3, the modality gap is induced by the learned subspace geometry and is therefore cannot be controlled directly. These findings suggest that improving contrastive pretraining should focus on enhancing shared space alignment rather than explicitly reducing the modality gap.

Following this idea, we propose Shared Space Alignment (SSA) to directly improve alignment of normalized projections $(X^{**}, Y^{**})$ within the shared space. This is achieve by optimizing the SSA loss, $\mathcal{L}_{\text{SSA}}(X^{**}, Y^{**})$, defined as:

$$\mathcal{L}_{\text{SSA}}(X^{**}, Y^{**}) = -\frac{1}{2} \log \frac{\exp\left(x_i^{**} \cdot y_i^{**}/\tau\right)}{\sum_{j=1}^N \exp\left(x_i^{**} \cdot y_j^{**}/\tau\right)}$$
$$-\frac{1}{2} \log \frac{\exp\left(x_i^{**} \cdot y_i^{**}/\tau\right)}{\sum_{j=1}^N \exp\left(x_j^{**} \cdot y_i^{**}/\tau\right)}). \quad (8)$$

We use this loss combined with $\mathcal{L}_{\text{MCL}}$ for pretraining:

$$\mathcal{L} = \mathcal{L}_{\text{MCL}}(X, Y) + \mathcal{L}_{\text{SSA}}(X^{**}, Y^{**}). \quad (9)$$

### 5.2. Experiments of SSA

**Implementation Details.** To evaluate the effectiveness of our proposed method, we train a CLIP ViT-B/16 model

from scratch on the Conceptual Captions 12M (CC12M) dataset (Changpinyo et al., 2021). We compare our method against three baselines reported in (Eslami & de Melo, 2025): SharedCLIP reduces the modality gap by sharing the learnable parameters between the modality encoders; IMSep improves intra-modal separation for image representations; and AlignCLIP combines parameter sharing with intra-modal separation together. All models are trained with the same model architecture and training settings. Additional implementation details are provided in Sec. J.1.

**Zero-Shot Image Classification.** After pretraining, we first evaluate all models on zero-shot image classification. We consider three standard benchmarks: CIFAR-10, CIFAR-100 (Krizhevsky et al., 2009), and ImageNet-1K (Deng et al., 2009). Evaluation details are provided in Sec. J.2. As summarized in Tab. 5, SSA consistently improves zero-shot image classification performance.

**Zero-Shot Cross Modal Retrieval.** Beyond classification, we evaluate all models on zero-shot image-to-text and text-to-image retrieval on MSCOCO (Lin et al., 2014). Evaluation details of are provided in Sec. J.2. As shown in Tab. 6, SSA improves zero-shot retrieval performance.

The results on both downstream tasks support our theoretical prediction that improving shared space alignment leads to better contrastive pretraining.

## 6. Conclusion

This paper addresses two fundamental questions in contrastive VLMs: (1) what causes the modality gap? and (2) what determines downstream performance? We identify *dimension collapse* as the structural source of the modality gap. Our theory further reveals that pair matching, and thus downstream performance, depends on both shared space alignment and the modality gap. However, across 78 pretrained VLMs, we find that exhibits a substantially stronger relationship with downstream performance. These findings suggest that improving contrastive pretraining should focus on enhancing shared space alignment rather than explicitly reducing the modality gap. Based on this insight, we propose SSA, a simple and effective approach for improving VLM pretraining.

## Acknowledgements

This work was partially supported by NSF Grant CCF-2144901, NIH Grants R01NS143143, R01CA297843, as well as funding from Coefficient Giving. Lingjie Yi was also partially supported by the Bloomberg Data Science Fellowship during his Ph.D. studies.

## Impact Statement

This paper presents work whose goal is to advance the field of Machine Learning. There are many potential societal consequences of our work, none which we feel must be specifically highlighted here.

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

# A. More Discussions

## A.1. Related Work

### A.1.1. REPRESENTATION LEARNING

Unimodal representations can be learned in an unsupervised manner using self-supervised contrastive learning (SSL) (Chen et al., 2020). When the InfoNCE loss (Wu et al., 2018) reaches its minimum, the representations of differently augmented views of an image converge to a single point, and the representation of all images converge to a uniform distribution on $\mathbb{S}^{h-1}$ (Wang & Isola, 2020). However, Jing et al. (2022) empirically show that this theoretical optimum may not be realized in practice: the learned representations tend to collapse into a lower-dimensional subspace rather than spanning the entire embedding space.

In the supervised setting, representations can be learned through a neural classifier. When the cross-entropy loss is minimized, representations of samples from different balanced classes converge to the vertices of a regular simplex inscribed in $\mathbb{S}^{h-1}$, a phenomenon known as *neural collapse* (Papyan et al., 2020). Graf et al. (2021) provide a theoretical explanation of this phenomenon. Representations can also be learned with supervised contrastive learning (SupCon) (Khosla et al., 2020). Graf et al. (2021) prove that the COR of a balanced dataset of SupCon also forms a regular simplex. Yi et al. (2025b) provide a refined proof and further show that, for imbalanced datasets, representations converge to a skewed simplex or even collapse into two distinct points. Other works extend the concept of neural collapse to semi-supervised learning (Yi et al., 2025a) and OOD detection (Liu & Qin, 2025).

Multimodal representations are learned through multimodal contrastive learning (MCL). However, the COR of MCL remains poorly understood. In this work, we address this gap by characterizing the COR of MCL. Our theorems suggest that MCL seeks to maximize the alignment between the two modalities in the shared space.

### A.1.2. MODALITY GGAP

Liang et al. (2022) first identify the modality gap, a geometric phenomenon characterized by the complete separation of representations of different modalities in the embedding space. They hypothesize that the gap arises from the cone effect due to random model initialization and is preserved by the contrastive objective. Fahim et al. (2024) argues that the modality gap is inherent to contrastive loss. Yaras et al. (2024); Udandarao (2022) examine the role of mismatch pairs and the temperature parameter. Shi et al. (2023) attribute the cause of the modality gap to insufficient training. Schrodi et al. (2025) suggests that problematic training data, which contain information bias, create the gap. Most of these works validate their hypotheses through numerical examples on a small number of data pairs. By contrast, we provide a theoretical analysis on the entire distribution.

In addition, several studies have proposed post-hoc methods to mitigate the modality gap. Liang et al. (2022) attempts to translate the representations of one modality toward those of another using a constant shift. Schrodi et al. (2025) explores removing the few dimensions that primarily drive the modality gap. However, experiments in both works reported that narrowing the modality gap post-hoc may lead to degraded downstream performance. Eslami & de Melo (2025) mitigates the modality gap by retraining CLIP from scratch. Our work focuses on improving pretrainin while leaving the modality gap aside.

## A.2. Connections Between Our Theorems and Previous Hypotheses

In this subsection, we examine the relationship between hypotheses proposed in prior studies and our theory.

**Cone Effect:** The cone effect hypothesis (Liang et al., 2022) posits that the representations of $X$ and $Y$ fall into distinct cones on the hypersphere, thereby causing the modality gap. In our theoretical framework, as described in Sec. 3.1, the size of a representation cone is characterized by the parameter $\kappa$. Contrary to this hypothesis, Theorem 2 demonstrates that the cone size has no effect on the convergence of the modality gap, even when the representations follow a uniform distribution (i.e., $\kappa \to 0$).

**Mismatched Pairs:** (Liang et al., 2022; Udandarao, 2022) study the role of mismatched pairs—i.e., $(x_i, y_j)$ where $y_j$ is closer to $x_i$ than $y_i$. They claim that, in the presence of mismatched pairs, the temperature induces the modality gap by affecting the landscape of the MCL loss. In Corollary 1, we prove that paired samples cannot be perfectly aligned, which gives rise to mismatched pairs. Therefore, the modality gap causes the formation of mismatched pairs, rather than the

reverse.

**Information Bias:** Schrodi et al. (2025) argue that information bias, i.e., images containing more information than the corresponding text, leads to the modality gap. The unequal amount of information across modalities prevents Intra-Modal Isometry of the representations (see Sec. K.4.1), making it difficult for the model to align representations from the two modalities. This results in sub-optimal inter-modal alignment, which in turn imposes a lower bound on the alignment terms. Schrodi et al. (2025) further claim that, when modality alignment is bounded, MCL compromises alignment to increase the uniformity of non-paired samples, thereby pushing the modalities apart and causing the modality gap.

However, this claim is not supported by our theorems. In Theorem 3 (Condition A7) and Theorem 4 (Condition A9), we prove that MCL always favors smaller alignment over larger uniformity, leading to a reduced the modality gap. The uniformity can only be increased by enlarging the size of the representation cone, i.e., $\kappa$. Instead, we posit that there is a strong connection between information bias and dimension collapse: information bias induces dimension collapse in the learned representations, thereby causing the modality gap.

### A.3. Limitations

While our work investigates the origin of the modality gap and attributes it to dimension collapse, we do not address the exact factors that lead to dimension collapse. (Jing et al., 2022) theoretically show that dimension collapse occurs whenever negative eigenvalues appear in the weight matrix of a neural network. (Schrodi et al., 2025) suggests that when training data with information bias are sufficiently aligned, 'more dimensions' are required to focus on objects and and 'less dimensions' to focus on attributes, ultimately resulting in dimension collapse. (Chun, 2025) provides a more comprehensive study of the inherent challenges within MCL, including intra-modal variability, asymmetries in information, and task-dependent alignment. We suspect that all these factors contribute to dimension collapse in the learned representations. Identifying the causes of dimension collapse thus constitutes a major open problem, parallel to understanding the origin of the modality gap, and represents an important direction for future research.

## B. Collection of Pretrained Vision-Language Models

**Collection of Contrastive VLMs**. To validate our proposed theorems, we examine a collection of 78 contrastive VLMs that were pretrained with $\mathcal{L}_{\mathrm{MCL}}$ on various datasets, model architectures, activation functions, and training settings, provided by OpenCLIP. The construction of this collection largely follows (Schrodi et al., 2025). From an initial set of 121 models, we retain models that exhibit the following characteristics:

- **CLIP variants**: OpenAI's CLIP (Radford et al., 2021), OpenCLIP (Cherti et al., 2023), MetaCLIP (Xu et al., 2024), CLIP-A (Li et al., 2023), EVA-CLIP and EVA-02-CLIP (Sun et al., 2023).

- **Vision backbones**: ResNet (He et al., 2016), ConvNeXt (Liu et al., 2022) and ViT (Alexey, 2021).

- **Pretraining datasets:** OpenAI's proprietary WebImageText dataset (400 M) (Radford et al., 2021), LAION-400 M, LAION-2 B, LAION-Aesthetic (900 M) (Schuhmann et al., 2022), Merged-2 B (Sun et al., 2023), WebLI (Chen et al., 2023), MetaCLIP (400 M) (Xu et al., 2024), Conceptual 12 M (Changpinyo et al., 2021), YFCC (15 M) (Thomee et al., 2016), CommonPool-s (max. 12.8 M), CommonPool-m (max. 128 M), CommonPool-l (max. 1.28 B), CommonPool-xl (max. 12.8 B) (Gadre et al., 2023), and DataComp-s (1.4 M), DataComp-m (14 M), DataComp-1 (140 M), DataComp-xl/DataComp-1B (1 B) (Gadre et al., 2023).

Similarly to (Schrodi et al., 2025), we further filter this list of models on the basis of the following conditions:

- Any models that were fine-tuned (including model soup (Wortsman et al., 2022)) or rewind are removed.

- When there were models that were trained on the same dataset but saved at different checkpoints, only the latest checkpoint is maintained.

- Models that achieve $< 5\%$ R1 on MS COCO are removed.

We specifically exclude the SigLIP models (Zhai et al., 2023) for two reasons. First, these models are pretrained using the SigLIP loss rather than $\mathcal{L}_{\mathrm{MCL}}$ loss, and therefore fall outside the scope of our theoretical framework. Second, the SigLIP

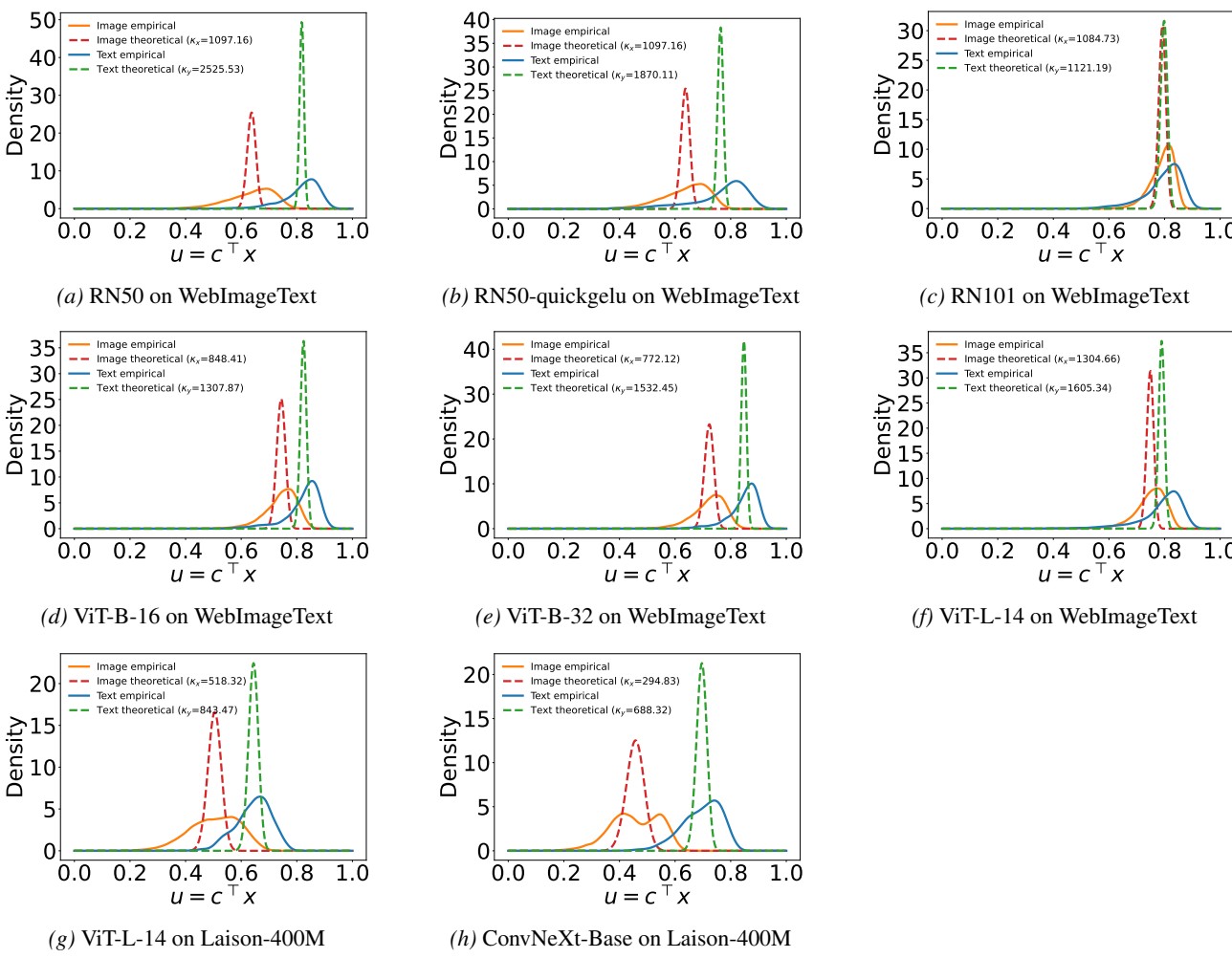

*Figure 9.* Empirical and theoretical density of the projected similarity $u = c^\top z$ for representative models. Results on ImageNet. Solid curves denote empirical densities estimated via kernel density estimation, while dashed curves correspond to the theoretical marginal distributions implied by the vMF model with the estimated concentration parameter $\kappa$.

model family comprises only 9 instances, which is insufficient to support statistically meaningful empirical analysis. After filtering, we use 78 contrastive VLMs for our experiments.

## C. Empirical Justification of Theoretical Assumptions

### C.1. Distribution Assumption

Our theoretical analysis models image and text representations using von Mises-Fisher (vMF) distributions on the hypersphere. In this subsection, we empirically evaluate how well this approximation captures the observed geometry of multimodal representations.

For each model, we compute the normalized image and text representations and estimate their mean directions $c_x$ and $c_y$. We also estimate their concentration parameter $\kappa_x$ and $\kappa_y$ using the method described in Sec. 3.3. We then project the representations into the corresponding mean direction and obtain similarity values $u = c^\top z$, where $z$ denotes an image or text embedding. We estimate the empirical density of $u$ using kernel density estimation and compare it with the theoretical marginal density implied by the vMF model with concentration parameter $\kappa$. Figure 10 shows representative examples for several models spanning different model architectures and training datasets. The empirical distributions (solid curves) generally align with the theoretical vMF densities (dashed curves) near the dominant concentration region, while deviations

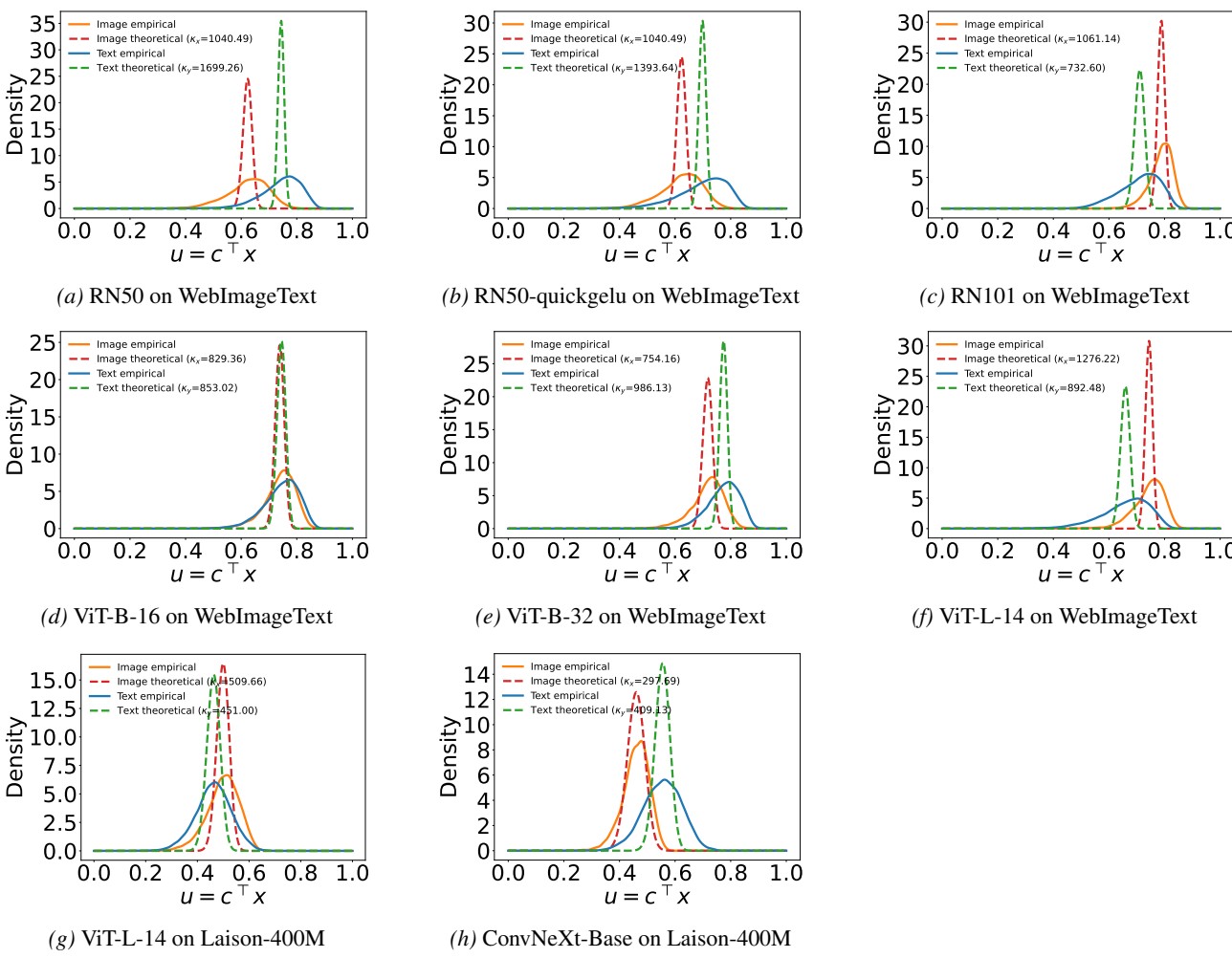

*Figure 10.* Empirical and theoretical density of the projected similarity $u = c^\top z$ for representative models. Results on MSCOCO. Solid curves denote empirical densities estimated via kernel density estimation, while dashed curves correspond to the theoretical marginal distributions implied by the vMF model with the estimated concentration parameter $\kappa$.

mainly occur in the low-similarity regime.

Our analysis aims to characterize the geometric structure of multimodal representations learned by contrastive VLMs. Rather than modeling the full empirical distribution of representations, we focus on extracting interpretable, first-order geometric quantities that are both analytically tractable and empirically informative. Overall, we observe that the vMF model captures the dominant concentration behavior of both modalities. While deviations from the theoretical density exist, particularly in the distribution tails, these differences primarily reflect higher-order semantic structure rather than failures of the geometric approximation. This empirical evidence supports the use of the vMF model as a compact first-order description of multimodal representation geometry. This approximation allows us to disentangle the dominant geometric factors governing multimodal alignment from residual distributional complexity. This perspective clarifies how the modality gaps arise, how they vary across models and training regimes, and why simple geometric descriptors can effectively explain observed performance trends despite the rich structure of real-world representations.

### C.2. Assumption in Theorem 4

In Theorem 4, we assume that the angle between a modality input and its center, $\theta_i^c$, satisfies $\theta_i^c \in \left(0, \frac{\pi}{2}\right)$. In Lemma 15, we provide a theoretical justification for this assumption. The density plot of $\cos\left(\theta_i^c\right)$ in Figure 9 and Figure 10 shows that almost all $\theta_i^c$ are indeed within $\left(0, \frac{\pi}{2}\right)$.

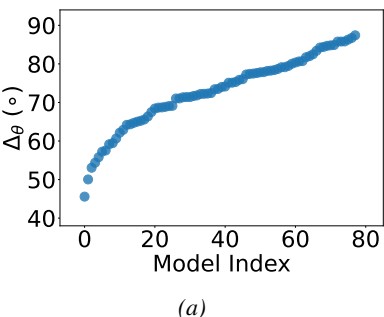 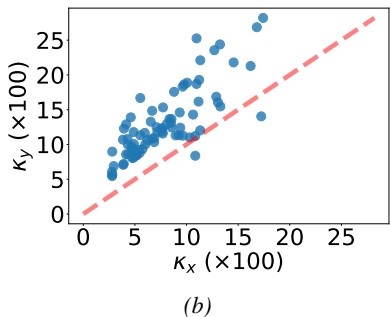

(a)                (b)

*Figure 11.* Results of 78 pretrained VLMs on ImageNet. **(a)**: Sorted values of the modality gap ($\Delta_\theta$). **(b)**: Estimated values of $\kappa_x$ and $\kappa_y$. The nominal values are scaled down by 100. The dashed line denotes $\kappa_x = \kappa_y$.

## D. Additional Details and Results for Unconstrained Convergence

### D.1. Evaluation Details

In this experiment, we evaluate the model convergence of 78 pretrained VLMs on ImageNet-1K (Deng et al., 2009) and MSCOCO (Lin et al., 2014) datasets. For both datasets, we use the test set for evaluation.

For ImageNet-1K, we adopt the large set of prompts provided by OpenAI for CLIP (Radford et al., 2021) (`https://colab.research.google.com/github/openai/CLIP/blob/main/notebooks/Prompt_Engineering_for_ImageNet.ipynb`) for text representations.

For MSCOCO dataset, rather than following the common practice of prepending prompts such as 'a photo of caption', we directly use raw captions to extract text representations. This design better preserves the natural geometry of the text space, instead of introducing distortions from artificial prompt templates.

### D.2. Details of $\kappa$ Estimation

The maximum likelihood estimate of $\kappa$ is given by:

$$A_h \left( \kappa_{x/y} \right) = \frac{I_{h/2} \left( \kappa_{x/y} \right)}{I_{h/2-1} \left( \kappa_{x/y} \right)} = \|\mu_{x/y}\|, \tag{10}$$

where $\mu_x$ and $\mu_y$ are described in Definition 2. An approximate solution (Sra, 2012) to Eq. (10) is:

$$\hat{\kappa}_{x/y} = \frac{\|\mu_{x/y}\| \left( h - \|\mu_{x/y}\|^2 \right)}{1 - \|\mu_{x/y}\|^2}. \tag{11}$$

### D.3. Results on ImageNet

Similar to Figure 2, we provide the results of unconstrained convergence on ImageNet in Figure 11. In Figure 11b, we observe that $\kappa_y$ are generally larger than $\kappa_x$. This is consistent with the fact that the sample size of texts (1000 class names) in ImageNet is much smaller than that of images (50000 images) and, therefore, the distributions of text representations are more concentrated across all evaluated models.

### D.4. Comparison of Different Modality Gap Metrics

In this subsection, we compare our modality gap metric $\Delta_\theta$ (Definition 2) with two alternative metrics. The first is $\Delta_\mu$ proposed in (Liang et al., 2022), which is defined as:

$$\Delta_\mu = \|\mu_x - \mu_y\|_2, \text{ where } \mu_x = \frac{1}{N} \sum_{i=1}^{N} x_i \text{ and } \mu_y = \frac{1}{N} \sum_{i=1}^{N} y_i. \tag{12}$$

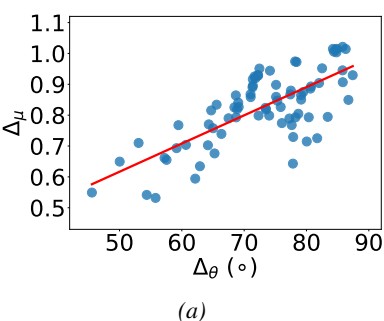 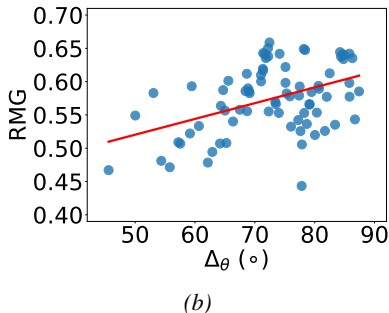

*(a)*                   *(b)*

*Figure 12.* Results of 78 pretrained VLMs on ImageNet. Comparison between different metrics: **(a)**: $\Delta_\mu$ vs. $\Delta_\mu$; **(b)**: $\Delta_\mu$ vs. RMG. The solid line denotes regression fit.

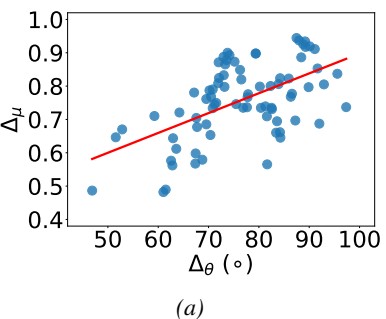 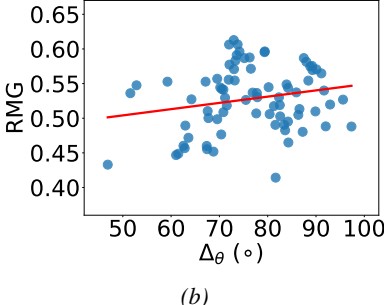

*(a)*                   *(b)*

*Figure 13.* Results of 78 pretrained VLMs on MSCOCO. Comparison between different metrics: **(a)**: $\Delta_\mu$ vs. $\Delta_\mu$; **(b)**: $\Delta_\mu$ vs. RMG. The solid line denotes regression fit.

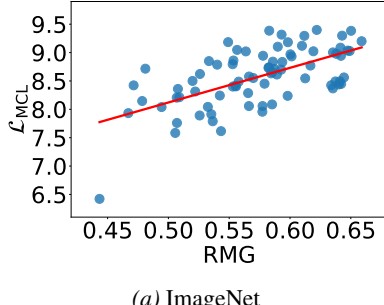 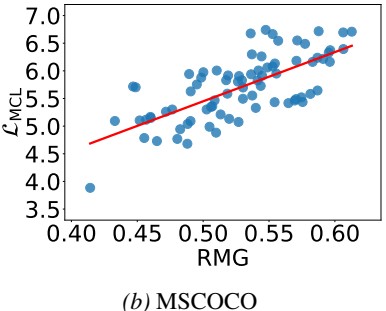

*(a)* ImageNet               *(b)* MSCOCO

*Figure 14.* Results of 78 pretrained VLMs. Comparison between RMG and $\mathcal{L}_{\mathrm{MCL}}$. **(a)**: Results on ImageNet. **(b)**: Results on MSCOCO. The solid line denotes regression fit.

| Metric | ImageNet | MSCOCO |
|---|---|---|
| RMG vs. $\mathcal{L}_{\mathrm{MCL}}$ | $0.422(\checkmark)$ | $0.514(\checkmark)$ |

*Table 7.* Results of 78 pretrained VLMs. Kendall's $\tau$ rank correlation between two metrics. $\checkmark$ denotes statistical significance in permutation test ($p < 0.05$).

The second is RMG proposed in (Schrodi et al., 2025), which is defined as:

$$\text{RMG} = \frac{\frac{1}{N}\sum_{i=1}^{N} d\left(\mathbf{x}_i, \mathbf{y}_i\right)}{\frac{1}{2N(N-1)}\left(\sum_{i,j=1;i\neq j}^{N} d\left(\mathbf{x}_i, \mathbf{x}_j\right) + \sum_{i,j=1;i\neq j}^{N} d\left(\mathbf{y}_i, \mathbf{y}_j\right)\right) + \frac{1}{N}\sum_{i=1}^{N} d\left(\mathbf{x}_i, \mathbf{y}_i\right)} \tag{13}$$

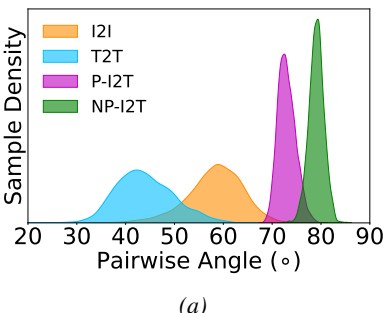 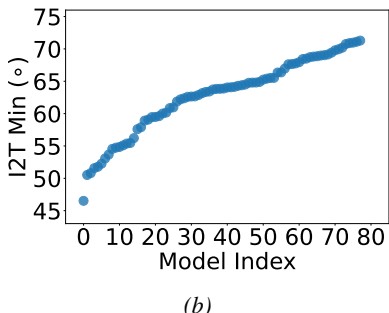

*(a)* *(b)*

*Figure 15.* Cone separation on ImageNet. **(a)**: Density of pairwise angles between image–image (I2I), text–text (T2T), paired image–text (P-I2T), and unpaired image–text (NP-I2T) of CLIP ViT-B/32. **(b)**: Sorted minimal angles between all image-text pairs (I2T-min) of 78 pretrained VLMs.

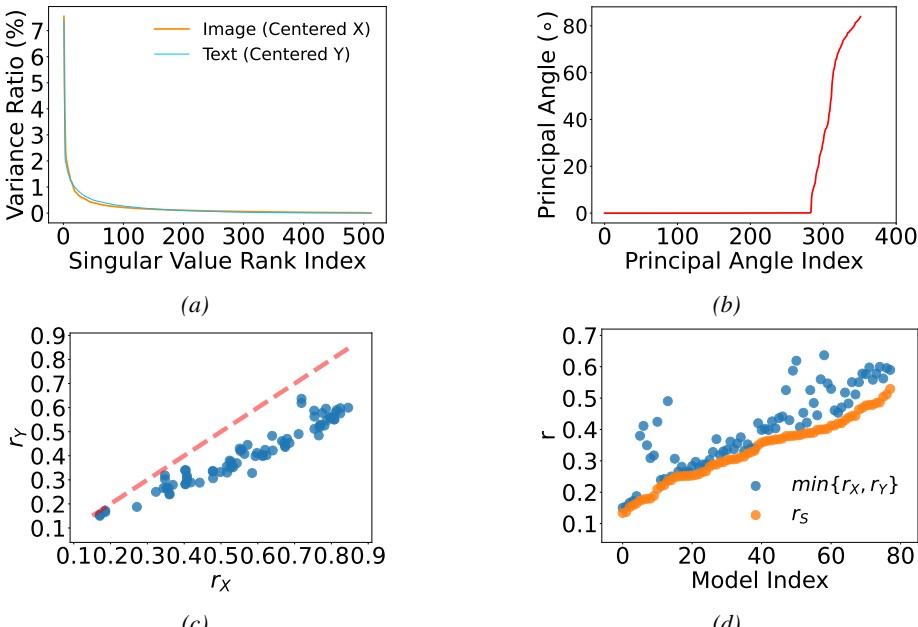

*(a)* *(b)*

*(c)* *(d)*

*Figure 16.* Dimension Collapse on ImageNet. **(a)**: Singular values of centered $X$ and $Y$ of CLIP ViT-B/32. **(b)**: Principal angles between two collapsed subspaces of CLIP ViT-B/32. **(c)**: The effective rank ratio, $r_X$ vs. $r_Y$, of 78 pretrained VLMs. The dashed line denotes $r_X = r_Y$. **(d)**: Sorted overlapping rank ratio $r_S$ vs. $\min\{r_X, r_Y\}$ of 78 pretrained VLMs.

As shown in Figure 12 and Figure 13, neither measure converges to 0 in all evaluated models, which is consistent with the results based on $\Delta_\theta$.

Notably, RMG (Schrodi et al., 2025) measures the modality gap through pairwise distances between $X$ and $Y$. We argue that, however, such a definition is largely redundant with $\mathcal{L}$MCL, because RMG exhibits a strong positive correlation with $\mathcal{L}$MCL. Figure 14 and Tab. 7 provide empirical support for this claim. In contrast, our analysis measures the modality gap through the cluster-level angular discrepancy between $X$ and $Y$, and characterizes pairwise cross-modal relationships using the shared space alignment metric (SAlign) introduced in Sec. 4.3.

## E. Additional Details and Results for Model Constraints

### E.1. Evaluation Details

The details of representation extraction of images and texts follow Sec. D.1.

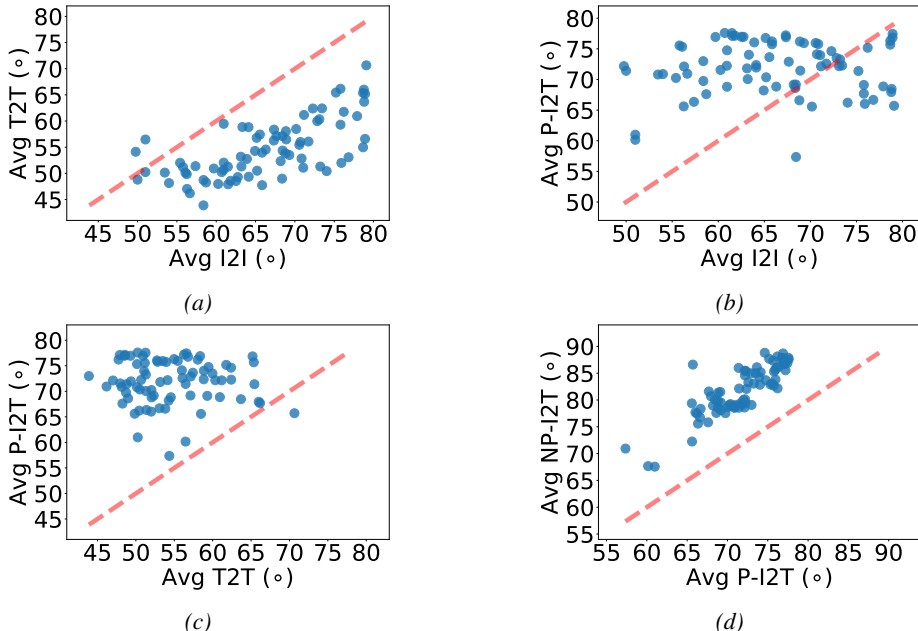

*Figure 17.* Results of 78 pretrained VLMs on ImageNet. Average of pairwise angles between **(a)**: image–image (I2I) vs. text–text (T2T); **(b)**: paired image–text (P-I2T) vs. unpaired image–text (NP-I2T); **(c)**: I2I vs. P-I2T; **(d)**: T2T vs. P-I2T. The dashed line denotes the identity line.

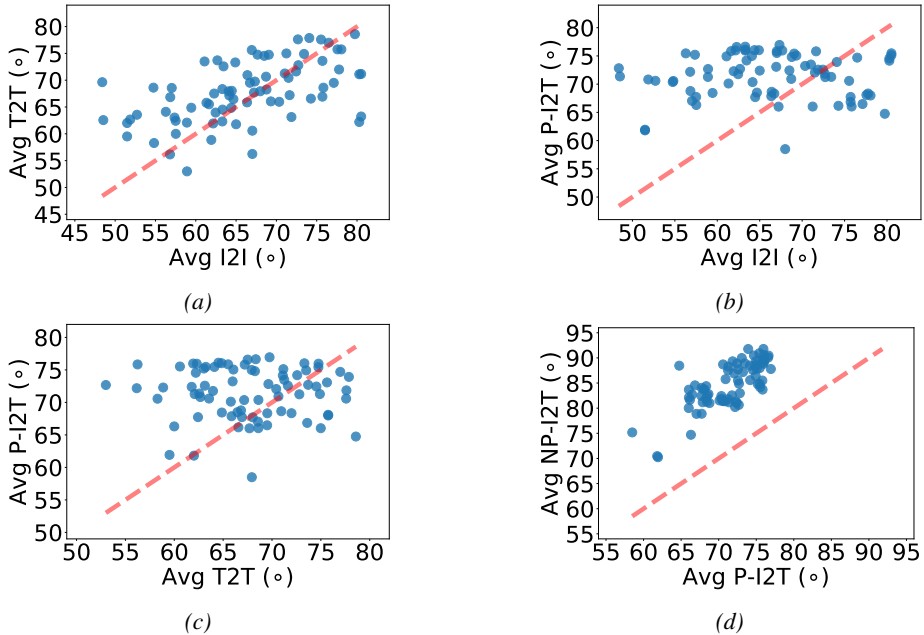

*Figure 18.* Results of 78 pretrained VLMs on MSCOCO. Average of pairwise angles between **(a)**: image–image (I2I) vs. text–text (T2T); **(b)**: paired image–text (P-I2T) vs. unpaired image–text (NP-I2T); **(c)**: I2I vs. P-I2T; **(d)**: T2T vs. P-I2T. The dashed line denotes the identity line.

### E.2. Results on ImageNet

Similarly to Figure 3 and Figure 5, we provide evidence of model constraints, including cone separation (in Figure 15) and dimension collapse (in Figure 16) evaluated on ImageNet. The results show that all 78 models exhibit the same cone and subspace constraint patterns with those evaluated on MSCOCO.

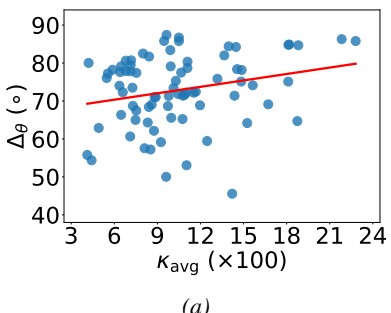 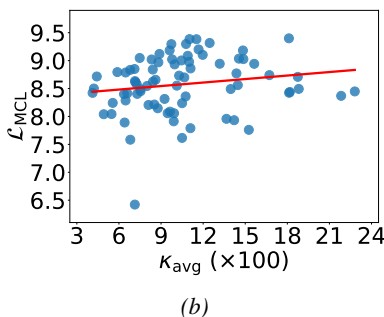

*(a)*                                     *(b)*

*Figure 19.* Results of 78 pretrained VLMs on ImageNet. **(a)**: $\kappa_{\mathrm{avg}}$ vs. $\Delta_\theta$. **(b)**: $\kappa_{\mathrm{avg}}$ vs. $\mathcal{L}_{\mathrm{MCL}}$. The nominal values of $\kappa_{\mathrm{avg}}$ are scaled down by 100. The solid line denotes regression fit.

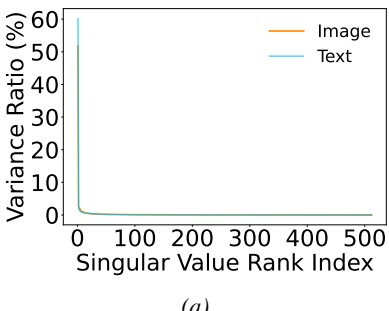 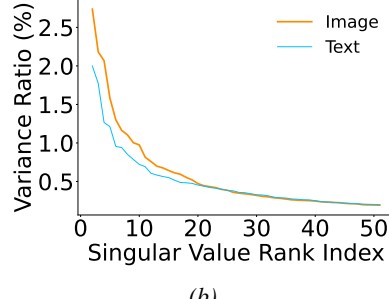 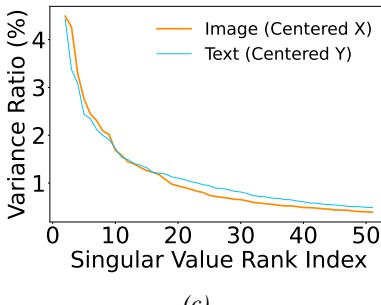

*(a)*                        *(b)*                        *(c)*

*Figure 20.* The explained variance ratio of singular values of CLIP ViT-B/32 on MSCOCO. **(a)**: All singular values of $X$ and $Y$. **(b)**: The $2^{\mathrm{nd}}$ to $50^{\mathrm{th}}$ singular values of $X$ and $Y$. **(c)**: The $1^{\mathrm{st}}$ to $50^{\mathrm{th}}$ singular values of the centered $X$ and the centered $Y$.

### E.3. Additional Results of Pairwise Angles

In this subsection, we provide more detailed results of pairwise angles all 78 contrastive pretrained models on ImageNet (in Figure 17) and MSCOCO (in Figure 18) datasets. These results provide more information on the geometry of the image and text representations of all evaluated models.

## F. Additional Details and Results for Cone Constrained Convergence

### F.1. Evaluation Details

The details of representation extraction of images and texts follow Sec. D.1.

### F.2. Results on ImageNet

Similarly to Figure 4, we provide evidence of cone constrained convergence evaluated on ImageNet in Figure 19. The results show that all 78 models exhibit the same convergence patterns as the results on MSCOCO.

## G. Details of Subspace Projection and Alignment

In this subsection, we describe in detail how to detect dimension collapse, how to detect the shared space of two subspaces, and how to conduct projection onto the shared space.

### G.1. Detect Dimension Collapse

Suppose that we have two point clouds, $X$ and $Y$, each consisting of $h$-dimensional normalized vectors: $X = (x_1, \ldots, x_N) \in (\mathbb{S}^{h-1})^N$ and $Y = (y_1, \ldots, y_N) \in (\mathbb{S}^{h-1})^N$. Then we have:

$$
\begin{aligned}
\mathbb{A} &= \operatorname{span}(X), \quad d_X = \dim(\mathbb{A}), \\
\mathbb{B} &= \operatorname{span}(Y), \quad d_Y = \dim(\mathbb{B}), \\
\mathbb{C} &= \mathbb{A} \cap \mathbb{B}, \quad d_S = \dim(\mathbb{C}).
\end{aligned}
\tag{14}
$$

Apply the Singular Value Decomposition (SVD) to $X$ and $Y$ and we get:

$$
\begin{aligned}
X &= U_X \Sigma_X V_X^\top, \\
Y &= U_Y \Sigma_Y V_Y^\top.
\end{aligned}
\tag{15}
$$

If $X$ and $Y$ collapse into subspaces of $\mathbb{S}^{h-1}$, then $\Sigma_X$ and $\Sigma_Y$ have $d_X < h$ and $d_Y < h$ significant singular values, respectively.

In the discussion in Sec. 3.6, $X$ and $Y$ represent the image and text representations. Since $X$ and $Y$ are not centered at zero, the first singular values, $\sigma_1^x$ and $\sigma_1^y$, dominate when SVD is applied. Correspondingly, the first right singular vectors of $X$ and $Y$ are $c_x$ and $c_y$, respectively. As shown in Figure 20a, these first right singular vectors account for approximately $50\%$ of the explained variance. Therefore, in Figure 5c, we plot the singular values of the centered $X$ and $Y$, which better capture the patterns of variation. In Figure 20b, we present the $2^{\text{nd}}$ to the $50^{\text{th}}$ singular values of $X$ and $Y$, while in Figure 20c, we show the $2^{\text{nd}}$ to the $50^{\text{th}}$ singular values of the centered $X$ and the centered $Y$.

And dimension collapse in $X$ and $Y$ occurs when zero values appear on the diagonals of $\Sigma_X$ and $\Sigma_Y$.

### G.2. Detect the Overlapping Dimensions

We then select the first $d_X$ columns from $V_X$ and the first $d_Y$ columns from $V_Y$, whose cumulative explained variance exceeds a predefined threshold $1 - c$ (e.g., $c_x = c_y = 1\%$), as the orthogonal bases of subspaces where $X$ and $Y$ collapse. We obtain:

$$
\begin{aligned}
B_X &= V_X[:,: d_X] \in \mathbb{R}^{h \times d_X} : \text{orthonormal basis for } \mathbb{A}^{d_X}, \\
B_Y &= V_Y[:,: d_Y] \in \mathbb{R}^{h \times d_Y} : \text{orthonormal basis for } \mathbb{B}^{d_Y}.
\end{aligned}
\tag{16}
$$

To investigate whether $\mathbb{A}$ and $\mathbb{B}$ have overlap dimensions, we need to check the principal angles between $\mathbb{A}^{d_X}$ and $\mathbb{B}^{d_Y}$, which are defined as:

**Definition 6.** The principal angles $\gamma_1 \leq \gamma_2 \leq \cdots \leq \gamma_k$ between $\mathbb{A}$ and $\mathbb{B}$ are recursively defined as:

$$
\cos\left(\frac{\gamma_i}{180}\pi\right) = \max_{u_i \in \mathbb{A}, v_i \in \mathbb{B}} u_i^\top v_i, \quad \|u_i\| = \|v_i\| = 1, \quad u_i^\top u_j = v_i^\top v_j = 0 \ (j < i),
\tag{17}
$$

where $k = \min(d_X, d_Y)$.

The principal angles quantify the closeness between these subspaces: (1) The smallest principal angle $\theta_1$ measures how close the two subspaces are: if $\gamma_1 = 0$, there is at least one common direction; and (2) if multiple principal angles are zero, then the intersection of the subspaces has a larger dimension. The principal angles between the subspaces $\mathbb{A}^{d_X}$ and $\mathbb{B}^{d_Y}$ can be computed as follows:

**Step 1**: Apply the SVD decomposition $G = B_X^\top B_Y \in \mathbb{R}^{d_X \times d_Y}$.

$$
G = U_G \Sigma_G V_G^\top
\tag{18}
$$

**Step 2**: Extract the singular values $\sigma_i^G = \Sigma_G[i, i] \in [0, 1]$. And calculate the principal angles as:

$$
\gamma_i = \arccos\left(\sigma_i^G\right) \frac{180}{\pi}
\tag{19}
$$

The number of principal angles equal to zero gives the dimension of the intersection:

$$d_S = \#\left\{i : \gamma_i = 0\right\}. \tag{20}$$

In practice, due to noise or finite precision, we use a threshold: count how many $\sigma_i^G > 1 - \epsilon$ (e.g., $\epsilon = 0.1$). Thus:

$$d_S = \#\left\{i : \sigma_i^G > 1 - \epsilon\right\}. \tag{21}$$

The empirical results on ImageNet is provided in Figure 16d and the empirical results on MSCOCO is provided in Figure 5d.

### G.3. Find the Shared Space

In this subsection, we provide the details on how to calculate the shared space between $X$ and $Y$.

**Step 1**: Apply the SVD decomposition to $X$ and $Y$ to get $V_X$ and $V_Y$ as Eq. (15).

**Step 2**: Select the first $d_X$ and $d_Y$ right singular vectors of $X$ and $Y$ whose cumulative explained variance are great than $99\%$. The resulting vectors, $B_X$ and $B_Y$, form the bases for $\mathbb{A}^{d_X}$ and $\mathbb{B}^{d_Y}$, as indicated by Eq. (16).

**Step 3**: Apply the SVD decomposition to $G = B_X^\top B_Y \in \mathbb{R}^{d_X \times d_Y}$ as Eq. (18).

**Step 4**: Compute $d_S$ according to Eq. (21) while setting $\epsilon = 0.1$. Compute the basis of the shared space $P$ by:

$$P = B_X U_G[:, : d_S] = B_Y V_G[:, : d_S]. \tag{22}$$

The ablation studies of the choice of $c_x$, $c_y$, $\epsilon$ are provided in Sec. I.4.

### G.4. Weighted Average of Principal Angles

In Sec. 4.3, to quantify the effective angular discrepancy between the two collapsed subspaces, we use the weighted average of none-zero principal angles, $\gamma_{\text{wavg}}$, as the proxy of $\phi_{\min}$. In this subsection, we provide the details on how to calculate $\gamma_{\text{wavg}}$.

Recall that $d_S$ is the estimated dimension of the shared space and $k = \min(d_X, d_Y)$. Since the zero principal angles ($\gamma_{i=1}^{d_S}$) only reflect the shared intersection $\mathbb{S}_X \cap \mathbb{S}_Y$, we exclude them and only retain the nonzero ones ($\gamma_{i=d_S+1}^k$), which characterize the residual geometric separation outside the shared subspace. Let $u_i$ and $v_i$ denote the $i^{th}$ vector of $U_G$ and $V_G$ respectively, then $p_i^{(X)} = B_X u_i$ and $p_i^{(Y)} = B_Y v_i$ denote the corresponding principal vectors. For each retained nonzero angle $\gamma_{i=d_S+1}^k$, to account for the fact that different principal directions may carry different amounts of variation, we weight each angle by the shared directional energy of its associated principal vectors. Concretely, we define the directional energy of $X$ and $Y$ as follows:

$$\lambda_i^{(X)} = \frac{1}{n}\left\|X p_i^{(X)}\right\|_2^2, \quad \lambda_i^{(Y)} = \frac{1}{n}\left\|Y p_i^{(Y)}\right\|_2^2, \tag{23}$$

and use the symmetric weight as the shared directional energy between $X$ and $Y$:

$$w_i = \sqrt{\lambda_i^{(X)} \lambda_i^{(Y)}}. \tag{24}$$

The weighted average of the nonzero principal angles is then given by

$$\gamma_{\text{wavg}} = \frac{\sum_{i \in \mathcal{I}} w_i \gamma_i}{\sum_{i \in \mathcal{I}} w_i}, \quad \mathcal{I} = \{d_S + 1, d_S + 2, \cdots, k\}. \tag{25}$$

Intuitively, $\gamma_{\text{wavg}}$ serves as a single summary statistic of the residual separation between $\mathbb{S}_X$ and $\mathbb{S}_Y$, while assigning greater importance to directions that are energetically significant for both modalities.

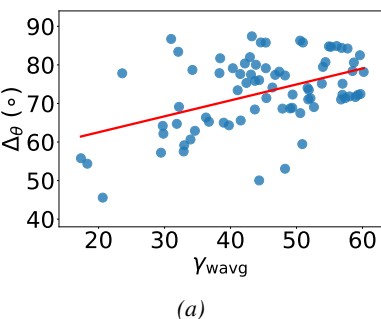 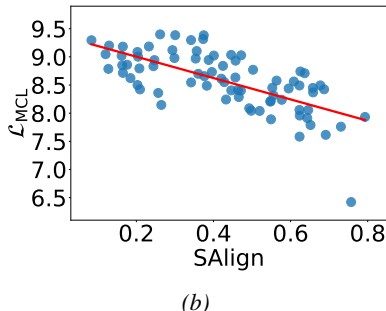

*(a)*            *(b)*

*Figure 21.* Results of 78 pretrained VLMs on ImageNet. **(a)**: $\gamma_{\mathrm{wavg}}$ vs. $\Delta_\theta$. **(b)**: SAlign vs. $\mathcal{L}_{\mathrm{MCL}}$. The solid line denotes regression fit.

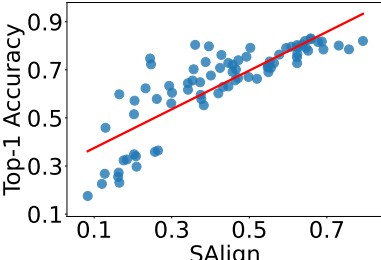

*Figure 22.* Results of 78 pretrained VLMs on ImageNet. **(a)**: SAlign vs. Top-1 accuracy. The solid line denotes regression fit. The solid line denotes regression fit.

## H. Additional Details and Results for Subspace Constrained Convergence

### H.1. Evaluation Details

The details of representation extraction of images and texts follow Sec. D.1. We evaluate the quantitative relationship between two metrics using Kendall's $\tau$ rank correlation coefficient. Statistical significance is assessed using a permutation test at the significance level $0.05$.

Our definition of 'well-match' in Definition 3 assumes a one-to-one correspondence between $X$ and $Y$. However, this assumption does not hold for ImageNet or MSCOCO. In ImageNet, multiple image representations $x$ from the same class correspond to a shared text representation $y$ (the class name). Conversely, in MSCOCO, multiple captions $y$ correspond to the same image representation $x$. To better evaluate model-level SAlign under these settings, we adapt the definition of 'well-match' accordingly. In ImageNet (Deng et al., 2009), we say $(x_i, y_i)$ are **well-matched** if and only if $\forall j \neq i$, $\|x_i - y_i\| < \|x_i - y_j\|$. In MSCOCO (Lin et al., 2014), we say $(x_i, y_i)$ are **well-matched** if and only if $\forall j \neq i$ $\|x_i - y_i\| < \|x_j - y_i\|$.

Consistent with these evaluation settings, downstream performance on ImageNet is measured by Top-1 accuracy in zero-shot image classification, while performance on MSCOCO is measured by Top-1 recall (T2I R@1) in text-to-image retrieval.

The estimation details of $\gamma_{\mathrm{wavg}}$ are elaborated in Sec. G.4

### H.2. Results on ImageNet

Similarly to Figure 7 and Figure 8, we provide evidence of subspace constrained convergence evaluated on ImageNet in Figure 21 and Figure 22. The results show that all 78 models exhibit the same convergence patterns as the results on MSCOCO.

| Model | CIFAR-10 | | | CIFAR-100 | | | ImageNet-1K | | |
|---|---|---|---|---|---|---|---|---|---|
| | $\Delta_\theta$ | R1 | R5 | $\Delta_\theta$ | R1 | R5 | $\Delta_\theta$ | R1 | R5 |
| $(X, Y)$ | 73.86° | 85.46 | 99.12 | 73.39° | 63.04 | 87.07 | 72.21° | 59.60 | 86.35 |
| $(X^*, Y^*)$ | 29.16° | 52.90 | 97.89 | 39.71° | 41.98 | 74.50 | 54.08° | 51.50 | 80.70 |
| $(X^{**}, Y^{**})$ | 45.81° | 67.69 | 97.94 | 38.79° | 54.89 | 81.90 | 54.03° | 55.62 | 84.53 |

*Table 8.* Performance of zero-shot image classification of ViT-B/32 with post-hoc transformations on various datasets.

| Model | $\Delta_\theta$ | I → T | | | T → I | | |
|---|---|---|---|---|---|---|---|
| | | R@1 | R@5 | R@10 | R@1 | R@5 | R@10 |
| $(X, Y)$ | 72.98° | 48.40 | 73.72 | 81.54 | 29.89 | 54.04 | 65.07 |
| $(X^*, Y^*)$ | 54.39° | 39.26 | 65.46 | 74.86 | 27.26 | 51.80 | 63.18 |
| $(X^{**}, Y^{**})$ | 54.22° | 36.70 | 65.92 | 75.48 | 29.54 | 53.97 | 65.09 |

*Table 9.* Performance of zero-shot cross-modal retrieval of ViT-B/32 with post-hoc transformations on MSCOCO.

# I. Additional Details and Results of Post-Hoc Transformations

### I.1. Zero-Shot Image Classification

**Datasets.** We first evaluate the post-hoc transformations on the zero-shot image classification task using three widely adopted datasets: two small-scale image dataset **CIFAR-10/100** (Krizhevsky et al., 2009) and one large scale image dataset **ImageNet-1k** (Deng et al., 2009). For CIFAR-10/100, we adopt the small set of prompts provided by OpenAI for CLIP (Radford et al., 2021) (https://github.com/openai/CLIP.com). For ImageNet-1k, the representation extraction settings follow Sec. D.1.

### I.2. Zero-Shot Cross-Modal Retrieval

**Datasets.** In addition to zero-shot image classification, we evaluate the post-hoc transformations on zero-shot image-to-text and text-to-image retrieval using the MSCOCO (Lin et al., 2014) dataset. The representation extraction settings follow Sec. D.1.

### I.3. Additional Results

The results of zero-shot image classification are reported in Tab. 8, and the full results of zero-shot cross-modal retrieval are reported in Tab. 9. These results show that reducing the modality gap while keeping shared space alignment unchanged does not improve downstream performance. These results also show that although transforming $(X, Y)$ into $(X^{**}, Y^{**})$ post-hoc reduces $\Delta_\theta$, it does not improve downstream performance. It is because the theoretical optimum, i.e., the COR is fully achieved, is not satisfied by real pretrained VLMs.

### I.4. Ablation Study of $c$ and $\epsilon$

In this subsection, we provide ablation studies on the choice of $c_x$, $c_y$ and $\epsilon$. The performance are evaluated with the post-hoc transformation $(X^{**}, Y^{**})$ on MSCOCO using the ViT-L/14 backbone that is pretrained on OpenAI's WebImageText dataset (Radford et al., 2021).

First, we fix $c_x = c_y$ and set them as [0.1, 0.05, 0.01, 0.001, 0.0001]. We also set $\epsilon$ as [0.5, 0.4, 0.3, 0.2, 0.1, 0.05, 0.01, 0.001, 0.0001]. Then, we compute $R1$ recall of zero-shot cross modal retrieval on MSCOCO. The results in Tab. 10 indicate that a good choice of $\epsilon$ is 0.1.

Second, we fix $\epsilon = 0.1$ and set $c_x$ and $c_y$ as [0.1, 0.05, 0.01, 0.001, 0.0001] respectively. We also evaluate different sets of

parameters with $R1$ accuracies of zero-shot cross modal retrieval on MSCOCO. The results in Tab. 11 indicate that a good choice of $c_x$ and $c_y$ is $c_x = c_y = 0.1$.

## J. Additional Details of Experiment of SSA

### J.1. Pretraining Details

All models, including the proposed method and the benchmark baselines, are trained from scratch on the Conceptual Captions 12M (CC12M) dataset (Changpinyo et al., 2021). Unless otherwise specified, all training settings follow (Eslami & de Melo, 2025). Specifically, all models employ the same vision transformer encoder with 12 layers and 12 attention heads. Input preprocessing and data augmentation follow the original CLIP implementation (Radford et al., 2021) and are kept identical across all methods. The image patch size is $16 \times 16$. For the text encoder, the maximum sequence length is 77 tokens and the vocabulary size is 49,408. The output embedding dimension of both the vision and text encoders is 768. This setup ensures that performance differences arise solely from the learning objectives rather than architectural or training variations. All models are implemented using OpenCLIP. We use the AdamW optimizer with an initial learning rate of $10^{-3}$, a cosine learning-rate schedule, 10,000 warmup steps, and a weight decay of 0.1. The temperature parameter is initialized to 0.07 for all models. Each model is trained for 30 epochs with a batch size of 512.

### J.2. Evaluation Details

Detailed evaluation protocols for downstream tasks are provided in Sec. I.

*Table 10.* Ablation results on MSCOCO for ViT-L/14. Columns correspond to different $\epsilon$, and rows correspond to metrics grouped by $c_x = c_y$.

| $c_x = c_y$ | metric | $\epsilon$ | | | | | | | | |
|---|---|---|---|---|---|---|---|---|---|---|
| | | 0.5 | 0.4 | 0.3 | 0.2 | 0.1 | 0.05 | 0.01 | 0.001 | 0.0001 |
| w/o proj | $d_X$ | 768 | 768 | 768 | 768 | 768 | 768 | 768 | 768 | 768 |
| | $d_Y$ | 768 | 768 | 768 | 768 | 768 | 768 | 768 | 768 | 768 |
| | $d_S$ | 768 | 768 | 768 | 768 | 768 | 768 | 768 | 768 | 768 |
| | $\Delta_\theta$ | 79.40 | 79.40 | 79.40 | 79.40 | 79.40 | 79.40 | 79.40 | 79.40 | 79.40 |
| | T2I@R1 | 0.36 | 0.36 | 0.36 | 0.36 | 0.36 | 0.36 | 0.36 | 0.36 | 0.36 |
| | I2T@R1 | 0.57 | 0.57 | 0.57 | 0.57 | 0.57 | 0.57 | 0.57 | 0.57 | 0.57 |
| 0.1 | $d_X$ | 171 | 171 | 171 | 171 | 171 | 171 | 171 | 171 | 171 |
| | $d_Y$ | 142 | 142 | 142 | 142 | 142 | 142 | 142 | 142 | 142 |
| | $d_S$ | 99 | 89 | 75 | 59 | 33 | 11 | 1 | 1 | 1 |
| | $\Delta_\theta$ | 60.42 | 59.05 | 59.76 | 63.13 | 50.67 | 43.92 | - | - | - |
| | T2I@R1 | 0.28 | 0.27 | 0.27 | 0.26 | 0.21 | 0.06 | 0.00 | 0.00 | 0.00 |
| | I2T@R1 | 0.27 | 0.30 | 0.34 | 0.33 | 0.27 | 0.07 | 0.00 | 0.00 | 0.00 |
| 0.05 | $d_X$ | 288 | 288 | 288 | 288 | 288 | 288 | 288 | 288 | 288 |
| | $d_Y$ | 244 | 244 | 244 | 244 | 244 | 244 | 244 | 244 | 244 |
| | $d_S$ | 187 | 174 | 158 | 137 | 103 | 73 | 23 | 1 | 1 |
| | $\Delta_\theta$ | 63.11 | 61.17 | 56.29 | 55.77 | 53.38 | 47.40 | 55.42 | 180.00 | 180.00 |
| | T2I@R1 | 0.33 | 0.33 | 0.33 | 0.33 | 0.30 | 0.28 | 0.10 | 0.00 | 0.00 |
| | I2T@R1 | 0.34 | 0.39 | 0.39 | 0.40 | 0.38 | 0.37 | 0.14 | 0.00 | 0.00 |
| 0.01 | $d_X$ | 489 | 489 | 489 | 489 | 489 | 489 | 489 | 489 | 489 |
| | $d_Y$ | 459 | 459 | 459 | 459 | 459 | 459 | 459 | 459 | 459 |
| | $d_S$ | 404 | 392 | 380 | 367 | 348 | 338 | 321 | 285 | 194 |
| | $\Delta_\theta$ | 64.71 | 64.57 | 61.46 | 60.42 | 61.84 | 63.99 | 64.61 | 65.35 | 65.51 |
| | T2I@R1 | 0.34 | 0.34 | 0.34 | 0.35 | 0.35 | 0.35 | 0.35 | 0.35 | 0.30 |
| | I2T@R1 | 0.42 | 0.44 | 0.46 | 0.47 | 0.47 | 0.47 | 0.47 | 0.46 | 0.41 |
| 0.001 | $d_X$ | 597 | 597 | 597 | 597 | 597 | 597 | 597 | 597 | 597 |
| | $d_Y$ | 587 | 587 | 587 | 587 | 587 | 587 | 587 | 587 | 587 |
| | $d_S$ | 579 | 577 | 575 | 572 | 569 | 563 | 548 | 496 | 417 |
| | $\Delta_\theta$ | 63.68 | 64.37 | 65.05 | 66.68 | 66.93 | 66.33 | 65.52 | 65.68 | 66.18 |
| | T2I@R1 | 0.34 | 0.34 | 0.34 | 0.35 | 0.35 | 0.35 | 0.35 | 0.35 | 0.34 |
| | I2T@R1 | 0.41 | 0.44 | 0.45 | 0.46 | 0.46 | 0.47 | 0.48 | 0.47 | 0.46 |
| 0.0001 | $d_X$ | 626 | 626 | 626 | 626 | 626 | 626 | 626 | 626 | 626 |
| | $d_Y$ | 621 | 621 | 621 | 621 | 621 | 621 | 621 | 621 | 621 |
| | $d_S$ | 620 | 620 | 620 | 620 | 620 | 619 | 607 | 549 | 479 |
| | $\Delta_\theta$ | 60.35 | 60.35 | 60.35 | 60.35 | 60.35 | 59.96 | 59.41 | 60.28 | 61.83 |
| | T2I@R1 | 0.34 | 0.34 | 0.34 | 0.34 | 0.34 | 0.34 | 0.34 | 0.34 | 0.33 |
| | I2T@R1 | 0.37 | 0.37 | 0.37 | 0.37 | 0.37 | 0.39 | 0.39 | 0.40 | 0.40 |

*Table 11.* Ablation results on MSCOCO for ViT-L/14. Columns correspond to different $c_x$ and rows correspond to different $c_y$. $\epsilon = 0.1$ in all cases.

| $c_y$ | metric | $c_x$ | | | | |
|---|---|---|---|---|---|---|
| | | 0.1 | 0.05 | 0.01 | 0.001 | 0.0001 |
| w/o proj | $d_X$ | 768 | 768 | 768 | 768 | 768 |
| | $d_Y$ | 768 | 768 | 768 | 768 | 768 |
| | $d_S$ | 768 | 768 | 768 | 768 | 768 |
| | $\Delta_\theta$ | 79.40 | 79.40 | 79.40 | 79.40 | 79.40 |
| | T2I@R1 | 0.36 | 0.36 | 0.36 | 0.36 | 0.36 |
| | I2T@R1 | 0.57 | 0.57 | 0.57 | 0.57 | 0.57 |
| 0.1 | $d_X$ | 171 | 171 | 171 | 171 | 171 |
| | $d_Y$ | 142 | 244 | 459 | 587 | 621 |
| | $d_S$ | 33 | 56 | 111 | 161 | 170 |
| | $\Delta_\theta$ | 50.67 | 55.75 | 58.43 | 59.80 | 63.40 |
| | T2I@R1 | 0.21 | 0.25 | 0.29 | 0.29 | 0.29 |
| | I2T@R1 | 0.27 | 0.34 | 0.37 | 0.38 | 0.36 |
| 0.05 | $d_X$ | 288 | 288 | 288 | 288 | 288 |
| | $d_Y$ | 142 | 244 | 459 | 587 | 621 |
| | $d_S$ | 64 | 103 | 195 | 271 | 286 |
| | $\Delta_\theta$ | 43.84 | 53.38 | 55.30 | 63.96 | 62.81 |
| | T2I@R1 | 0.29 | 0.30 | 0.33 | 0.34 | 0.33 |
| | I2T@R1 | 0.36 | 0.38 | 0.43 | 0.45 | 0.42 |
| 0.01 | $d_X$ | 489 | 489 | 489 | 489 | 489 |
| | $d_Y$ | 142 | 244 | 459 | 587 | 621 |
| | $d_S$ | 123 | 193 | 348 | 465 | 487 |
| | $\Delta_\theta$ | 52.24 | 55.04 | 61.84 | 65.99 | 64.72 |
| | T2I@R1 | 0.32 | 0.34 | 0.35 | 0.35 | 0.35 |
| | I2T@R1 | 0.41 | 0.43 | 0.47 | 0.47 | 0.42 |
| 0.001 | $d_X$ | 597 | 597 | 597 | 597 | 597 |
| | $d_Y$ | 142 | 244 | 459 | 587 | 621 |
| | $d_S$ | 139 | 238 | 444 | 569 | 595 |
| | $\Delta_\theta$ | 56.80 | 61.27 | 65.57 | 66.93 | 64.37 |
| | T2I@R1 | 0.32 | 0.35 | 0.35 | 0.35 | 0.35 |
| | I2T@R1 | 0.42 | 0.46 | 0.47 | 0.46 | 0.41 |
| 0.0001 | $d_X$ | 626 | 626 | 626 | 626 | 626 |
| | $d_Y$ | 142 | 244 | 459 | 587 | 621 |
| | $d_S$ | 141 | 243 | 458 | 586 | 620 |
| | $\Delta_\theta$ | 51.57 | 55.00 | 57.81 | 59.50 | 60.35 |
| | T2I@R1 | 0.31 | 0.34 | 0.34 | 0.34 | 0.34 |
| | I2T@R1 | 0.36 | 0.40 | 0.42 | 0.40 | 0.37 |

# K. Appendix E: Proofs

## K.1. Details of Theorem 1

In this section, we provide proofs of Theorem 1 that is proposed in Sec. 3.2. We also provide details of the auxiliary theorems (Theorem S1 and Theorem S2) and technical lemmas (Lemma 1, Lemma 2, Lemma 3, Lemma 4) that support the proof of Theorem 1. For convenience in reading, let us recall some related notions and definitions.

- $h, N \in \mathbb{N}$.

- $\mathbb{S}^{h-1} = \left\{ z \in \mathbb{R}^h : \|z\| = 1 \right\}$.

- $\sigma_{h-1}$: the uniform probability measure of $\mathbb{S}^{h-1}$.

**Definition** (Multimodal Contrastive Loss (MCL Loss)). Let $(X, Y)$ be an $N$-pair configuration, where $X = (x_1, \ldots, x_N) \in (\mathbb{S}^{h-1})^N$ and $Y = (y_1, \ldots, y_N) \in (\mathbb{S}^{h-1})^N$. $\forall \tau > 0$, the multimodal contrastive loss $\mathcal{L}_{\mathrm{MCL}}(\cdot, \cdot) : (\mathbb{S}^{h-1})^N \times (\mathbb{S}^{h-1})^N \to \mathbb{R}$ is defined as:

$$\mathcal{L}_{\mathrm{MCL}} = \frac{1}{N} \sum_{i=1}^{N} \mathcal{L}_{\mathrm{MCL}}^i, \quad \text{where} \ \ \mathcal{L}_{\mathrm{MCL}}^i = \mathcal{L}_{\mathcal{X} \to \mathcal{Y}}(x_i; Y) + \mathcal{L}_{\mathcal{Y} \to \mathcal{X}}(y_i; X).$$

Here, $\mathcal{L}_{\mathcal{X} \to \mathcal{Y}}$ is the $\mathcal{X}$-to-$\mathcal{Y}$ alignment and $\mathcal{L}_{\mathcal{Y} \to \mathcal{X}}$ is the $\mathcal{Y}$-to-$\mathcal{X}$ alignment, which are defined respectively as:

$$\mathcal{L}_{\mathcal{X} \to \mathcal{Y}}(x_i; Y) = -\log \frac{\exp\left(x_i \cdot y_i / \tau\right)}{\sum_{j=1}^{N} \exp\left(x_i \cdot y_j / \tau\right)}, \quad \mathcal{L}_{\mathcal{Y} \to \mathcal{X}}(y_i; X) = -\log \frac{\exp\left(x_i \cdot y_i / \tau\right)}{\sum_{j=1}^{N} \exp\left(x_j \cdot y_i / \tau\right)}.$$

### K.1.1. PROOF OF THEOREM 1

In this subsection, we provide the proof of Theorem 1. For convenience in reading, we first restate Theorem 1 here.

**Theorem 1.** [Restate] Let $(X, Y)$ be an $N$-pair configuration, where $X = (x_1, \ldots, x_N) \in (\mathbb{S}^{h-1})^N$ are $iid$ samples from $\mu_x$ and $Y = (y_1, \ldots, y_N) \in (\mathbb{S}^{h-1})^N$ are $iid$ samples from $\mu_y$. Let $\nu = h/2 - 1$, it holds that:

$$\lim_{N \to \infty} \mathcal{L}_{\mathrm{MCL}} - 2\log(N) = \mathbb{E}_{x_i \sim \mu_x}\left[ -\frac{x_i \cdot y_i}{\tau} \right] + \mathbb{E}_{x_i \sim \mu_x}\left[ \log \mathbb{E}_{y_i \sim \mu_y}\left[ \exp\left( \frac{x_i \cdot y_i}{\tau} \right) \right] \right]$$
$$+ \mathbb{E}_{y_i \sim \mu_y}\left[ -\frac{x_i \cdot y_i}{\tau} \right] + \mathbb{E}_{y_i \sim \mu_y}\left[ \log \mathbb{E}_{x_j \sim \mu_x}\left[ \exp\left( \frac{x_i \cdot y_i}{\tau} \right) \right] \right]$$
$$\geq -\frac{2}{\tau} + 2\log\left( \Gamma\left(\nu + 1\right) (2\tau)^\nu I_\nu\left( \frac{1}{\tau} \right) \right)$$

where equality is attained if and only if there exists a configuration of $(X, Y)$ such that:

(A1) $\forall i \in [N]$, $x_i = y_i$.

(A2) $\mu_x = \sigma_{h-1}$ and $\mu_y = \sigma_{h-1}$.

*Proof.* We first decompose $\lim_{N \to \infty} \mathcal{L}_{\mathrm{MCL}}^c - 2\log(N)$ into two parts:

$$\lim_{N \to \infty} (\mathcal{L}_{\mathrm{MCL}} - 2\log(N)) = \lim_{N \to \infty} \left( \frac{1}{N} \sum_{i=1}^{N} \mathcal{L}_{\mathcal{X} \to \mathcal{Y}}(x_i; Y) - \log(N) \right)$$
$$+ \lim_{N \to \infty} \left( \frac{1}{N} \sum_{i=1}^{N} \mathcal{L}_{\mathcal{Y} \to \mathcal{X}}(y_i; X) - \log(N) \right). \tag{26}$$

According to Theorem S2, the convergent function and its lower bound of $\mathcal{L}_{\mathcal{X} \to \mathcal{Y}}$ are:

$$\lim_{N \to \infty} \frac{1}{N} \sum_{i=1}^{N} \mathcal{L}_{\mathcal{X} \to \mathcal{Y}}(x_i; Y) - \log(N)$$

$$= \mathbb{E}_{x_i \sim \mu_x} \left[ -\frac{x_i \cdot y_i}{\tau} \right] + \mathbb{E}_{x_i \sim \mu_x} \left[ \log \mathbb{E}_{y_i \sim \mu_y} \left[ \exp \left( \frac{x_i \cdot y_j}{\tau} \right) \right] \right] \tag{27}$$

$$\geq -\frac{1}{\tau} + \log \left[ \Gamma \left( \frac{h}{2} \right) (2\tau)^{\frac{h}{2} - 1} I_{\frac{h}{2} - 1} \left( \frac{1}{\tau} \right) \right]$$

where equality is attained if and only if there exists a configuration of $(X, Y)$ such that:

(i) $\forall i \in [N], x_i = y_i$.

(ii) $\mu_x = \sigma_{h-1}$ and $\mu_y = \sigma_{h-1}$.

This Theorem also holds for $\mathcal{L}_{\mathcal{Y} \to \mathcal{X}}$:

$$\lim_{N \to \infty} \frac{1}{N} \sum_{i=1}^{N} \mathcal{L}_{\mathcal{Y} \to \mathcal{X}}(y_i; X) - \log(N)$$

$$= \mathbb{E}_{y_i \sim \mu_x} \left[ -\frac{x_i \cdot y_i}{\tau} \right] + \mathbb{E}_{y_i \sim \mu_y} \left[ \log \mathbb{E}_{x_i \sim \mu_x} \left[ \exp \left( \frac{x_i \cdot y_j}{\tau} \right) \right] \right] \tag{28}$$

$$\geq -\frac{1}{\tau} + \log \left[ \Gamma \left( \frac{h}{2} \right) (2\tau)^{\frac{h}{2} - 1} I_{\frac{h}{2} - 1} \left( \frac{1}{\tau} \right) \right]$$

where equality is attained if and only if there exists a configuration of $(X, Y)$ such that:

(iii) $\forall i \in [N], x_i = y_i$.

(iv) $\mu_x = \sigma_{h-1}$ and $\mu_y = \sigma_{h-1}$.

Combining Eq. (26), Eq. (27) and Eq. (28), we conclude that:

$$\lim_{N \to \infty} \mathcal{L}_{\mathrm{MCL}} - 2\log(N) = \mathbb{E}_{x_i \sim \mu_x} \left[ -\frac{x_i \cdot y_i}{\tau} \right] + \mathbb{E}_{x_i \sim \mu_x} \left[ \log \mathbb{E}_{y_i \sim \mu_y} \left[ \exp \left( \frac{x_i \cdot y_i}{\tau} \right) \right] \right]$$

$$+ \mathbb{E}_{y_i \sim \mu_y} \left[ -\frac{x_i \cdot y_i}{\tau} \right] + \mathbb{E}_{y_i \sim \mu_y} \left[ \log \mathbb{E}_{x_j \sim \mu_x} \left[ \exp \left( \frac{x_i \cdot y_i}{\tau} \right) \right] \right] \tag{29}$$

$$\geq -\frac{2}{\tau} + 2\log \left[ \Gamma \left( \frac{h}{2} \right) (2\tau)^{\frac{h}{2} - 1} I_{\frac{h}{2} - 1} \left( \frac{1}{\tau} \right) \right]$$

where equality is attained if and only if the following conditions hold:

(A1) $\forall i \in [N], x_i = y_i$.

(A2) $\mu_x = \sigma_{h-1}$ and $\mu_y = \sigma_{h-1}$.

$\square$

### K.1.2. AUXILIARY THEOREMS PART 1

In this subsection, we provide details and proofs of the auxiliary theorems (Theorem S1 and Theorem S2) that support the proof of Theorem 1.

**Theorem S1.** Let $(X, Y)$ be an $N$-pair configuration, where $X = (x_1, \ldots, x_N) \in (\mathbb{S}^{h-1})^N$ are $iid$ samples from $\mu_x$ and $Y = (y_1, \ldots, y_N) \in (\mathbb{S}^{h-1})^N$ are $iid$ samples from $\mu_y$. It holds that:

$$\lim_{N \to \infty} \frac{1}{N} \sum_{i=1}^{N} \mathcal{L}_{\mathcal{X} \to \mathcal{Y}}(x_i; Y) - \log(N) = \lim_{N \to \infty} \frac{1}{N} \sum_{i=1}^{N} -\log \frac{\exp(x_i \cdot y_i / \tau)}{\sum_{j=1}^{N} \exp(x_i \cdot y_j / \tau)} - \log(N)$$

$$= \mathbb{E}_{x_i \cdot y_i} \left[ -\frac{x_i \cdot y_i}{\tau} \right] + \mathbb{E}_{x_i \sim \mu_x} \left[ \log \mathbb{E}_{y_i \sim \mu_y} \left[ \exp \left( \frac{x_i \cdot y_j}{\tau} \right) \right] \right] \tag{30}$$

*Proof.* $\forall x_i \in X$, the $\mathcal{X}$-to-$\mathcal{Y}$ alignment of $x_i$ can be rewritten as:

$$\mathcal{L}_{\mathcal{X} \to \mathcal{Y}}(x_i; Y) = -\log \frac{\exp(x_i \cdot y_i / \tau)}{\sum_j \exp(x_i \cdot y_j / \tau)}$$

$$= -\frac{x_i \cdot y_i}{\tau} + \log \left( N \frac{1}{N} \sum_{j=1}^{N} \exp \left( \frac{x_i \cdot y_j}{\tau} \right) \right) \tag{31}$$

$$= -\frac{x_i \cdot y_i}{\tau} + \log \left( \frac{1}{N} \sum_{j=1}^{N} \exp \left( \frac{x_i \cdot y_j}{\tau} \right) \right) + \log(N).$$

Denote $h_N(x)$ and $h(x)$ as:

$$h_N(x) = \log \left( \frac{1}{N} \sum_{j=1}^{N} \exp \left( \frac{x \cdot y_j}{\tau} \right) \right),$$

$$\text{and } h(x) = \log \left( \mathbb{E}_{y \sim \mu_y} \left[ \exp \left( \frac{x \cdot y}{\tau} \right) \right] \right). \tag{32}$$

Lemma 2 reveals that $h_N(x)$ uniformly converges to $h(x)$ almost surely. Thus, we have:

$$\sup_{x \in \mathbb{S}^{h-1}} |h_N(x) - h(x)| \xrightarrow[N \to \infty]{\text{a.s.}} 0. \tag{33}$$

According to the Strong Law of Large Numbers (SLLN), we have:

$$\frac{1}{N} \sum_{i=1}^{N} h(x_i) \xrightarrow[N \to \infty]{\text{a.s.}} \mathbb{E}_{x \sim \mu_x}[h(x)]. \tag{34}$$

Combining Eq. (33) and Eq. (34), we get:

$$\frac{1}{N} \sum_{i=1}^{N} h_N(x_i) = \frac{1}{N} \sum_{i=1}^{N} h(x_i) + \frac{1}{N} \sum_{i=1}^{N} (h_N(x_i) - h(x_i))$$

$$\xrightarrow[N \to \infty]{\text{a.s.}} \mathbb{E}_{x \sim \mu_x}[h(x)]. \tag{35}$$

Similarly, by the Strong Law of Large Numbers (SLLN), we have:

$$\frac{1}{N} \sum_{i=1}^{N} -\frac{x_i \cdot y_i}{\tau} \xrightarrow[N \to \infty]{\text{a.s.}} \mathbb{E}_{x_i \sim \mu_x} \left[ -\frac{x_i \cdot y_i}{\tau} \right]. \tag{36}$$

Putting Eq. (31), Eq. (35) and Eq. (36) together, the convergent function of $\frac{1}{N} \sum_{i=1}^{N} \mathcal{L}_{\mathcal{X} \to \mathcal{Y}}(x_i; Y)$ can be derived as:

$$\lim_{N \to \infty} \frac{1}{N} \sum_{i=1}^{N} \mathcal{L}_{\mathcal{X} \to \mathcal{Y}}(x_i; Y) - \log(N) = \lim_{N \to \infty} \frac{1}{N} \sum_{i=1}^{N} \left( -\frac{x_i \cdot y_i}{\tau} + h_N(x_i) \right)$$

$$= \mathbb{E}_{x_i \cdot y_i} \left[ -\frac{x_i \cdot y_i}{\tau} \right] + \mathbb{E}_{x_i \sim \mu_x} \left[ h(x_i) \right] \tag{37}$$

$$= \mathbb{E}_{x_i \cdot y_i} \left[ -\frac{x_i \cdot y_i}{\tau} \right] + \mathbb{E}_{x_i \sim \mu_x} \left[ \log \mathbb{E}_{y_j \sim \mu_y} \left[ \exp \left( \frac{x_i \cdot y_j}{\tau} \right) \right] \right].$$

$\square$

**Theorem S2.** Let $(X, Y)$ be an $N$-pair configuration, where $X = (x_1, \ldots, x_N) \in (\mathbb{S}^{h-1})^N$ are $iid$ samples from $\mu_x$ and $Y = (y_1, \ldots, y_N) \in (\mathbb{S}^{h-1})^N$ are $iid$ samples from $\mu_y$. Let $\nu = h/2 - 1$, it holds that:

$$\lim_{N \to \infty} \frac{1}{N} \sum_{i=1}^{N} \mathcal{L}_{\mathcal{X} \to \mathcal{Y}}(x_i; Y) - \log(N)$$

$$= \mathbb{E}_{x_i \sim \mu_x} \left[ -\frac{x_i \cdot y_i}{\tau} \right] + \mathbb{E}_{x_i \sim \mu_x} \left[ \log \mathbb{E}_{y_i \sim \mu_y} \left[ \exp \left( \frac{x_i \cdot y_j}{\tau} \right) \right] \right] \tag{38}$$

$$\geq \log \left( \Gamma \left( \nu + 1 \right) \left( 2\tau \right)^{\nu} I_{\nu} \left( \frac{1}{\tau} \right) \right)$$

where equality is attained if and only if the following conditions hold:

(B1) $\forall i \in [N]$, $x_i = y_i$.

(B2) $\mu_x = \sigma_{h-1}$ and $\mu_y = \sigma_{h-1}$.

*Proof.* **Step 1**: We start the proof by find the convergent function of $\frac{1}{N} \sum_{i=1}^{N} \mathcal{L}_{\mathcal{X} \to \mathcal{Y}}(x_i; Y)$ as $N \to \infty$. $\forall x_i \in X$, as prove in Theorem S1:

$$\lim_{N \to \infty} \frac{1}{N} \sum_{i=1}^{N} \mathcal{L}_{\mathcal{X} \to \mathcal{Y}}(x_i; Y) - \log(N) = \lim_{N \to \infty} \frac{1}{N} \sum_{i=1}^{N} -\log \frac{\exp \left( x_i \cdot y_i / \tau \right)}{\sum_{j=1}^{N} \exp \left( x_i \cdot y_j / \tau \right)} - \log(N)$$

$$= \mathbb{E}_{x_i \cdot y_i} \left[ -\frac{x_i \cdot y_i}{\tau} \right] + \mathbb{E}_{x_i \sim \mu_x} \left[ \log \mathbb{E}_{y_i \sim \mu_y} \left[ \exp \left( \frac{x_i \cdot y_j}{\tau} \right) \right] \right]. \tag{39}$$

**Step 2**: Next, we find the minimal value and the optimal condition of convergent function.

According to the Cauchy-Schwarz inequality, the first term in Eq. (39) can be bounded below:

$$\mathbb{E}_{x_i \cdot y_i} \left[ -\frac{x_i \cdot y_i}{\tau} \right] \geq \mathbb{E}_{x_i \cdot y_i} \left[ -\frac{\|x_i\| \|y_i\|}{\tau} \right] = -\frac{1}{\tau}. \tag{40}$$

where equality is attained if and only if there exists a configuration of $(X, Y)$ such that :

(B1) $\forall i \in [N]$, $x_i = y_i$.

Note that condition (B1) implies $\mu_x = \mu_y$. Applying this condition to the second term in Eq. (39), we can transform it as:

$$\mathbb{E}_{x \sim \mu_x} \left[ \log \left( \mathbb{E}_{y \sim \mu_y} \left[ \exp \left( \frac{x \cdot y}{\tau} \right) \right] \right) \right] = \mathbb{E}_{x \sim \mu} \left[ \log \left( \mathbb{E}_{y \sim \mu} \left[ \exp \left( \frac{x \cdot y}{\tau} \right) \right] \right) \right]. \tag{41}$$

Let $\mathbb{M}(\mathbb{S}^{h-1})$ be the set of Borel probability measures in $\mathbb{S}^{h-1}$. The RHS of Eq. (41) is then a functional $\mathcal{F}[\cdot] : \mathbb{M}(\mathbb{S}^{h-1}) \to \mathbb{R}$:

$$\mathcal{F}[\mu] = \mathbb{E}_{x \sim \mu} \left[ \log \left( \mathbb{E}_{y \sim \mu} \left[ \exp \left( \frac{x \cdot y}{\tau} \right) \right] \right) \right]. \tag{42}$$

According to Lemma 3, $\mathcal{F}[\mu]$ is minimized when $\mu = \sigma_{h-1}$ where $\sigma_{h-1}$ is the uniform measure of $\mathbb{S}^{h-1}$:

$$\sigma_{h-1} = \underset{\mu \in \mathbb{M}(\mathbb{S}^{h-1})}{\arg\min} \mathcal{F}[\mu]. \tag{43}$$

Therefore, we have:

$$\mathcal{F}[\mu] \geq \mathcal{F}[\sigma_{h-1}]. \tag{44}$$

where equality is attained if and only if there exists a configuration of $(X, Y)$ such that :

(B2) $\mu_x = \mu_y = \sigma_{h-1}$

Let $\Gamma\left(\cdot\right)$ be the Gamma function, Lemma 4 derives that:

$$
\begin{aligned}
\mathcal{F}[\sigma_{h-1}] &= \mathbb{E}_{x \sim \sigma_{h-1}}\left[\mathbb{E}_{y \sim \sigma_{h-1}}\left[\exp\left(\frac{x \cdot y}{\tau}\right)\right]\right] \\
&= \log\left[\Gamma\left(\frac{h}{2}\right)(2\tau)^{\frac{h}{2}-1} I_{\frac{h}{2}-1}\left(\frac{1}{\tau}\right)\right]
\end{aligned} \tag{45}
$$

Combining Eq. (39), Eq. (40), Eq. (41), Eq. (45), we conclude that:

$$
\begin{aligned}
\lim_{N \to \infty} \frac{1}{N} \sum_{i=1}^{N} \mathcal{L}_{\mathcal{X} \to \mathcal{Y}}(x_i; Y) - \log(N) \\
= \mathbb{E}_{x_i \sim \mu_x}\left[-\frac{x_i \cdot y_i}{\tau}\right] + \mathbb{E}_{x_i \sim \mu_x}\left[\log \mathbb{E}_{y_i \sim \mu_y}\left[\exp\left(\frac{x_i \cdot y_j}{\tau}\right)\right]\right] \\
\geq -\frac{1}{\tau} + \log\left[\Gamma\left(\frac{h}{2}\right)(2\tau)^{\frac{h}{2}-1} I_{\frac{h}{2}-1}\left(\frac{1}{\tau}\right)\right]
\end{aligned} \tag{46}
$$

where equality is attained if and only if the following conditions hold:

(B1) $\forall i \in [N]$, $x_i = y_i$.

(B2) $\mu_x = \sigma_{d-1}$ and $\mu_y = \sigma_{d-1}$.

$\square$

### K.1.3. TECHNICAL LEMMAS PART 1

In this section, we provide details and proofs of the technical lemmas (technical lemmas (Lemma 1, Lemma 2, Lemma 3, Lemma 4) that support the proof of Theorem 1, Theorem S1 and Theorem S2.

**Lemma 1.** *Let $x \in \mathbb{S}^{h-1}$ and $Y$ be an $N$-point configuration, where $Y = (y_1, \ldots, y_N) \in (\mathbb{S}^{h-1})^N$ are iid samples from $\mu_y$. $\forall \tau > 0$, define a sequence of functions $\{g_N\} : \mathbb{S}^{h-1} \to \mathbb{R}$ as:*

$$g_N(x) = \frac{1}{N} \sum_{j=1}^{N} \exp\left(\frac{x \cdot y_j}{\tau}\right). \tag{47}$$

*Define a function $g : \mathbb{S}^{h-1} \to \mathbb{R}$ as:*

$$g(x) = \mathbb{E}_{y \sim \mu_y}\left[\exp\left(\frac{x \cdot y}{\tau}\right)\right]. \tag{48}$$

*It holds that $\{g_N\}$ converges uniformly to $g$:*

$$g_N(x) \xrightarrow[N \to \infty]{\text{unif.}} g(x). \tag{49}$$

*Proof.* **Step 1 Boundedness and Lipschitz Property:**

Consider a function class $\mathcal{F} = \left\{ f_x(y) = \exp\left(\frac{x \cdot y}{\tau}\right) : x, y \in \mathbb{S}^{h-1} \right\}$. Since $\|x\| = \|y\| = 1, x \cdot y \in [-1, 1]$, hence $\forall f_x \in F$, we have:

$$|f_x(y)| \le e^{1/\tau}. \tag{50}$$

Therefore, $f_x(y)$ is uniformly bounded in $y$, so is its derivative:

$$\|\nabla_x f_x(y)\| = \left\| \frac{y}{\tau} f_x(y) \right\| \le \frac{1}{\tau} e^{1/\tau}. \tag{51}$$

Then $\forall x_k \in \mathbb{S}^{h-1}$,

$$|f_x(y) - f_{x_k}(y)| \le \frac{1}{\tau} e^{1/\tau} =: L. \tag{52}$$

Thus, $f_x(y)$ is Lipschitz in $x$ with constant $L = \frac{e^{1/\tau}}{\tau}$, uniformly in $y$.

**Step 2 $\eta$-Net:**

According to Lemma 5.2 in (Vershynin, 2010), $\forall \varepsilon > 0$ and $\eta = \frac{\varepsilon}{4L}$, there exists a finite $\eta$-net, $\mathcal{N}_\eta = \{x_1, x_2, \ldots, x_K\} \subset \mathbb{S}^{h-1}$, with cardinality:

$$K = |\mathcal{N}_\eta| \le \left(1 + \frac{2}{\eta}\right)^h < \left(\frac{3}{\eta}\right)^h. \tag{53}$$

$\forall x \in \mathbb{S}^{h-1}, \exists x_k \in \mathcal{N}_\eta$ such that $\|x - x_k\| < \eta$. Because $f_x(y)$ is $L$-Lipschitz in $x$, we have:

$$|f_x(y) - f_{x_k}(y)| \le L \|x - x_k\| = L\eta. \tag{54}$$

And we also have:

$$\begin{aligned} |g_N(x) - g_N(x_k)| &\le L\eta, \\ |g(x) - g(x_k)| &\le L\eta. \end{aligned} \tag{55}$$

**Step 3 Probability Bound:**

$\forall x_k \in \mathcal{N}_\eta$, the random variables $Z_j := f_{x_k}(y_j)$ are iid and lie in $\left[e^{-1/\tau}, e^{1/\tau}\right]$. According to the Hoeffding's inequality:

$$P\left(|g_N(x_k) - g(x_k)| > \frac{\varepsilon}{2}\right) \le 2\exp\left(-\frac{2N(\varepsilon/2)^2}{(2e^{1/\tau})^2}\right) = 2e^{-cN\varepsilon^2}, \tag{56}$$

where $c = \frac{1}{8e^{2/\tau}} > 0$. Taking a union bound over the $\eta$-net:

$$P\left(\max_{x_k \in \mathcal{N}_\eta} |g_N(x_k) - g(x_k)| > \frac{\varepsilon}{2}\right) \le 2Ke^{-cN\varepsilon^2}. \tag{57}$$

**Step 4 Uniform Convergence:**

Since $\forall x \in \mathbb{S}^{h-1}$, $|g_N(x) - g(x)|$ can be decomposed as:

$$
\begin{aligned}
|g_N(x) - g(x)| &\leq |g_N(x) - g_N(x_k)| + |g_N(x_k) - g(x_k)| + |g(x_k) - g(x)| \\
&\leq 2L\eta + \max_{x_k \in \mathcal{N}_\eta} |g_N(x_k) - g(x_k)| \\
&= \frac{\varepsilon}{2} + \max_{x_k \in \mathcal{N}_\eta} |g_N(x_k) - g(x_k)| .
\end{aligned}
\tag{58}
$$

Plugging Eq. (57) into Eq. (58), we have:

$$
\begin{aligned}
P\left( \sup_{x \in \mathbb{S}^{h-1}} |g_N(x) - g(x)| > \varepsilon \right) &\leq P\left( \max_{x_k \in \mathcal{N}_\eta} |g_N(x_k) - g(x_k)| > \frac{\varepsilon}{2} \right) \\
&\leq 2K e^{-cN\varepsilon^2},
\end{aligned}
\tag{59}
$$

and therefore:

$$
\sup_{x \in \mathbb{S}^{h-1}} |g_N(x) - g(x)| \xrightarrow[N \to \infty]{P} 0.
\tag{60}
$$

Eq. (59) justifies that:

$$
\sum_{N=1}^{\infty} P\left( \sup_{x \in \mathbb{S}^{h-1}} |g_N(x) - g(x)| > \varepsilon \right) \leq 2K \sum_{N=1}^{\infty} e^{-cN\varepsilon^2} < \infty.
\tag{61}
$$

According to the Borel–Cantelli lemma:

$$
P\left( \limsup_{N \to \infty} \sup_{x \in \mathbb{S}^{h-1}} |g_N(x) - g(x)| > \varepsilon \right) = 0.
\tag{62}
$$

Therefore:

$$
\sup_{x \in \mathbb{S}^{h-1}} |g_N(x) - g(x)| \xrightarrow[N \to \infty]{\text{a.s.}} 0.
\tag{63}
$$

We conclude now the empirical averages $g_N(\cdot)$ converge uniformly in $\mathbb{S}^{h-1}$ to $g(\cdot)$:

$$
g_N(x) \xrightarrow[N \to \infty]{\text{unif.}} g(x).
\tag{64}
$$

$\square$

**Lemma 2.** *Let $x \in \mathbb{S}^{h-1}$ and $Y$ be an $N$-point configuration, where $Y = (y_1, \ldots, y_N) \in (\mathbb{S}^{h-1})^N$ are iid samples from $\mu_y$. $\forall \tau > 0$, define a sequence of functions $\{h_N\} : \mathbb{S}^{h-1} \to \mathbb{R}$ as:*

$$
h_N(x) = \log\left( \frac{1}{N} \sum_{j=1}^{N} \exp\left( \frac{x \cdot y_j}{\tau} \right) \right).
\tag{65}
$$

*Define a function $h : \mathbb{S}^{h-1} \to \mathbb{R}$ as:*

$$
h(x) = \log\left( \mathbb{E}_{y \sim \mu_y} \left[ \exp\left( \frac{x \cdot y}{\tau} \right) \right] \right).
\tag{66}
$$

*It holds that $\{h_N\}$ converges uniformly to $h$:*

$$\lim_{N\to\infty} h_N(x) \xrightarrow[N\to\infty]{\text{unif.}} h(x). \tag{67}$$

*Proof.* According to Lemma 1:

$$\sum_{j=1}^{N} \exp\left(\frac{x \cdot y_j}{\tau}\right) = g_N(x) \xrightarrow[N\to\infty]{\text{unif.}} g(x) = \mathbb{E}_{y\sim\mu_y}\left[\exp\left(\frac{x \cdot y}{\tau}\right)\right], \tag{68}$$

and

$$\sup_{x\in\mathbb{S}^{h-1}} |g_N(x) - g(x)| \xrightarrow[N\to\infty]{\text{a.s.}} 0. \tag{69}$$

Because $\langle x, y\rangle \in [-1, 1]$ for unit vectors, $\exp\left(x \cdot y/\tau\right)$ satisfies:

$$e^{-1/\tau} \le \exp\left(\frac{x \cdot y}{\tau}\right) \le e^{1/\tau}. \tag{70}$$

Hence $\forall x$, $g_N(x), g(x) \in [a, b]$ with $a = e^{-1/\tau} > 0, b = e^{1/\tau} > 0$. In the compact interval $[a, b]$, by the mean value theorem, $\forall u < v \in [a, b], \exists u < \xi < v$ such that:

$$|\log u - \log v| = \frac{|u - v|}{\xi} \le \frac{1}{a}|u - v| = e^{1/\tau}|u - v|. \tag{71}$$

Thus, the function $\log(\cdot)$ is Lipschitz . Therefore:

$$\sup_{x\in\mathbb{S}^{h-1}} |h_N(x) - h(x)| = \sup_{x\in\mathbb{S}^{h-1}} |\log g_N(x) - \log g(x)| \le \frac{1}{a} \sup_{x\in\mathbb{S}^{h-1}} |g_N(x) - g(x)| \xrightarrow[N\to\infty]{\text{a.s.}} 0 \tag{72}$$

We conclude now $h_N(\cdot)$ converge uniformly in $\mathbb{S}^{h-1}$ to $h(\cdot)$:

$$\lim_{N\to\infty} h_N(x) \xrightarrow{\text{unif.}} h(x) \tag{73}$$

$\square$

**Lemma 3.** *Let $M\left(\mathbb{S}^{h-1}\right)$ be the set of Borel probability measures in $\mathbb{S}^{h-1}$. Let $\sigma_{h-1} \in M\left(\mathbb{S}^{h-1}\right)$ be the uniform probability measure in $\mathbb{S}^{h-1}$. $\forall x, y \in \mathbb{S}^{h-1}$ and $\tau > 0$, a function $f : \mathbb{S}^{h-1} \times \mathbb{S}^{h-1} \to \mathbb{R}^+$ is defined as:*

$$f(x, y) = \exp\left(\frac{x \cdot y}{\tau}\right). \tag{74}$$

*$\forall \mu \in M\left(\mathbb{S}^{h-1}\right)$, a functional $\mathcal{F} : M\left(\mathbb{S}^{h-1}\right) \to \mathbb{R}^+$ is defined as:*

$$\mathcal{F}_f[\mu] = \int_{\mathbb{S}^{h-1}} \log\left(\int_{\mathbb{S}^{h-1}} f(x, y)\mathrm{d}\mu(y)\right) \mathrm{d}\mu(x). \tag{75}$$

*It holds that $\sigma_{h-1}$ is the unique minimizer of $\mathcal{F}$:*

$$\min_{\mu\in\mathcal{M}(\mathbb{S}^{h-1})} \mathcal{F}_f[\mu] = \min_{\mu\in\mathcal{M}(\mathbb{S}^{h-1})} \int_{\mathbb{S}^{h-1}} \log\left(\int_{\mathbb{S}^{h-1}} f(x, y)\mathrm{d}\mu(y)\right) \mathrm{d}\mu(x). \tag{76}$$

*Proof.* See the Proof of result 2 of Theorem 1 in (Wang & Isola, 2020) □

**Lemma 4.** *Let $M\left(\mathbb{S}^{h-1}\right)$ be the set of Borel probability measures in $\mathbb{S}^{h-1}$. Let $\sigma_{h-1} \in M\left(\mathbb{S}^{h-1}\right)$ be the uniform probability measure in $\mathbb{S}^{h-1}$. $\forall x, y \in \mathbb{S}^{h-1}$ and $\tau > 0$, a function $f : \mathbb{S}^{h-1} \times \mathbb{S}^{h-1} \to \mathbb{R}^{+}$ is defined as:*

$$f(x, y) = \exp\left(\frac{x \cdot y}{\tau}\right). \tag{77}$$

$\forall \mu \in M\left(\mathbb{S}^{h-1}\right)$, *a functional $\mathcal{F} : M\left(\mathbb{S}^{h-1}\right) \to \mathbb{R}^{+}$ is defined as:*

$$\mathcal{F}_f[\mu] = \int_{\mathbb{S}^{h-1}} \log\left(\int_{\mathbb{S}^{h-1}} f(x, y) \mathrm{d}\mu(y)\right) \mathrm{d}\mu(x). \tag{78}$$

*Let $\Gamma\left(\cdot\right)$ be the Gamma function and $\nu = h/2 - 1$, it holds that:*

$$\mathcal{F}_f[\sigma_{h-1}] = \log\left(\Gamma\left(\nu + 1\right)(2\tau)^{\nu} I_{\nu}\left(\frac{1}{\tau}\right)\right) \tag{79}$$

*Proof.* **Step 1: Rotational Invariance**

Since the measure $\sigma_{h-1}$ is invariant under orthogonal transformations. For any fixed $x \in \mathbb{S}^{h-1}$, the inner integral:

$$\int_{\mathbb{S}^{h-1}} \exp\left(\frac{x \cdot y}{\tau}\right) d\sigma_{h-1}(y), \tag{80}$$

depends only on the distribution of $(x \cdot y)$, and by rotational symmetry, this integral is independent of $x$. Thus, define:

$$Z_{\tau} := \int_{\mathbb{S}^{h-1}} \exp\left(\frac{x \cdot y}{\tau}\right) d\sigma_{h-1}(y), \tag{81}$$

and $Z_{\tau}$ is constant for all $x$. Since $\log Z_{\tau}$ is constant and $\sigma_{h-1}$ is a probability measure, we have:

$$\mathcal{F}_f[\sigma_{h-1}] = \int_{\mathbb{S}^{h-1}} \log Z_{\tau} d\sigma_{h-1}(x) = \log Z_{\tau}, \tag{82}$$

**Step 2: Compute $Z_{\tau}$**

Without the loss of generality, we assume the coordinate of $x$ as:

$$x = e_h = (0, \ldots, 0, 1). \tag{83}$$

Then $x \cdot y = y_h$, the last coordinate of $y$. So:

$$Z_{\tau} = \int_{\mathbb{S}^{h-1}} \exp\left(\frac{y_h}{\tau}\right) d\sigma_{h-1}(y). \tag{84}$$

Let $t = y_h = x \cdot y \in [-1, 1]$. The pushforward of $\sigma_{h-1}$ under the map $y \mapsto x \cdot y$ has probability density:

$$p_h(t) = \frac{\Gamma\left(\frac{h}{2}\right)}{\Gamma\left(\frac{h-1}{2}\right)\sqrt{\pi}} \left(1 - t^2\right)^{\frac{h-3}{2}}, \quad t \in [-1, 1]. \tag{85}$$

Then:

$$Z_\tau = \int_{-1}^{1} \exp\left(\frac{t}{\tau}\right) p_h(t)\, dt = \frac{\Gamma\left(\frac{h}{2}\right)}{\Gamma\left(\frac{h-1}{2}\right)\sqrt{\pi}} \int_{-1}^{1} e^{t/\tau}\left(1-t^2\right)^{\frac{h-3}{2}} dt. \tag{86}$$

A classical integral (equivalently, an integral representation of the modified Bessel $I_\nu$ ) is:

$$\int_{-1}^{1} e^{\kappa t}\left(1-t^2\right)^{\nu-\frac{1}{2}} dt = \sqrt{\pi}\,\Gamma\left(\nu+\frac{1}{2}\right)\left(\frac{2}{\kappa}\right)^\nu I_\nu(\kappa), \quad \kappa > 0, \nu > -\frac{1}{2}. \tag{87}$$

Set: $\kappa = \frac{1}{\tau}$ and $\nu = \frac{h-2}{2}$, so that $\nu - \frac{1}{2} = \frac{h-3}{2}$. Then:

$$\int_{-1}^{1} e^{t/\tau}\left(1-t^2\right)^{\frac{h-3}{2}} dt = \sqrt{\pi}\,\Gamma\left(\frac{h-1}{2}\right)(2\tau)^{\frac{h}{2}-1} I_{\frac{h}{2}-1}\left(\frac{1}{\tau}\right). \tag{88}$$

Substitute into $Z_\tau$ :

$$Z_\tau = \frac{\Gamma\left(\frac{h}{2}\right)}{\Gamma\left(\frac{h-1}{2}\right)\sqrt{\pi}} \cdot \sqrt{\pi}\,\Gamma\left(\frac{h-1}{2}\right)(2\tau)^{\frac{h}{2}-1} I_{\frac{h}{2}-1}\left(\frac{1}{\tau}\right). \tag{89}$$

Simplify:

$$Z_\tau = \Gamma\left(\frac{h}{2}\right)(2\tau)^{\frac{h}{2}-1} I_{\frac{h}{2}-1}\left(\frac{1}{\tau}\right). \tag{90}$$

**Step 3: Compute $\mathcal{F}_f\left[\sigma_{h-1}\right]$**

$$\begin{aligned}
\mathcal{F}_f\left[\sigma_{h-1}\right] &= \log Z_\tau \\
&= \log\left[\Gamma\left(\frac{h}{2}\right)(2\tau)^{\frac{h}{2}-1} I_{\frac{h}{2}-1}\left(\frac{1}{\tau}\right)\right] \\
&= \log\left(\Gamma(\nu+1)(2\tau)^\nu I_\nu\left(\frac{1}{\tau}\right)\right)
\end{aligned} \tag{91}$$

$\square$

## K.2. Details of Theorem 2

In this section, we provide proofs of Theorem 2 that is proposed in Sec. 3.4. We also provide details and proofs of the auxiliary theorems (Theorem S3 and Theorem S4) and the technical lemmas (Lemma 5, Lemma 6, Lemma 7, Lemma 8 and Lemma 9) that support the proof Theorem 2. For convenience in reading, let us recall some related notions and definitions.

- $h, N \in \mathbb{N}$.

- $\mathbb{S}^{h-1} = \{ z \in \mathbb{R}^h : \|z\| = 1 \}$.

- $X = (x_1, \ldots, x_N) \in (\mathbb{S}_{h-1})^N$.

- $Y = (y_1, \ldots, y_N) \in (\mathbb{S}_{h-1})^N$.

- $\mu_x = \frac{1}{N} \sum_{i=1}^N x_i$.

- $\mu_y = \frac{1}{N} \sum_{i=1}^N y_i$.

- $c_x = \frac{\mu_x}{\|\mu_x\|}$.

- $c_y = \frac{\mu_y}{\|\mu_y\|}$.

**Definition** (Multimodal Contrastive Loss (MCL Loss)). Let $(X, Y)$ be an $N$-pair configuration, where $X = (x_1, \ldots, x_N) \in (\mathbb{S}^{h-1})^N$ and $Y = (y_1, \ldots, y_N) \in (\mathbb{S}^{h-1})^N$. $\forall \tau > 0$, the multimodal contrastive loss $\mathcal{L}_{\mathrm{MCL}}(\cdot, \cdot) : (\mathbb{S}^{h-1})^N \times (\mathbb{S}^{h-1})^N \to \mathbb{R}$ is defined as:

$$\mathcal{L}_{\mathrm{MCL}} = \frac{1}{N} \sum_{i=1}^N \mathcal{L}_{\mathrm{MCL}}^i, \quad \text{where} \quad \mathcal{L}_{\mathrm{MCL}}^i = \mathcal{L}_{\mathcal{X} \to \mathcal{Y}}(x_i; Y) + \mathcal{L}_{\mathcal{Y} \to \mathcal{X}}(y_i; X).$$

Here, $\mathcal{L}_{\mathcal{X} \to \mathcal{Y}}$ is the $\mathcal{X}$-to-$\mathcal{Y}$ alignment and $\mathcal{L}_{\mathcal{Y} \to \mathcal{X}}$ is the $\mathcal{Y}$-to-$\mathcal{X}$ alignment, which are defined respectively as:

$$\mathcal{L}_{\mathcal{X} \to \mathcal{Y}}(x_i; Y) = -\log \frac{\exp(x_i \cdot y_i / \tau)}{\sum_{j=1}^N \exp(x_i \cdot y_j / \tau)}, \quad \mathcal{L}_{\mathcal{Y} \to \mathcal{X}}(y_i; X) = -\log \frac{\exp(x_i \cdot y_i / \tau)}{\sum_{j=1}^N \exp(x_j \cdot y_i / \tau)}.$$

**Definition**(Modality Gap) Let $(X, Y)$ be an $N$-pair configuration, where $X = (x_1, \ldots, x_N) \in (\mathbb{S}^{h-1})^N$ and $Y = (y_1, \ldots, y_N) \in (\mathbb{S}^{h-1})^N$. The modality gap between $X$ and $Y$ can be expressed as the angle between the center representations:

$$\Delta_\theta = \cos^{-1}(c_x \cdot c_y).$$

**Definition** (vMF Distribution). $\forall c \in \mathbb{S}^{h-1}$ and $\kappa \geq 0$, the probability density of a random $h$-dimensional unit vector $z \sim \mathrm{vMF}(c, \kappa)$ is given by:

$$f_h(z; c, \kappa) = D_h(\kappa) e^{\kappa c^\top z}, \quad \text{where} \quad D_h(\kappa) = \frac{\kappa^\nu}{(2\pi)^{\nu+1} I_\nu(\kappa)}.$$

Here, $\nu = h/2 - 1$, and $I_\nu(\cdot) : \mathbb{R} \to \mathbb{R}$ is the modified Bessel function of the first kind of order $\nu$, which is defined as:

$$I_\nu(x) = \sum_{k=0}^\infty \frac{1}{k! \Gamma(\nu + k + 1)} \left( \frac{x}{2} \right)^{2k+\nu}.$$

**Definition** (Function $\tilde{M}$). $\forall \kappa, \tau > 0$, a function $\tilde{M}_\kappa(\cdot, \cdot) : [-1, 1] \times [0, 1] \to \mathbb{R}_0^+$ is defined as:

$$\tilde{M}_\kappa(w, t) = \sqrt{\kappa^2 + \frac{2\kappa w}{\tau} + \frac{t^2}{\tau^2}}.$$

**Definition** (Function $\tilde{\mathcal{J}}$). $\forall \kappa, \nu, \tau > 0$, $\tilde{\mathcal{J}}(\cdot, \cdot, \cdot; \kappa, \nu) : [-1, 1] \times [-1, 1] \times [0, 1] \to \mathbb{R}$ is defined as:

$$\tilde{\mathcal{J}}(w_1, w_2, t; \kappa, \nu) = -\frac{w_1}{\tau} + \log\left(\frac{I_\nu\left(\tilde{M}_\kappa(w_2, t)\right)}{\tilde{M}_\kappa(w_2, t)^\nu}\right) - \log\left(\frac{I_\nu(\kappa)}{\kappa^\nu}\right).$$

**Definition** (Function $M$). $\forall \kappa, \tau > 0$, a function $M_\kappa(\cdot) : [-1, 1] \to \mathbb{R}_0^+$ is defined as:

$$M_\kappa(w) = \sqrt{\kappa^2 + \frac{2\kappa w}{\tau} + \frac{1}{\tau^2}}$$
$$= \tilde{M}_\kappa(w, 1).$$

**Definition** (Function $\mathcal{J}$). $\forall \kappa, \nu, \tau > 0$, a function $\mathcal{J}(\cdot; \kappa, \nu) : [-1, 1] \to \mathbb{R}$ is defined as:

$$\mathcal{J}(w; \kappa, \nu) = -\frac{w}{\tau} + \log\left(\frac{I_\nu(M_\kappa(w))}{M_\kappa(w)^\nu}\right) - \log\left(\frac{I_\nu(\kappa)}{\kappa^\nu}\right)$$
$$= \tilde{\mathcal{J}}(w, w, 1; \kappa, \nu).$$

**Definition** (Function $\hat{\mathcal{J}}$). $\forall \kappa, \nu, \tau > 0$, a function $\hat{\mathcal{J}}(\cdot, \cdot; \kappa, \nu) : [-1, 1] \times [0, 1] \to \mathbb{R}$ is defined as:

$$\hat{\mathcal{J}}(w, t; \kappa, \nu) = -\frac{w}{\tau} + \log\left(\frac{I_\nu\left(\tilde{M}_\kappa(w, t)\right)}{\tilde{M}_\kappa(w, t)^\nu}\right) - \log\left(\frac{I_\nu(\kappa)}{\kappa^\nu}\right)$$
$$= \tilde{\mathcal{J}}(w, w, t; \kappa, \nu).$$

### K.2.1. PROOF OF THEOREM 2

In this subsection, we provide the proof of Theorem 2. For convenience in reading, we first restate Theorem 2 here.

**Theorem 2.** [Restate] Let $(X, Y)$ be an $N$-pair configuration, where $X = (x_1, \ldots, x_N) \in (\mathbb{S}^{h-1})^N$ are *iid* samples from $\mu_x = \text{vMF}(c_x, \kappa_x)$, and $Y = (y_1, \ldots, y_N) \in (\mathbb{S}^{h-1})^N$ are *iid* samples from $\mu_y = \text{vMF}(c_y, \kappa_y)$. Let $\nu = h/2 - 1$. Suppose there exists an index $i = c$ such that $x_c = c_x$, $y_c = c_y$. Denote $\Delta_\theta = \cos^{-1}(c_x \cdot c_y)$. For any fixed $\kappa_x, \kappa_y > 0$, it holds that:

$$\begin{aligned}
\lim_{N \to \infty} \mathcal{L}_{\text{MCL}}^c - 2\log(N) &= \mathcal{J}(\cos(\Delta_\theta); \kappa_y, \nu) + \mathcal{J}(\cos(\Delta_\theta); \kappa_x, \nu) \\
&= \tilde{\mathcal{J}}(\cos(\Delta_\theta), \cos(\Delta_\theta), 1; \kappa_y, \nu) + \tilde{\mathcal{J}}(\cos(\Delta_\theta), \cos(\Delta_\theta), 1; \kappa_x, \nu) \\
&\geq \mathcal{J}(1; \kappa_y, \nu) + \mathcal{J}(1; \kappa_x, \nu) \\
&= \tilde{\mathcal{J}}(1, 1, 1; \kappa_y, \nu) + \tilde{\mathcal{J}}(1, 1, 1; \kappa_x, \nu),
\end{aligned}$$

where equality is attained if and only if there exists a configuration of $(X, Y)$ such that:

(A3) $\Delta_\theta = \cos^{-1}(c_x \cdot c_y) = 0$.

*Proof.* We first decompose $\lim_{N \to \infty} \mathcal{L}_{\text{MCL}}^c - 2\log(N)$ into two parts:

$$\lim_{N\to\infty} \mathcal{L}^c_{\mathrm{MCL}} - 2\log(N) = \lim_{N\to\infty} \mathcal{L}_{\mathcal{X}\to\mathcal{Y}}(c_x; Y) - \log(N)$$
$$+ \lim_{N\to\infty} \mathcal{L}_{\mathcal{Y}\to\mathcal{X}}(c_y; X) - \log(N). \tag{92}$$

According to Theorem S4, the convergent function and its lower bound of $\mathcal{L}_{\mathcal{X}\to\mathcal{Y}}$ are:

$$\lim_{N\to\infty} \mathcal{L}_{\mathcal{X}\to\mathcal{Y}}(c_x; Y) - \log(N) = \mathcal{J}(\cos{(\Delta_\theta)}; \kappa_y, \nu) \geq \mathcal{J}(1; \kappa_y, \nu), \tag{93}$$

where equality is attained if and only if there exists a configuration of $(X, Y)$ such that:

(i) $\Delta_\theta = \cos^{-1}(c_x \cdot c_y) = 0.$

This Theorem also holds for $\mathcal{L}_{\mathcal{Y}\to\mathcal{X}}$:

$$\lim_{N\to\infty} \mathcal{L}_{\mathcal{Y}\to\mathcal{X}}(c_y; X) - \log(N) = \mathcal{J}(\cos{(\Delta_\theta)}; \kappa_x, \nu) \geq \mathcal{J}(1; \kappa_x, \nu), \tag{94}$$

where equality is attained if and only if there exists a configuration of $(X, Y)$ such that:

(ii) $\Delta_\theta = \cos^{-1}(c_x \cdot c_y) = 0.$

Combining Eq. (93), Eq. (94), and consider $\mathcal{J}(w; \kappa, \nu) = \tilde{\mathcal{J}}(w, w, 1; \kappa, \nu)$, we reach the conclusion that:

$$\begin{aligned}
\lim_{N\to\infty} \mathcal{L}^c_{\mathrm{MCL}} - 2\log(N) &= \mathcal{J}(\cos{(\Delta_\theta)}; \kappa_y, \nu) + \mathcal{J}(\cos{(\Delta_\theta)}; \kappa_x, \nu) \\
&= \tilde{\mathcal{J}}(\cos{(\Delta_\theta)}, \cos{(\Delta_\theta)}, 1; \kappa_y, \nu) + \tilde{\mathcal{J}}(\cos{(\Delta_\theta)}, \cos{(\Delta_\theta)}, 1; \kappa_x, \nu) \\
&\geq \mathcal{J}(1; \kappa_y, \nu) + \mathcal{J}(1; \kappa_x, \nu) \\
&= \tilde{\mathcal{J}}(1, 1, 1; \kappa_y, \nu) + \tilde{\mathcal{J}}(1, 1, 1; \kappa_x, \nu),
\end{aligned} \tag{95}$$

where equality is attained if and only if there exists a configuration of $(X, Y)$ such that:

(A3) $\Delta_\theta = \cos^{-1}(c_x \cdot c_y) = 0.$

$\square$

### K.2.2. AUXILIARY THEOREMS PART 2

In this subsection, we provide details and proofs of the auxiliary theorems (Theorem S3 and Theorem S4) that support the proof of Theorem 2.

**Theorem S3.** Let $(X, Y)$ be an $N$-pair configuration, where $X = (x_1, \ldots, x_N) \in (\mathbb{S}^{h-1})^N$ are *iid* samples from $\mu_x = \mathrm{vMF}(c_x, \kappa_x)$, and $Y = (y_1, \ldots, y_N) \in (\mathbb{S}^{h-1})^N$ are *iid* samples from $\mu_y = \mathrm{vMF}(c_y, \kappa_y)$. Let $\nu = h/2 - 1$ and $\kappa_y > 0$.

$\forall x_i \in X$, denote $w_i = x_i \cdot y_i$ and $w_{x_i, c_y} = x_i \cdot c_y$. It holds that:

$$\begin{aligned}
\lim_{N\to\infty} \mathcal{L}_{\mathcal{X}\to\mathcal{Y}}(x_i; Y) - \log(N) &= \lim_{N\to\infty} -\log \frac{\exp{(x_i \cdot y_i/\tau)}}{\sum_{j=1}^{N} \exp{(x_i \cdot y_j/\tau)}} - \log(N) \\
&= -\frac{w_i}{\tau} + \log\left( \frac{I_\nu\left(M_{\kappa_y}\left(w_{x_i, c_y}\right)\right)}{M_{\kappa_y}\left(w_{x_i, c_y}\right)^\nu} \right) - \log\left( \frac{I_\nu\left(\kappa_y\right)}{\kappa_y^\nu} \right) \\
&= \tilde{\mathcal{J}}(w_i, w_{x_i, c_y}, 1; \kappa_y, \nu),
\end{aligned} \tag{96}$$

where $\forall \kappa \geq 0, \tau > 0, M_\kappa (\cdot) : [-1, 1] \to \mathbb{R}_0^+$ is defined as:

$$M_\kappa (w) = \sqrt{\kappa^2 + \frac{2\kappa w}{\tau} + \frac{1}{\tau^2}}. \tag{97}$$

and $I_\nu$ is the modified Bessel function of the first kind of order $\nu$, which is defined as:

$$I_\nu (m) = \sum_{k=0}^{\infty} \frac{1}{k!\Gamma(\nu + k + 1)} \left(\frac{m}{2}\right)^{2k+\nu}. \tag{98}$$

Suppose there exists an index $i = c$ such that $x_c = c_x$, $y_c = c_y$. Denote $w_c = c_x \cdot c_y$. It holds that:

$$\lim_{N \to \infty} \mathcal{L}_{\mathcal{X} \to \mathcal{Y}}(c_x; Y) - \log(N) = -\frac{w_c}{\tau} + \log\left(\frac{I_\nu \left(M_{\kappa_y} (w_c)\right)}{M_{\kappa_y} (w_c)^\nu}\right) - \log\left(\frac{I_\nu (\kappa_y)}{\kappa_y^\nu}\right)$$

$$= \mathcal{J}(w_c; \kappa_y, \nu) = \tilde{\mathcal{J}}(w_c, w_c, 1; \kappa_y, \nu). \tag{99}$$

*Proof.* **Step 1**: We start the proof by find the convergent function of $\mathcal{L}_{\mathcal{X} \to \mathcal{Y}}(x_i; Y)$ as $N \to \infty$. Same with Eq. (31) of Theorem S1, $\forall x_i \in X$, the $\mathcal{X}$-to-$\mathcal{Y}$ alignment of $x_i$ can be rewritten as:

$$\mathcal{L}_{\mathcal{X} \to \mathcal{Y}}(x_i; Y) = -\log \frac{\exp\left(x_i \cdot y_i/\tau\right)}{\sum_j \exp\left(x_i \cdot y_j/\tau\right)}$$

$$= -\frac{x_i \cdot y_i}{\tau} + \log\left(N\frac{1}{N}\sum_{j=1}^{N} \exp\left(\frac{x_i \cdot y_j}{\tau}\right)\right) \tag{100}$$

$$= -\frac{x_i \cdot y_i}{\tau} + \log\left(\frac{1}{N}\sum_{j=1}^{N} \exp\left(\frac{x_i \cdot y_j}{\tau}\right)\right) + \log(N).$$

Lemma 2 shows that:

$$\lim_{N \to \infty} \log\left(\frac{1}{N}\sum_{j=1}^{N} \exp\left(\frac{x_i \cdot y_j}{\tau}\right)\right) = \log\left(\mathbb{E}_{y \sim \mu_y}\left[\exp\left(\frac{x_i \cdot y}{\tau}\right)\right]\right). \tag{101}$$

According to the moment-generating function of the vMF distribution:

$$\mathbb{E}_{y \sim \mu_y}[\exp\left(\frac{x_i \cdot y}{\tau}\right)] = \mathbb{E}_{y \sim \mu_y}\left[\exp\left(\frac{x_i}{\tau} \cdot y\right)\right] = \frac{I_\nu (\kappa_y')}{I_\nu (\kappa_y)} \left(\frac{\kappa_y}{\kappa_y'}\right)^\nu, \tag{102}$$

$$\text{where } \kappa_y' = \|\kappa_y c_y + \frac{x_i}{\tau}\|_2.$$

Then we have:

$$\lim_{N \to \infty} \mathcal{L}_{\mathcal{X} \to \mathcal{Y}}(x_i; Y) - \log(N) = -\frac{x_i \cdot y_i}{\tau} + \log\left(\frac{I_\nu (\kappa_y')}{\kappa_y'^\nu}\right) - \log\left(\frac{I_\nu (\kappa_y)}{\kappa_y^\nu}\right). \tag{103}$$

**Step 2**: we will transform $\mathcal{L}_{\mathcal{X} \to \mathcal{Y}}$ from a function of vectors to a function of angles between vectors.

Without loss of generality, we assume the coordinate of $c_y$ as

$$c_y = (1, 0, \cdots, 0). \tag{104}$$

Denote $\cos\left(\theta_{x_i, c_y}\right) = x_i \cdot c_y$. Then $x_i$ can be represented as:

$$
\begin{aligned}
x_i &= \left(\cos\left(\theta_{x_i, c_y}\right), u\sin\left(\theta_{x_i, c_y}\right)\right) \\
&= \left(\cos\left(\theta_{x_i, c_y}\right), u_2\sin\left(\theta_{x_i, c_y}\right), u_3\sin\left(\theta_{x_i, c_y}\right), \ldots, u_h\sin\left(\theta_{x_i, c_y}\right)\right),
\end{aligned}
\tag{105}
$$

where $u = (0, u_2, u_3, \ldots, u_h) \cong \mathbb{S}^{h-2} \in \mathbb{S}^{h-1}$ is a unit vector orthogonal to the first axis with:

$$\|u\| = 0 + u_2^2 + u_3^2 + \cdots + u_h^2 = 1. \tag{106}$$

According to Eq. (104), Eq. (105) and Eq. (106), $\kappa'_y$ (in Eq. (102)) can re-rewritten as:

$$
\begin{aligned}
\kappa'_y &= \left\| \kappa_y c_y + \frac{x_i}{\tau} \right\|_2 \\
&= \sqrt{\left(\kappa_y + \frac{\cos\left(\theta_{x_i, c_y}\right)}{\tau}\right)^2 + \sum_{i=2}^{h}\left(\frac{\sin\left(\theta_{x_i, c_y}\right)u_i}{\tau}\right)^2} \\
&= \sqrt{\left(\kappa_y + \frac{\cos\left(\theta_{x_i, c_y}\right)}{\tau}\right)^2 + \frac{\sin^2\left(\theta_{x_i, c_y}\right)}{\tau^2}} \\
&= \sqrt{\kappa_y^2 + \frac{2\kappa_y\cos\left(\theta_{x_i, c_y}\right)}{\tau} + \frac{1}{\tau^2}} \\
&= M_{\kappa_y}\left(\cos\left(\theta_{x_i, c_y}\right)\right).
\end{aligned}
\tag{107}
$$

Consider that $w_i = x_i \cdot y_i$, $w_{x_i, c_y} = \cos\left(\theta_{x_i, c_y}\right) = x_i \cdot c_y$, putting Eq. (103) and Eq. (107) together, we have:

$$
\begin{aligned}
\lim_{N \to \infty} \mathcal{L}_{\mathcal{X} \to \mathcal{Y}}(x_i; Y) - \log(N) &= -\frac{x_i \cdot y_i}{\tau} + \log\left(\frac{I_\nu\left(\kappa'_y\right)}{\kappa'^\nu_y}\right) - \log\left(\frac{I_\nu\left(\kappa_y\right)}{\kappa_y^\nu}\right) \\
&= -\frac{w_i}{\tau} + \log\left(\frac{I_\nu\left(M_{\kappa_y}\left(w_{x_i, c_y}\right)\right)}{M_{\kappa_y}\left(w_{x_i, c_y}\right)^\nu}\right) - \log\left(\frac{I_\nu\left(\kappa_y\right)}{\kappa_y^\nu}\right) \\
&= \tilde{\mathcal{J}}(w_i, w_{x_i, c_y}, 1; \kappa_y, \nu).
\end{aligned}
\tag{108}
$$

When there exists a data pair $i = c$ such that $x_c = c_x$, $y_c = c_y$, $w_i = w_{x_i, c_y} = w_c$, then we have:

$$
\begin{aligned}
\lim_{N \to \infty} \mathcal{L}_{\mathcal{X} \to \mathcal{Y}}(c_x; Y) - \log(N) &= -\frac{w_c}{\tau} + \log\left(\frac{I_\nu\left(M_{\kappa_y}\left(w_c\right)\right)}{M_{\kappa_y}\left(w_c\right)^\nu}\right) - \log\left(\frac{I_\nu\left(\kappa_y\right)}{\kappa_y^\nu}\right) \\
&= \mathcal{J}(w_c; \kappa_y, \nu) = \tilde{\mathcal{J}}(w_c, w_c, 1; \kappa_y, \nu).
\end{aligned}
\tag{109}
$$

$\square$

**Theorem S4.** Let $(X, Y)$ be an $N$-pair configuration, where $X = (x_1, \ldots, x_N) \in (\mathbb{S}^{h-1})^N$ are *iid* samples from $\mu_x = \text{vMF}(c_x, \kappa_x)$, and $Y = (y_1, \ldots, y_N) \in (\mathbb{S}^{h-1})^N$ are *iid* samples from $\mu_y = \text{vMF}(c_y, \kappa_y)$. Let $\nu = h/2 - 1$. Suppose there exists an index $i = c$ such that $x_c = c_x$, $y_c = c_y$. Denote $\Delta_\theta = \cos^{-1}(c_x \cdot c_y)$. For any fixed $\kappa_y > 0$, it holds that:

$$\lim_{N \to \infty} \mathcal{L}_{\mathcal{X} \to \mathcal{Y}}(c_x; Y) - \log(N) = \mathcal{J}(\cos\left(\Delta_\theta\right); \kappa_y, \nu) \geq \mathcal{J}(1; \kappa_y, \nu), \tag{110}$$

where equality is attained if and only if there exists a configuration of $(X, Y)$ such that:

(B3)  $\Delta_\theta = \cos^{-1}(c_x \cdot c_y) = 0.$

*Proof.* **Step 1**: We start the proof by find the convergent function of $\mathcal{L}_{\mathcal{X} \to \mathcal{Y}}(c_x; Y)$ as $N \to \infty$. Denote $w_c = c_x \cdot c_y$. $\forall \kappa_y > 0$, as prove in Theorem S3:

$$
\begin{aligned}
\lim_{N \to \infty} \mathcal{L}_{\mathcal{X} \to \mathcal{Y}}(c_x; Y) - \log(N) &= \lim_{N \to \infty} -\log \frac{\exp(c_x \cdot c_y / \tau)}{\sum_{j=1}^{N} \exp(c_x \cdot y_j / \tau)} - \log(N) \\
&= -\frac{w_c}{\tau} + \log\left(\frac{I_\nu\left(M_{\kappa_y}(w_c)\right)}{M_{\kappa_y}(w_c)^\nu}\right) - \log\left(\frac{I_\nu(\kappa_y)}{\kappa_y^\nu}\right) \\
&= \mathcal{J}(w_c; \kappa_y, \nu).
\end{aligned}
\tag{111}
$$

where $\forall \kappa \geq 0, \tau > 0$, $\mathcal{J}(\cdot; \kappa, \nu)$ is a function on $[-1, 1]$ and $M_\kappa(\cdot) : [-1, 1] \to \mathbb{R}_0^+$ is defined as:

$$
M_\kappa(w) = \sqrt{\kappa^2 + \frac{2\kappa w}{\tau} + \frac{1}{\tau^2}},
\tag{112}
$$

and $I_\nu$ is the modified Bessel function of the first kind of order $\nu$, which is defined as:

$$
I_\nu(m) = \sum_{k=0}^{\infty} \frac{1}{k!\Gamma(\nu + k + 1)} \left(\frac{m}{2}\right)^{2k+\nu}.
\tag{113}
$$

**Step 2**: Next, we find the minimal value and the optimal condition of convergent function.

As shown in Lemma 5 (set $s = 1$), $\mathcal{J}(w; \kappa, \nu) = \tilde{J}(w, w, 1; \kappa, \nu)$ is a concave function of $w$. When a function is concave, its minimal value occurs at the endpoints of its domain. Therefore :

$$
\mathcal{J}(w_c; \kappa_y, \nu) \geq \min\{\mathcal{J}(-1; \kappa_y, \nu), \mathcal{J}(1; \kappa_y, \nu)\}.
\tag{114}
$$

According to Lemma 6:

$$
\mathcal{J}(-1; \kappa_y, \nu) \geq \mathcal{J}(1; \kappa_y, \nu).
\tag{115}
$$

Therefore, we conclude:

$$
\lim_{N \to \infty} \mathcal{L}_{\mathcal{X} \to \mathcal{Y}}(c_x; Y) - \log(N) = \mathcal{J}(\cos(\Delta_\theta); \kappa_y, \nu) \geq \mathcal{J}(1; \kappa_y, \nu),
\tag{116}
$$

where equality is attained if and only if the following conditions hold:

(B3)  $\Delta_\theta = \cos^{-1}(c_x \cdot c_y) = 0.$

$\square$

### K.2.3. TECHNICAL LEMMAS PART 2

In this subsection, we provide details and proofs of technical lemmas (Lemma 5, Lemma 6, Lemma 7, Lemma 8 and Lemma 9) that support the proof of Theorem 2, Theorem S3 and Theorem S4.

**Lemma 5.** $\forall \kappa, \nu, \tau > 0$ and $s \in [0, 1]$, a function $\hat{\mathcal{J}}_{t=s}(\cdot; \kappa, \nu) : (-1, 1] \to \mathbb{R}$ is defined as:

$$\hat{\mathcal{J}}_{t=s}(w; \kappa, \nu) = -\frac{w}{\tau} + \log\left(\frac{I_\nu\left(\tilde{M}_{t=s}(w)\right)}{\tilde{M}_{t=s}(w)^\nu}\right) - \log\left(\frac{I_\nu(\kappa)}{\kappa^\nu}\right) \tag{117}$$

$$= \hat{\mathcal{J}}(w, t = s; \kappa, \nu) = \tilde{J}(w, w, t = s; \kappa, \nu),$$

where $\tilde{M}_{t=s}(\cdot) : (-1, 1] \to \mathbb{R}^+$ is defined as:

$$\tilde{M}_{t=s}(w) = \sqrt{\kappa^2 + \frac{2\kappa w}{\tau} + \frac{s^2}{\tau^2}} = \tilde{M}_\kappa(w, t = s), \tag{118}$$

and $I_\nu$ is the modified Bessel function of the first kind of order $\nu$, which is defined as:

$$I_\nu(m) = \sum_{k=0}^\infty \frac{1}{k! \Gamma(\nu + k + 1)} \left(\frac{m}{2}\right)^{2k+\nu}. \tag{119}$$

It holds that, for any fixed $s$, $\hat{\mathcal{J}}_{t=s}(\cdot)$ is a strictly decreasing function when $w \in [0, s]$ and a concave function $w \in (-1, 1]$.

*Proof.* Let us first decompose the function $\hat{\mathcal{J}}_{t=s}$. Denote two functions $G_1(w)$ and $G_2(w)$ as:

$$\begin{aligned} G_1(w) &= -\frac{w}{\tau}, \\ G_3(m) &= \log(I_\nu(m)) - \nu \log(m), \\ G_2(w) &= G_3\left(\tilde{M}_{t=s}(w)\right) \\ &= \log\left(I_\nu\left(\tilde{M}_{t=s}(w)\right)\right) - \nu \log\left(\tilde{M}_{t=s}(w)\right). \end{aligned} \tag{120}$$

Denote the function $G(w)$ and the constant $C$ as:

$$\begin{aligned} G(w) &= G_1(w) + G_2(w), \\ C &= -\log\left(\frac{I_\nu(\kappa)}{\kappa^\nu}\right). \end{aligned} \tag{121}$$

Then the function $\hat{\mathcal{J}}_{t=s}$ can be written as:

$$\hat{\mathcal{J}}_{t=s}(w; \kappa, \nu) = -\frac{w}{\tau} + \log\left(\frac{I_\nu\left(\tilde{M}_{t=s}(w)\right)}{\tilde{M}_{t=s}(w)^\nu}\right) - \log\left(\frac{I_\nu(\kappa)}{\kappa^\nu}\right) \tag{122}$$

$$= G(w) + C.$$

Now, we investigate derivatives of $\hat{\mathcal{J}}_{t=s}$.

The first derivative of $G_1$ is:

$$G_1'(w) = -\frac{1}{\tau} < 0. \tag{123}$$

The second derivative of $G_1$ is:

$$G_1'' (w) = 0. \tag{124}$$

According to Lemma 7, the first derivative of $G_3 (m)$ is:

$$G_3' (m) = \frac{I_{\nu+1} (m)}{I_\nu (m)} \in (0, 1). \tag{125}$$

The derivative of $\tilde{M}_{t=s}$ is:

$$
\begin{aligned}
\tilde{M}_{t=s}' (w) &= \frac{d}{dw} \left( \kappa_y^2 + \frac{s^2}{\tau^2} + 2 \frac{\kappa}{\tau} w \right)^{1/2} \\
&= \frac{1}{2} \left( \kappa_y^2 + \frac{s^2}{\tau^2} + 2 \frac{\kappa}{\tau} w \right)^{-1/2} \cdot 2 \frac{\kappa}{\tau} \\
&= \frac{\kappa}{\tau} \frac{1}{\tilde{M}_{t=s} (w)} \\
&> 0.
\end{aligned}
\tag{126}
$$

Then, the first derivative of $G_2$ is:

$$
\begin{aligned}
G_2' (w) &= G_3' \left( \tilde{M}_{t=s} (w) \right) \tilde{M}_{t=s}' (w) \\
&= \frac{I_{\nu+1} \left( \tilde{M}_{t=s} (w) \right)}{I_\nu \left( \tilde{M}_{t=s} (w) \right)} \tilde{M}_{t=s}' (w) \\
&= \frac{\kappa}{\tau} \frac{1}{\tilde{M}_{t=s} (w)} \frac{I_{\nu+1} \left( \tilde{M}_{t=s} (w) \right)}{I_\nu \left( \tilde{M}_{t=s} (w) \right)}.
\end{aligned}
\tag{127}
$$

Combining Eq. (123) and Eq. (127), we have:

$$
\begin{aligned}
\hat{\mathcal{J}}_{t=s}' (w; \kappa, \nu) &= G' (w) \\
&= -\frac{1}{\tau} + \frac{\kappa}{\tau} \frac{1}{\tilde{M}_{t=s} (w)} \frac{I_{\nu+1} \left( \tilde{M}_{t=s} (w) \right)}{I_\nu \left( \tilde{M}_{t=s} (w) \right)}.
\end{aligned}
\tag{128}
$$

Since:

$$
\begin{aligned}
\tilde{M}_{t=s} (w) \geq \kappa &\Leftrightarrow \frac{2\kappa w}{\tau} + \frac{s^2}{\tau^2} \geq 0 \\
&\Leftrightarrow w \geq -\frac{s^2}{2\kappa\tau} \\
&\Leftarrow w \geq 0.
\end{aligned}
\tag{129}
$$

thus, when $w \in [0, 1]$, $\tilde{M}_{t=s} (w) \geq \kappa$ holds. Combining this and Eq. (125), we have:

$$G'(w) \leq -\frac{1}{\tau} + \frac{\kappa}{\tau} \frac{1}{\kappa} \frac{I_{\nu+1}\left(\tilde{M}_{t=s}(w)\right)}{I_\nu\left(\tilde{M}_{t=s}(w)\right)}$$
$$< -\frac{1}{\tau} + \frac{1}{\tau} \tag{130}$$
$$= 0.$$

So we can conclude that, for any fixed $s$, $\hat{\mathcal{J}}_{t=s}(\cdot)$ is a strictly decreasing function on $[0, s]$.

Denote:

$$H(m) = \frac{1}{m} \frac{I_{\nu+1}(m)}{I_\nu(m)}, \tag{131}$$

according to Lemma 8,

$$H'(m) < 0. \tag{132}$$

Since $G_2'(w)$ can be written as:

$$G_2'(w) = \frac{\kappa}{\tau} H\left(\tilde{M}_{t=s}(w)\right), \tag{133}$$

combining Eq. (126) and Eq. (133), we have

$$G_2''(w) = \frac{\kappa}{\tau} H'\left(\tilde{M}_{t=s}(w)\right) \tilde{M}_{t=s}'(w)$$
$$< 0. \tag{134}$$

Given Eq. (127) and Eq. (134), we can conclude that $G_2$ is an increasing and concave function. Combining Eq. (124) and Eq. (134), we have:

$$\hat{\mathcal{J}}_{t=s}''(w; \kappa, \nu) = G''(w)$$
$$= 0 + G_2''(w) \tag{135}$$
$$< 0.$$

So we can conclude that, for any fixed $s$, $\hat{\mathcal{J}}_{t=s}(\cdot)$ is a concave function on $(-1, 1]$.

$\square$

**Lemma 6.** $\forall \kappa, \nu, \tau > 0$, a function $\mathcal{J}(\cdot): [-1, 1] \to \mathbb{R}$ is defined as:

$$\mathcal{J}(w) = -\frac{w}{\tau} + \log\left(\frac{I_\nu(M(w))}{M(w)^\nu}\right) - \log\left(\frac{I_\nu(\kappa)}{\kappa^\nu}\right) + \log(N), \tag{136}$$

where $M_\kappa(\cdot): [-1, 1] \to \mathbb{R}$ is defined as:

$$M_\kappa(w) = \sqrt{\kappa^2 + \frac{2\kappa w}{\tau} + \frac{1}{\tau^2}}, \tag{137}$$

and $I_\nu$ is the modified Bessel function of the first kind of order $\nu$, which is defined as:

$$I_\nu(m) = \sum_{k=0}^{\infty} \frac{1}{k!\Gamma(\nu+k+1)} \left(\frac{m}{2}\right)^{2k+\nu}. \tag{138}$$

$\forall 0 < w \le 1$, *it holds that:*

$$\mathcal{J}(w) < \mathcal{J}(-w). \tag{139}$$

*Proof.* Let us first re-write Eq. (139) as:

$$\mathcal{J}(w) < \mathcal{J}(-w) \Leftrightarrow \mathcal{J}(-w) - \mathcal{J}(w) > 0, \tag{140}$$

and we will prove the inequality on RHS. Denote:

$$
\begin{aligned}
a &= M(-w) = \sqrt{\kappa^2 + \frac{1}{\tau^2} - \frac{2\kappa w}{\tau}}, \\
b &= M(w) = \sqrt{\kappa^2 + \frac{1}{\tau^2} + \frac{2\kappa w}{\tau}}.
\end{aligned}
\tag{141}
$$

In (Eq. (125) of) Lemma 5, it is shown that $M(\cdot)$ is a strictly increasing function. Then, we have:

$$0 < a < b, \tag{142}$$

and then we have:

$$
\begin{aligned}
\mathcal{J}(-w) - \mathcal{J}(w) &= \frac{w}{\tau} - \left(-\frac{w}{\tau}\right) + \log\left(\frac{I_\nu(a)}{I_\nu(b)}\right) - \nu\log\left(\frac{a}{b}\right) \\
&= \frac{2w}{\tau} + \log\frac{I_\nu(a)}{I_\nu(b)} - \nu\log\left(\frac{a}{b}\right).
\end{aligned}
\tag{143}
$$

According to Lemma 9:

$$\log\left(\frac{I_\nu(a)}{I_\nu(b)}\right) - \nu\log\left(\frac{a}{b}\right) > (a-b). \tag{144}$$

Plugging Eq. (144) into Eq. (143), we get:

$$\mathcal{J}(-w) - \mathcal{J}(w) > \frac{2w}{\tau} + (a-b) = f(w). \tag{145}$$

Combining Eq. (140) and Eq. (145), we have:

$$\mathcal{J}(w) \le \mathcal{J}(-w) \Leftrightarrow f(w) \ge 0. \tag{146}$$

Denote:

$$
\begin{aligned}
A &= \kappa^2 + \frac{1}{\tau^2}, \\
B &= \frac{2\kappa}{\tau},
\end{aligned}
\tag{147}
$$

then we have:

$$a = M(-w) = \sqrt{A - Bw},$$
$$b = M(w) = \sqrt{A + Bw}. \tag{148}$$

Observe that:

$$b - a = M(w) - M(-w) = \frac{(A + Bw) - (A - Bw)}{\sqrt{A + Bw} + \sqrt{A - Bw}}$$
$$= \frac{2Bw}{\sqrt{A + Bw} + \sqrt{A - Bw}}, \tag{149}$$

and then:

$$f(w) = \frac{2w}{\tau} \left[ 1 - \frac{2\kappa}{\sqrt{A + Bw} + \sqrt{A - Bw}} \right]. \tag{150}$$

Therefore, we have:

$$f(w) \geq 0 \Leftrightarrow \sqrt{A + Bw} + \sqrt{A - Bw} \geq 2\kappa$$
$$\Leftrightarrow \left( \sqrt{A + Bw} + \sqrt{A - Bw} \right)^2 \geq 4\kappa^2$$
$$\Leftrightarrow 2A + 2\sqrt{A^2 - B^2 w^2} \geq 4\kappa^2 \tag{151}$$
$$\Leftrightarrow \sqrt{A^2 - B^2 w^2} \geq 2\kappa^2 - A$$
$$\Leftrightarrow \sqrt{A^2 - B^2 w^2} \geq \kappa^2 - \frac{1}{\tau^2}$$

**Case 1**: $0 < \kappa < \frac{1}{\tau}$.

$\kappa^2 - \frac{1}{\tau^2} < 0$ and the last equation in Eq. (151) holds.

**Case 2**: $0 < \frac{1}{\tau} \leq \kappa$.

The Eq. (151) becomes:

$$f(w) \geq 0 \Leftrightarrow A^2 - B^2 w^2 \geq \left( \kappa^2 - \frac{1}{\tau^2} \right)^2$$
$$\Leftrightarrow \frac{4\kappa^2}{\tau^2} (1 - w^2) \geq 0 \tag{152}$$
$$\Leftrightarrow |w| \leq 1.$$

Since $0 < w \leq 1$, $f(w) \geq 0$ holds. According to Eq. (146), we conclude that:

$$0 < w \leq 1 \Rightarrow \mathcal{J}(w) \leq \mathcal{J}(-w). \tag{153}$$

$$\square$$

**Lemma 7.** $\forall \nu > 0$, a function $G_3 : \mathbb{R}_0^+ \to \mathbb{R}$ is defined as:

$$G_3(m) = \log(I_\nu(m)) - \nu \log(m). \tag{154}$$

where $I_\nu$ is the modified Bessel function of the first kind of order $\nu$, which is defined as:

$$I_\nu(m) = \sum_{k=0}^{\infty} \frac{1}{k!\Gamma(\nu + k + 1)} \left(\frac{m}{2}\right)^{2k+\nu}. \tag{155}$$

*It holds that $G_3(\cdot)$ is a strictly increasing function with $G_3'(\cdot) \in (0, 1)$*

*Proof.* The first derivative of $G_3$ is:

$$G_3'(m) = \frac{I_\nu'(m)}{I_\nu(m)} - \frac{\nu}{m}. \tag{156}$$

According to (Olver, 2010):

$$I_\nu'(m) = I_{\nu+1}(m) + \frac{\nu}{m} I_\nu(m), \tag{157}$$

then we have:

$$\frac{I_\nu'(m)}{I_\nu(m)} - \frac{\nu}{m} = \frac{I_{\nu+1}(m)}{I_\nu(m)}. \tag{158}$$

Plugging Eq. (158) into Eq. (156), we get:

$$G_3'(m) = \frac{I_{\nu+1}(m)}{I_\nu(m)}. \tag{159}$$

Since:

$$0 < I_{\nu+1}(m) < I_\nu(m), \tag{160}$$

therefore:

$$G_3'(m) = \frac{I_{\nu+1}(m)}{I_\nu(m)} \in (0, 1). \tag{161}$$

This shows that $G_3(\cdot)$ is a strictly increasing function with $G_3'(\cdot) \in (0, 1)$.

$\square$

**Lemma 8.** $\forall \nu > 0$, *a function $H(\cdot) : R^+ \to \mathbb{R}$ is defined as:*

$$H(m) = \frac{1}{m} \frac{I_{\nu+1}(m)}{I_\nu(m)}, \tag{162}$$

*where $I_\nu$ is the modified Bessel function of the first kind of order $\nu$, which is defined as:*

$$I_\nu(m) = \sum_{k=0}^{\infty} \frac{1}{k!\Gamma(\nu + k + 1)} \left(\frac{m}{2}\right)^{2k+\nu}. \tag{163}$$

*It holds that $H(m)$ is a strictly decreasing function.*

*Proof.* $\forall \nu, m \in R^+$, denote $R_\nu(m)$ as:

$$R_\nu(m) = \frac{I_{\nu+1}(m)}{I_\nu(m)}. \tag{164}$$

According to (Olver, 2010), we have:

$$I'_\nu(m) = I_{\nu+1}(m) + \frac{\nu}{m} I_\nu(m), \tag{165}$$

then:

$$
\begin{aligned}
R'_\nu(m) &= \frac{I'_{\nu+1}(m) I_\nu(m) - I_{\nu+1}(m) I'_\nu(m)}{I_\nu(m)^2} \\
&= \frac{\left(I_{\nu+2}(m) + \frac{\nu+1}{m} I_{\nu+1}(m)\right) I_\nu(m) - I_{\nu+1}(m) \left(I_{\nu+1}(m) + \frac{\nu}{m} I_\nu(m)\right)}{I_\nu(m)^2} \\
&= \frac{I_{\nu+2}(m) I_\nu(m) - I_{\nu+1}^2(m) + \frac{1}{m} I_{\nu+1}(m) I_\nu(m)}{I_\nu(m)^2} \\
&= \frac{I_{\nu+2}(m) I_\nu(m) - I_{\nu+1}^2(m)}{I_\nu(m)^2} + \frac{1}{m} R_\nu(m).
\end{aligned} \tag{166}
$$

Since $H(m)$ can be rewritten as:

$$H(m) = \frac{R_\nu(m)}{m}, \tag{167}$$

then:

$$
\begin{aligned}
H'(m) &= \frac{R'_\nu(m) m - R_\nu(m)}{m^2} \\
&= \frac{1}{m} \left(R'_\nu(m) - \frac{1}{m} R_\nu(m)\right) \\
&= \frac{1}{m} \left(\frac{I_{\nu+2}(m) I_\nu(m) - I_{\nu+1}^2(m)}{I_\nu(m)^2}\right).
\end{aligned} \tag{168}
$$

According to the Turán type inequalities for modified Bessel functions (Baricz, 2010), when $m > 0$:

$$\frac{I_{\nu+2}(m) I_\nu(m) - I_{\nu+1}^2(m)}{I_\nu(m)^2} < 0, \tag{169}$$

so

$$H'(m) < 0. \tag{170}$$

Then we can conclude that $H(m)$ is a strictly decreasing function.

$\square$

**Lemma 9.** $\forall \nu > 0$ *and* $0 < a < b$, *it holds that:*

$$\log\left(\frac{I_\nu(a)}{I_\nu(b)}\right) > \nu \log\left(\frac{a}{b}\right) + (a - b), \tag{171}$$

where $I_\nu$ is the modified Bessel function of the first kind of order $\nu$, which is defined as:

$$I_\nu(m) = \sum_{k=0}^{\infty} \frac{1}{k!\Gamma(\nu+k+1)} \left(\frac{m}{2}\right)^{2k+\nu}. \tag{172}$$

*Proof.* According to (Olver, 2010), $\forall x > 0$ and $0 < \nu_1 < \nu_2 < \infty$, we have:

$$I_{\nu_1}(x) > I_{\nu_2}(x). \tag{173}$$

Denote a function L as:

$$L(x) = \log I_\nu(x) - \nu \log(x) - x. \tag{174}$$

According to (Olver, 2010):

$$I'_\nu(m) = I_{\nu+1}(m) + \frac{\nu}{m} I_\nu(m), \tag{175}$$

then we have:

$$\frac{I'_\nu(m)}{I_\nu(m)} - \frac{\nu}{m} = \frac{I_{\nu+1}(m)}{I_\nu(m)}. \tag{176}$$

Taking Eq. (173) and Eq. (176) into account, the derivative of $L$ is:

$$\begin{aligned}
L'(x) &= \frac{I'_\nu(x)}{I_\nu(x)} - \frac{\nu}{x} - 1 \\
&= \frac{I_{\nu+1}(x)}{I_\nu(x)} - 1 \\
&< 0.
\end{aligned} \tag{177}$$

Therefore, $\forall \nu > 0, 0 < b < a$, it holds that:

$$\begin{aligned}
\log(I_\nu(a)) - \nu \log(a) - a &= L(a) \\
&> L(a) \\
&= \log(I_\nu(b)) - \nu \log(b) - b,
\end{aligned} \tag{178}$$

then we have:

$$\log\left(\frac{I_\nu(a)}{I_\nu(b)}\right) > \nu \log\left(\frac{a}{b}\right) + (a - b). \tag{179}$$

$\square$

### K.3. Details of Theorem 3

In this section, we provide proofs of Theorem 3 that is proposed in Sec. 3.6. We also provide details and proofs of the auxiliary theorems (Theorem S5 and Theorem S6) and the technical lemmas (Lemma 10, Lemma 11, Lemma 12 and Lemma 13) that support the proof Theorem 3. For convenience in reading, let us recall some related notions and definitions.

- $h, N \in \mathbb{N}$.

- $\mathbb{S}^{h-1} = \left\{ z \in \mathbb{R}^h : \|z\| = 1 \right\}$.

- $\mathbb{A} = \left\{ x \in \mathbb{R}^h : n_A \cdot x = 0 \right\}$ where $n_A$ is the normal vector of $\mathbb{A}$.

- $\mathbb{B} = \left\{ y \in \mathbb{R}^h : n_B \cdot y = 0 \right\}$ where $n_A$ is the normal vector of $\mathbb{B}$.

- $\phi = \cos^{-1}\left( \frac{n_x \cdot n_y}{\|n_x\| \cdot \|n_y\|} \right)$ and $0 < \phi_{\min} \le \phi < \frac{\pi}{2}$.

- $\mathbb{S}_X = \mathbb{S}^{h-1} \cap \mathbb{A} = \left\{ x \in \mathbb{R}^h : \|x\| = 1, n_A \cdot x = 0 \right\} \cong S^{h-2} \in \mathbb{S}^{h-1}$.

- $\mathbb{S}_Y = \mathbb{S}^{h-1} \cap \mathbb{B} = \left\{ y \in \mathbb{R}^h : \|y\| = 1, n_B \cdot y = 0 \right\} \cong S^{h-2} \in \mathbb{S}^{h-1}$.

- $\mathbb{C} = \mathbb{A} \cap \mathbb{B}$.

- $h_X = h_Y = h - 1$.

- $h_C = h - 2$.

- $P_A$: the projection matrix of $\mathbb{A}$.

- $P_B$: the projection matrix of $\mathbb{B}$.

- $P_C$: the projection matrix of $\mathbb{C}$.

- $e_A = \{ z \in \mathbb{S}_X : z \perp \mathbb{C} \}$.

- $e_B = \{ z \in \mathbb{S}_Y : z \perp \mathbb{C} \}$.

- $\mathbb{C}^\perp = \mathrm{span}\,\{e_A\} \oplus \mathrm{span}\,\{e_B\}$

- $\mathbb{R}^h = \mathbb{C} \oplus \mathbb{C}^\perp$.

- $X = (x_1, \ldots, x_N) \in (\mathbb{S}_X)^N$.

- $Y = (y_1, \ldots, y_N) \in (\mathbb{S}_Y)^N$.

- $\mu_x = \frac{1}{N} \sum_{i=1}^N x_i$.

- $\mu_y = \frac{1}{N} \sum_{i=1}^N y_i$.

- $c_x = \frac{\mu_x}{\|\mu_x\|}$.

- $c_y = \frac{\mu_y}{\|\mu_y\|}$.

**Definition** (Multimodal Contrastive Loss (MCL Loss)). Let $(X, Y)$ be an $N$-pair configuration, where $X = (x_1, \ldots, x_N) \in (\mathbb{S}^{h-1})^N$ and $Y = (y_1, \ldots, y_N) \in (\mathbb{S}^{h-1})^N$. $\forall \tau > 0$, the multimodal contrastive loss $\mathcal{L}_{\mathrm{MCL}}(\cdot, \cdot) : (\mathbb{S}^{h-1})^N \times (\mathbb{S}^{h-1})^N \to \mathbb{R}$ is defined as:

$$\mathcal{L}_{\mathrm{MCL}} = \frac{1}{N} \sum_{i=1}^N \mathcal{L}_{\mathrm{MCL}}^i, \quad \text{where } \mathcal{L}_{\mathrm{MCL}}^i = \mathcal{L}_{\mathcal{X} \to \mathcal{Y}}(x_i; Y) + \mathcal{L}_{\mathcal{Y} \to \mathcal{X}}(y_i; X).$$

Here, $\mathcal{L}_{\mathcal{X} \to \mathcal{Y}}$ is the $\mathcal{X}$-to-$\mathcal{Y}$ alignment and $\mathcal{L}_{\mathcal{Y} \to \mathcal{X}}$ is the $\mathcal{Y}$-to-$\mathcal{X}$ alignment, which are defined respectively as:

$$\mathcal{L}_{\mathcal{X} \to \mathcal{Y}}(x_i; Y) = -\log \frac{\exp(x_i \cdot y_i / \tau)}{\sum_{j=1}^N \exp(x_i \cdot y_j / \tau)}, \quad \mathcal{L}_{\mathcal{Y} \to \mathcal{X}}(y_i; X) = -\log \frac{\exp(x_i \cdot y_i / \tau)}{\sum_{j=1}^N \exp(x_j \cdot y_i / \tau)}.$$

**Definition**(Modality Gap) Let $(X, Y)$ be an $N$-pair configuration, where $X = (x_1, \ldots, x_N) \in (\mathbb{S}^{h-1})^N$ and $Y = (y_1, \ldots, y_N) \in (\mathbb{S}^{h-1})^N$. The modality gap between $X$ and $Y$ can be expressed as the angle between the center representations:

$$\Delta_\theta = \cos^{-1}(c_x \cdot c_y).$$

**Definition** (vMF Distribution). $\forall c \in \mathbb{S}^{h-1}$ and $\kappa \geq 0$, the probability density of a random $h$-dimensional unit vector $z \sim \text{vMF}(c, \kappa)$ is given by:

$$f_h(z; c, \kappa) = D_h(\kappa) e^{\kappa c^\top z}, \quad \text{where } D_h(\kappa) = \frac{\kappa^\nu}{(2\pi)^{\nu+1} I_\nu(\kappa)}.$$

Here, $\nu = h/2 - 1$, and $I_\nu(\cdot) : \mathbb{R} \to \mathbb{R}$ is the modified Bessel function of the first kind of order $\nu$, which is defined as:

$$I_\nu(x) = \sum_{k=0}^\infty \frac{1}{k! \Gamma(\nu + k + 1)} \left(\frac{x}{2}\right)^{2k+\nu}.$$

**Definition** (Function $\tilde{M}$). $\forall \kappa, \tau > 0$, a function $\tilde{M}_\kappa(\cdot, \cdot) : [-1, 1] \times [0, 1] \to \mathbb{R}_0^+$ is defined as:

$$\tilde{M}_\kappa(w, t) = \sqrt{\kappa^2 + \frac{2\kappa w}{\tau} + \frac{t^2}{\tau^2}}.$$

**Definition** (Function $\tilde{\mathcal{J}}$). $\forall \kappa, \nu, \tau > 0$, $\tilde{\mathcal{J}}(\cdot, \cdot, \cdot; \kappa, \nu) : [-1, 1] \times [-1, 1] \times [0, 1] \to \mathbb{R}$ is defined as:

$$\tilde{\mathcal{J}}(w_1, w_2, t; \kappa, \nu) = -\frac{w_1}{\tau} + \log\left(\frac{I_\nu\left(\tilde{M}_\kappa(w_2, t)\right)}{\tilde{M}_\kappa(w_2, t)^\nu}\right) - \log\left(\frac{I_\nu(\kappa)}{\kappa^\nu}\right).$$

**Definition** (Function $M$). $\forall \kappa, \tau > 0$, a function $M_\kappa(\cdot) : [-1, 1] \to \mathbb{R}_0^+$ is defined as:

$$M_\kappa(w) = \sqrt{\kappa^2 + \frac{2\kappa w}{\tau} + \frac{1}{\tau^2}}$$
$$= \tilde{M}_\kappa(w, 1).$$

**Definition** (Function $\mathcal{J}$). $\forall \kappa, \nu, \tau > 0$, a function $\mathcal{J}(\cdot; \kappa, \nu) : [-1, 1] \to \mathbb{R}$ is defined as:

$$\mathcal{J}(w; \kappa, \nu) = -\frac{w}{\tau} + \log\left(\frac{I_\nu(M_\kappa(w))}{M_\kappa(w)^\nu}\right) - \log\left(\frac{I_\nu(\kappa)}{\kappa^\nu}\right)$$
$$= \tilde{\mathcal{J}}(w, w, 1; \kappa, \nu).$$

**Definition** (Function $\tilde{M}$). $\forall \kappa, \tau > 0$, a function $\tilde{M}_\kappa(\cdot, \cdot) : [-1, 1] \times [0, 1] \to \mathbb{R}_0^+$ is defined as:

$$\tilde{M}_\kappa(w, t) = \tilde{M}_\kappa(w, t).$$

**Definition** (Function $\hat{\mathcal{J}}$). $\forall \kappa, \nu, \tau > 0$, a function $\hat{\mathcal{J}}(\cdot, \cdot; \kappa, \nu) : [-1, 1] \times [0, 1] \to \mathbb{R}$ is defined as:

$$\hat{\mathcal{J}}(w, t; \kappa, \nu) = -\frac{w}{\tau} + \log\left(\frac{I_\nu\left(\tilde{M}_\kappa(w, t)\right)}{\tilde{M}_\kappa(w, t)^\nu}\right) - \log\left(\frac{I_\nu(\kappa)}{\kappa^\nu}\right)$$

$$= \tilde{\mathcal{J}}(w, w, t; \kappa, \nu).$$

### K.3.1. PROOF OF THEOREM 3

In this subsection, we provide the proof of Theorem 3. For convenience in reading, we first restate Theorem 3 here.

**Theorem 3.** [Restate] Let $(X, Y)$ be an $N$-pair configuration, where $X = (x_1, \ldots, x_N) \in (\mathbb{S}_X \setminus \mathbb{C})^N$ are $iid$ samples from $\mu_x = \text{vMF}(c_x, \kappa_x)$, and $Y = (y_1, \ldots, y_N) \in (\mathbb{S}_Y \setminus \mathbb{C})^N$ are $iid$ samples from $\mu_y = \text{vMF}(c_y, \kappa_y)$. Let $\tilde{\nu} = (h-1)/2 - 1$. Suppose there exists an index $i = c$ such that $x_c = c_x$, $y_c = c_y$. Denote $\Delta_\theta = \cos^{-1}(c_x \cdot c_y)$ and assume that $c_x, c_y \notin \mathbb{C}$ with $c_x \cdot c_y > 0$. For any fixed $\kappa_x, \kappa_y > 0$, it holds that:

$$\lim_{N \to \infty} \mathcal{L}_{\text{MCL}}^c - 2\log(N)$$
$$= \tilde{\mathcal{J}}(\cos(\Delta_\theta), \cos(\Delta_\theta), \|P_B c_x\|; \kappa_y, \tilde{\nu}) + \tilde{\mathcal{J}}(\cos(\Delta_\theta), \cos(\Delta_\theta), \|P_A c_y\|; \kappa_x, \tilde{\nu})$$
$$\geq \tilde{\mathcal{J}}(\cos(\phi_{\min}), \cos(\phi_{\min}), \cos(\phi_{\min}); \kappa_y, \tilde{\nu}) + \tilde{\mathcal{J}}(\cos(\phi_{\min}), \cos(\phi_{\min}), \cos(\phi_{\min}); \kappa_x, \tilde{\nu}),$$

where equality is attained if and only if there exists a configuration of $(X, Y)$ such that:

(A4) $c_x \perp \mathbb{C}$ and $c_y \perp \mathbb{C}$.

(A5) $\Delta_\theta = \cos^{-1}(c_x \cdot c_y) = \phi_{\min}$.

*Proof.* We first decompose $\lim_{N \to \infty} \mathcal{L}_{\text{MCL}}^c - 2\log(N)$ into two parts:

$$\lim_{N \to \infty} \mathcal{L}_{\text{MCL}}^c - 2\log(N) = \lim_{N \to \infty} \mathcal{L}_{\mathcal{X} \to \mathcal{Y}}(c_x; Y) - \log(N)$$
$$+ \lim_{N \to \infty} \mathcal{L}_{\mathcal{Y} \to \mathcal{X}}(c_y; X) - \log(N). \tag{180}$$

Set:

$$\hat{\mathcal{J}}(w, t; \kappa, \nu) = \tilde{\mathcal{J}}(w, w, t; \kappa, \nu),$$
$$\tilde{\nu} = \tilde{\nu}, \tag{181}$$

According to Theorem S6, the convergent function and its lower bound of $\mathcal{L}_{\mathcal{X} \to \mathcal{Y}}$ are:

$$\lim_{N \to \infty} \mathcal{L}_{\mathcal{X} \to \mathcal{Y}}(c_x; Y) - \log(N) = \hat{\mathcal{J}}(\cos(\Delta_\theta), \|P_B c_x\|; \kappa_y, \tilde{\nu})$$
$$\geq \hat{\mathcal{J}}(\|P_A c_y\|, \|P_A c_y\|, \cos(\phi); \kappa_y, \tilde{\nu}). \tag{182}$$

where equality is attained if and only if there exists a configuration of $(X, Y)$ such that:

(i) $c_x \perp \mathbb{C}$.

(ii) $c_x = \frac{P_A c_y}{\|P_A c_y\|}$.

This Theorem also holds for $\mathcal{L}_{\mathcal{Y} \to \mathcal{X}}$:

$$\lim_{N \to \infty} \mathcal{L}_{\mathcal{Y} \to \mathcal{X}}(c_y; X) - \log(N) = \hat{\mathcal{J}}(\cos(\Delta_\theta), \|P_A c_y\|; \kappa_x, \tilde{\nu}) \tag{183}$$
$$\geq \hat{\mathcal{J}}(\|P_B c_x\|, \|P_B c_x\|, \cos(\phi); \kappa_x, \tilde{\nu}).$$

where equality is attained if and only if there exists a configuration of $(X, Y)$ such that:

(iii) $c_y \perp \mathbb{C}$.

(iv) $c_y = \frac{P_B c_x}{\|P_B c_x\|}$.

According to Lemma 13, for some $\lambda_x, \lambda_y > 0$ such that the projections of $x$ and $y$ are collinear with the other vector:

(1) The orthogonal projection of $x$ on $\mathbb{B}$ is a scalar multiple of $y$:

$$P_B x = \lambda_x y, \quad \lambda_x \neq 0,$$

(2) The orthogonal projection of $y$ on $\mathbb{A}$ is a scalar multiple of $x$:

$$P_A y = \lambda_y x, \quad \lambda_y \neq 0,$$

if and only if the following condition holds:

(v) Either $x \perp \mathbb{C}$ and $y \perp \mathbb{C}$, or $x = \pm y \in \mathbb{C}$.

Since $c_x, c_y \notin \mathbb{C}$, there is only one configuration in (v) that satisfies (ii) + (iv), that is $c_x \perp \mathbb{C}$ and $c_y \perp \mathbb{C}$. In this case, Lemma 13 shows that:

$$\cos(\Delta_\theta) = \cos(\phi) \geq \cos(\phi_{\min}),$$
$$\|P_A c_y\| = \|P_B c_x\| = \cos(\phi),$$
$$P_B c_x = \cos(\phi) c_y, \tag{184}$$
$$P_A c_y = \cos(\phi) c_x.$$

Combining Eq. (182), Eq. (183) and Eq. (184), we have:

$$\lim_{N \to \infty} \mathcal{L}_{\text{MCL}}^c - 2\log(N) = \hat{\mathcal{J}}(\cos(\Delta_\theta), \|P_B c_x\|; \kappa_y, \tilde{\nu}) + \hat{\mathcal{J}}(\cos(\Delta_\theta), \|P_A c_y\|; \kappa_x, \tilde{\nu}) \tag{185}$$
$$\geq \hat{\mathcal{J}}(\cos(\phi), \cos(\phi); \kappa_y, \tilde{\nu}) + \hat{\mathcal{J}}(\cos(\phi), \cos(\phi); \kappa_x, \tilde{\nu}).$$

where equality is attained if and only if there exists a configuration of $(X, Y)$ such that:

(A4) $c_x \perp \mathbb{C}$ and $c_y \perp \mathbb{C}$.

Since Lemma 11 shows that $\hat{\mathcal{J}}(\cos(\phi), \cos(\phi); \kappa, \tilde{\nu})$ is a strictly decreasing function of $\cos(\phi)$, we have:

$$\lim_{N \to \infty} \mathcal{L}_{\text{MCL}}^c - 2\log(N) = \hat{\mathcal{J}}(\cos(\Delta_\theta), \|P_B c_x\|; \kappa_y, \tilde{\nu}) + \hat{\mathcal{J}}(\cos(\Delta_\theta), \|P_A c_y\|; \kappa_x, \tilde{\nu})$$
$$\geq \hat{\mathcal{J}}(\cos(\phi), \cos(\phi); \kappa_y, \tilde{\nu}) + \hat{\mathcal{J}}(\cos(\phi), \cos(\phi); \kappa_x, \tilde{\nu}) \tag{186}$$
$$\geq \hat{\mathcal{J}}(\cos(\phi_{\min}), \cos(\phi_{\min}); \kappa_y, \tilde{\nu}) + \hat{\mathcal{J}}(\cos(\phi_{\min}), \cos(\phi_{\min}); \kappa_x, \tilde{\nu}),$$

where equality is attained if and only if there exists a configuration of $(X, Y)$ such that:

(A5) $\Delta_\theta = \cos^{-1}(c_x \cdot c_y) = \phi_{\min}$.

Replacing $\hat{\mathcal{J}}(w, t; \kappa, \nu)$ with $\tilde{\mathcal{J}}(w, w, t; \kappa, \nu)$, we conclude that:

$$
\begin{aligned}
&\lim_{N \to \infty} \mathcal{L}^c_{\mathrm{MCL}} - 2\log(N) \\
&= \tilde{\mathcal{J}}(\cos(\Delta_\theta), \cos(\Delta_\theta), \|P_B c_x\|; \kappa_y, \tilde{\nu}) + \tilde{\mathcal{J}}(\cos(\Delta_\theta), \cos(\Delta_\theta), \|P_A c_y\|; \kappa_x, \tilde{\nu}) \\
&\geq \tilde{\mathcal{J}}(\cos(\Delta_\theta), \cos(\Delta_\theta), \cos(\Delta_\theta); \kappa_y, \tilde{\nu}) + \tilde{\mathcal{J}}(\cos(\Delta_\theta), \cos(\Delta_\theta), \cos(\Delta_\theta); \kappa_x, \tilde{\nu}) \\
&\geq \tilde{\mathcal{J}}(\cos(\phi_{\min}), \cos(\phi_{\min}), \cos(\phi_{\min}); \kappa_y, \tilde{\nu}) + \tilde{\mathcal{J}}(\cos(\phi_{\min}), \cos(\phi_{\min}), \cos(\phi_{\min}); \kappa_x, \tilde{\nu}),
\end{aligned}
\tag{187}
$$

where equality is attained if and only if there exists a configuration of $(X, Y)$ such that:

(A4) $c_x \perp \mathbb{C}$ and $c_y \perp \mathbb{C}$.

(A5) $\Delta_\theta = \cos^{-1}(c_x \cdot c_y) = \phi_{\min}$.

$\square$

### K.3.2. AUXILIARY THEOREMS PART 3

In this subsection, we provide details and proofs of the auxiliary theorems (Theorem S5 and Theorem S6) that support the proof of Theorem 3.

**Theorem S5.** Let $(X, Y)$ be an $N$-pair configuration, where $X = (x_1, \ldots, x_N) \in (\mathbb{S}_X \setminus \mathbb{C})^N$ are $iid$ samples from $\mu_x = \mathrm{vMF}(c_x, \kappa_x)$, and $Y = (y_1, \ldots, y_N) \in (\mathbb{S}_Y \setminus \mathbb{C})^N$ are $iid$ samples from $\mu_y = \mathrm{vMF}(c_y, \kappa_y)$. Let $\tilde{\nu} = (h-1)/2 - 1$ and $\kappa_y > 0$.

$\forall x_i \in X$, denote $w_i = x_i \cdot y_i$ and $w_{x_i, c_y} = x_i \cdot c_y$. It holds that:

$$
\begin{aligned}
\lim_{N \to \infty} \mathcal{L}_{\mathcal{X} \to \mathcal{Y}}(x_i; Y) - \log(N) &= \lim_{N \to \infty} -\log \frac{\exp(x_i \cdot y_i / \tau)}{\sum_{j=1}^{N} \exp(x_i \cdot y_j / \tau)} - \log(N) \\
&= -\frac{w_i}{\tau} + \log\left( \frac{I_{\tilde{\nu}}\left( \tilde{M}_{\kappa_y}\left( w_{x_i, c_y}, \|P_B x_i\| \right) \right)}{\tilde{M}_{\kappa_y}\left( w_{x_i, c_y}, \|P_B x_i\| \right)^{\tilde{\nu}}} \right) - \log\left( \frac{I_{\tilde{\nu}}(\kappa_y)}{\kappa_y^{\tilde{\nu}}} \right) \\
&= \tilde{\mathcal{J}}\left( w_i, w_{x_i, c_y}, \|P_B x_i\|; \kappa, \tilde{\nu} \right),
\end{aligned}
\tag{188}
$$

where $\forall \kappa, \tau > 0$, $\tilde{M}_\kappa(\cdot, \cdot) : [-1, 1] \times [0, 1] \to \mathbb{R}_0^+$ is defined as:

$$
\tilde{M}_\kappa(w, t) = \sqrt{\kappa^2 + \frac{2\kappa w}{\tau} + \frac{t^2}{\tau^2}},
\tag{189}
$$

and $I_\nu$ is the modified Bessel function of the first kind of order $\nu$, which is defined as:

$$
I_\nu(m) = \sum_{k=0}^{\infty} \frac{1}{k! \Gamma(\nu + k + 1)} \left( \frac{m}{2} \right)^{2k + \nu}.
\tag{190}
$$

Suppose there exists an index $i = c$ such that $x_c = c_x$, $y_c = c_y$. Denote $w_c = c_x \cdot c_y$. It holds that:

$$
\begin{aligned}
\lim_{N \to \infty} \mathcal{L}_{\mathcal{X} \to \mathcal{Y}}(c_x; Y) - \log(N) &= -\frac{w_c}{\tau} + \log\left( \frac{I_{\tilde{\nu}}\left( \tilde{M}_{\kappa_y}\left( w_c, \|P_B c_x\| \right) \right)}{\tilde{M}_{\kappa_y}\left( w_c, \|P_B c_x\| \right)^{\tilde{\nu}}} \right) - \log\left( \frac{I_{\tilde{\nu}}(\kappa_y)}{\kappa_y^{\tilde{\nu}}} \right) \\
&= \hat{\mathcal{J}}(w_c, \|P_B c_x\|; \kappa_y, \tilde{\nu}) = \tilde{\mathcal{J}}\left( w_c, w_c, \|P_B x_i\|; \kappa, \tilde{\nu} \right).
\end{aligned}
\tag{191}
$$

*Proof.* **Step 1**: We start the proof by find the convergent function of $\mathcal{L}_{\mathcal{X}\to\mathcal{Y}}(x_i; Y)$ as $N \to \infty$. Same with Eq. (31) of Theorem S1, $\forall x_i \in X$, the $\mathcal{X}$-to-$\mathcal{Y}$ alignment of $x_i$ can be rewritten as:

$$
\begin{aligned}
\mathcal{L}_{\mathcal{X}\to\mathcal{Y}}(x_i; Y) &= -\log \frac{\exp(x_i \cdot y_i / \tau)}{\sum_j \exp(x_i \cdot y_j / \tau)} \\
&= -\frac{x_i \cdot y_i}{\tau} + \log\left(N \frac{1}{N}\sum_{j=1}^{N} \exp\left(\frac{x_i \cdot y_j}{\tau}\right)\right) \\
&= -\frac{x_i \cdot y_i}{\tau} + \log\left(\frac{1}{N}\sum_{j=1}^{N} \exp\left(\frac{x_i \cdot y_j}{\tau}\right)\right) + \log(N).
\end{aligned}
\tag{192}
$$

Lemma 2 shows that:

$$
\lim_{N\to\infty} \log\left(\frac{1}{N}\sum_{j=1}^{N} \exp\left(\frac{x_i \cdot y_j}{\tau}\right)\right) = \log\left(\mathbb{E}_{y\sim\mu_y}\left[\exp\left(\frac{x_i \cdot y}{\tau}\right)\right]\right).
\tag{193}
$$

According to the moment-generating function of the vMF distribution:

$$
\mathbb{E}_{y\sim\mu_y}\left[\exp\left(\frac{x_i \cdot y}{\tau}\right)\right] = \mathbb{E}_{y\sim\mu_y}\left[\exp\left(\frac{x_i}{\tau} \cdot y\right)\right] = \frac{I_{\tilde{\nu}}(\tilde{\kappa}'_y)}{I_{\tilde{\nu}}(\kappa_y)}\left(\frac{\kappa_y}{\tilde{\kappa}'_y}\right)^{\tilde{\nu}},
$$
$$
\text{where } \tilde{\kappa}'_y = \|\kappa_y c_y + \frac{P_B x_i}{\tau}\|_2.
\tag{194}
$$

Then we have:

$$
\lim_{N\to\infty} \mathcal{L}_{\mathcal{X}\to\mathcal{Y}}(x_i; Y) - \log(N) = -\frac{x_i \cdot y_i}{\tau} + \log\left(\frac{I_{\tilde{\nu}}(\tilde{\kappa}'_y)}{\tilde{\kappa}'^{\tilde{\nu}}_y}\right) - \log\left(\frac{I_{\tilde{\nu}}(\kappa_y)}{\kappa_y^{\tilde{\nu}}}\right).
\tag{195}
$$

**Step 2**: we will transform $\mathcal{L}_{\mathcal{X}\to\mathcal{Y}}$ from a function of vectors to a function of angles between vectors.

Without loss of generality, we assume the coordinate of $c_y$ as

$$
c_y = (1, 0, \cdots, 0),
\tag{196}
$$

the hyperplane $\mathbb{B}$ as:

$$
\mathbb{B} = \left\{x \in \mathbb{R}^h : n_A \cdot x = 0\right\}, \quad \text{where } n_B = (0, 0, \cdots, 1).
\tag{197}
$$

Let $\hat{x}_i = P_B x_i$, then we have:

$$
\cos\left(\theta_{x_i, c_y}\right) = x_i \cdot c_y = P_B x_i \cdot c_y = \hat{x}_i \cdot c_y.
\tag{198}
$$

Define:

$$
\cos\left(\hat{\theta}_{x_i, c_y}\right) = \frac{\hat{x}_i}{\|\hat{x}_i\|} \cdot c_y = \frac{P_B x_i}{\|P_B x_i\|} \cdot c_y,
\tag{199}
$$

then we have:

$$\|P_B x_i\| \cos\left(\hat{\theta}_{x_i,c_y}\right) = P_B x_i \cdot c_y = \cos\left(\theta_{x_i,c_y}\right). \tag{200}$$

And $\hat{x}_i$ can be represented as:

$$\begin{aligned}
\hat{x}_i &= \|P_B x_i\| \left(\cos\left(\hat{\theta}_{x_i,c_y}\right), u \sin\left(\hat{\theta}_{x_i,c_y}\right)\right) \\
&= \|P_B x_i\| \left(\cos\left(\hat{\theta}_{x_i,c_y}\right), u_2 \sin\left(\hat{\theta}_{x_i,c_y}\right), u_3 \sin\left(\hat{\theta}_{x_i,c_y}\right), \ldots, u_{h-1} \sin\left(\hat{\theta}_{x_i,c_y}\right), 0\right),
\end{aligned} \tag{201}$$

where $u = (0, u_2, u_3, \ldots, u_{h-1}, 0) \cong \mathbb{S}^{h-3} \in \mathbb{S}^{h-1}$ is a unit vector orthogonal to the first and the last axes with:

$$\|u\| = 0 + u_2^2 + u_3^2 + \cdots + u_{h-1}^2 + 0 = 1. \tag{202}$$

According to Eq. (196), Eq. (201) and Eq. (202), $\tilde{\kappa}'_y$ (in Eq. (194)) can re-rewritten as:

$$\begin{aligned}
\tilde{\kappa}'_y &= \left\|\kappa_y c_y + \frac{x_i}{\tau}\right\|_2 \\
&= \sqrt{\left(\kappa_y + \frac{\|P_B x_i\| \cos\left(\hat{\theta}_{x_i,c_y}\right)}{\tau}\right)^2 + \sum_{i=2}^{h-1}\left(\frac{\|P_B x_i\| \sin\left(\hat{\theta}_{x_i,c_y}\right) u_i}{\tau}\right)^2} \\
&= \sqrt{\left(\kappa_y + \frac{\|P_B x_i\| \cos\left(\hat{\theta}_{x_i,c_y}\right)}{\tau}\right)^2 + \frac{\|P_B x_i\|^2 \sin^2\left(\hat{\theta}_{x_i,c_y}\right)}{\tau^2}} \\
&= \sqrt{\kappa_y^2 + \frac{2\kappa_y \|P_B x_i\| \cos\left(\hat{\theta}_{x_i,c_y}\right)}{\tau} + \frac{\|P_B x_i\|^2}{\tau^2}} \\
&= \sqrt{\kappa_y^2 + \frac{2\kappa_y \cos\left(\theta_{x_i,c_y}\right)}{\tau} + \frac{\|P_B x_i\|^2}{\tau^2}} \\
&= \tilde{M}_{\kappa_y}\left(\cos\left(\theta_{x_i,c_y}\right), \|P_B x_i\|\right).
\end{aligned} \tag{203}$$

Consider that $w_i = x_i \cdot y_i$, $w_{x_i,c_y} = \cos\left(\theta_{x_i,c_y}\right) = x_i \cdot c_y$, putting Eq. (195) and Eq. (203) together, we have:

$$\begin{aligned}
\lim_{N\to\infty} \mathcal{L}_{\mathcal{X}\to\mathcal{Y}}(x_i; Y) - \log(N) &= -\frac{x_i \cdot y_i}{\tau} + \log\left(\frac{I_{\tilde{\nu}}\left(\tilde{\kappa}'_y\right)}{\tilde{\kappa}'^{\tilde{\nu}}_y}\right) - \log\left(\frac{I_{\tilde{\nu}}\left(\kappa_y\right)}{\kappa_y^{\tilde{\nu}}}\right) \\
&= -\frac{w_i}{\tau} + \log\left(\frac{I_{\tilde{\nu}}\left(\tilde{M}_{\kappa_y}\left(w_{x_i,c_y}, \|P_B x_i\|\right)\right)}{\tilde{M}_{\kappa_y}\left(w_{x_i,c_y}, \|P_B x_i\|\right)^{\tilde{\nu}}}\right) - \log\left(\frac{I_{\tilde{\nu}}\left(\kappa_y\right)}{\kappa_y^{\tilde{\nu}}}\right) \\
&= \tilde{\mathcal{J}}\left(w_i, w_{x_i,c_y}, \|P_B x_i\|; \kappa, \tilde{\nu}\right).
\end{aligned} \tag{204}$$

When there exists a data pair $i = c$ such that $x_c = c_x$, $y_c = c_y$, $w_i = w_{x_i,c_y} = w_c$, then we have:

$$\begin{aligned}
\lim_{N\to\infty} \mathcal{L}_{\mathcal{X}\to\mathcal{Y}}(c_x; Y) - \log(N) &= -\frac{w_c}{\tau} + \log\left(\frac{I_{\tilde{\nu}}\left(\tilde{M}_{\kappa_y}\left(w_c, \|P_B c_x\|\right)\right)}{\tilde{M}_{\kappa_y}\left(w_c, \|P_B c_x\|\right)^{\tilde{\nu}}}\right) - \log\left(\frac{I_{\tilde{\nu}}\left(\kappa_y\right)}{\kappa_y^{\tilde{\nu}}}\right) \\
&= \hat{\mathcal{J}}\left(w_c, \|P_B c_x\|; \kappa_y, \tilde{\nu}\right) = \tilde{\mathcal{J}}\left(w_c, w_c, \|P_B x_i\|; \kappa, \tilde{\nu}\right).
\end{aligned} \tag{205}$$

$\square$

**Theorem S6.** Let $(X, Y)$ be an $N$-pair configuration, where $X = (x_1, \ldots, x_N) \in (\mathbb{S}_X \setminus \mathbb{C})^N$ are *iid* samples from $\mu_x = \text{vMF}(c_x, \kappa_x)$, and $Y = (y_1, \ldots, y_N) \in (\mathbb{S}_Y \setminus \mathbb{C})^N$ are *iid* samples from $\mu_y = \text{vMF}(c_y, \kappa_y)$. Let $\tilde{\nu} = (h-1)/2 - 1$. Suppose there exists an index $i = c$ such that $x_c = c_x$, $y_c = c_y$. Denote $\Delta_\theta = \cos^{-1}(c_x \cdot c_y)$ and assume that $c_x, c_y \notin \mathbb{C}$ with $c_x \cdot c_y > 0$. For any fixed $\kappa_x, \kappa_y > 0$ and $\forall \phi \in [0, \frac{\pi}{2}]$, it holds that:

$$\lim_{N \to \infty} \mathcal{L}_{\mathcal{X} \to \mathcal{Y}}(c_x; Y) - \log(N) = \hat{\mathcal{J}}(w_c, \|P_B c_x\|; \kappa_y, \tilde{\nu}) \geq \hat{\mathcal{J}}(\|P_A c_y\|, \cos(\phi); \kappa_y, \tilde{\nu}), \tag{206}$$

where equality is attained if and only if there exists a configuration of $(X, Y)$ such that:

(B4) $c_x \perp \mathbb{C}$.

(B5) $c_x = \frac{P_A c_y}{\|P_A c_y\|}$.

*Proof.* **Step 1**: Similarly to the proof of Theorem S4 in Sec. K.2.2, we start the proof by finding the convergent function of $\mathcal{L}_{\mathcal{X} \to \mathcal{Y}}(c_x; Y)$ as $N \to \infty$. Denote $w_c = c_x \cdot c_y$. $\forall \kappa_y > 0$, as proven in Theorem S5:

$$\begin{aligned}
\lim_{N \to \infty} \mathcal{L}_{\mathcal{X} \to \mathcal{Y}}(c_x; Y) - \log(N) &= \lim_{N \to \infty} -\log \frac{\exp(c_x \cdot c_y / \tau)}{\sum_{j=1}^N \exp(c_x \cdot y_j / \tau)} - \log(N) \\
&= -\frac{w_c}{\tau} + \log \left( \frac{I_{\tilde{\nu}}\left( \tilde{M}_{\kappa_y}(w_c, \|P_B c_x\|) \right)}{\tilde{M}_{\kappa_y}(w_c, \|P_B c_x\|)^{\tilde{\nu}}} \right) - \log \left( \frac{I_{\tilde{\nu}}(\kappa_y)}{\kappa_y^{\tilde{\nu}}} \right) \\
&= \hat{\mathcal{J}}(w_c, \|P_B c_x\|; \kappa_y, \tilde{\nu}),
\end{aligned} \tag{207}$$

where $\forall \kappa, \tau > 0$, $\hat{\mathcal{J}}(\cdot, \cdot; \kappa, \tilde{\nu})$ is a function on $[-1, 1] \times [0, 1]$ and $\tilde{M}_\kappa(\cdot, \cdot) : [-1, 1] \times [0, 1] \to \mathbb{R}_0^+$ is defined as:

$$\tilde{M}_\kappa(w, t) = \sqrt{\kappa^2 + \frac{2\kappa w}{\tau} + \frac{t^2}{\tau^2}}. \tag{208}$$

and $I_\nu$ is the modified Bessel function of the first kind of order $\nu$, which is defined as:

$$I_\nu(m) = \sum_{k=0}^\infty \frac{1}{k! \Gamma(\nu + k + 1)} \left( \frac{m}{2} \right)^{2k+\nu}. \tag{209}$$

**Step 2**: Next, we find the minimal value and the optimal condition of convergent function.

$\forall c_x \in \mathbb{S}_X, \phi \in [0, \frac{\pi}{2}]$ it holds that:

$$0 \leq \cos(\phi) \leq \|P_B c_x\| \leq 1. \tag{210}$$

As shown in Lemma 10, $\forall w_c \in [0, 1]$, $\hat{\mathcal{J}}(w = w_c, t; \kappa_y, \tilde{\nu})$ is a strictly increasing function of $t$ on $(0, 1)$. Therefore, it holds that:

$$\hat{\mathcal{J}}(w_c, \cos(\phi); \kappa_y, \tilde{\nu}) \leq \hat{\mathcal{J}}(w_c, \|P_B c_x\|; \kappa_y, \tilde{\nu}) \leq \hat{\mathcal{J}}(w_c, 1; \kappa_y, \tilde{\nu}). \tag{211}$$

where equality in the above chain holds if and only if the following conditions are satisfied:

(i) The first inequality becomes equality: $c_x \perp \mathbb{C}$.

(ii) The second inequality becomes equality: $c_x \in \mathbb{C}$.

According to Lemma 5 (set $s = \cos(\phi)$), $\hat{\mathcal{J}}(w_c, \cos(\phi); \kappa_y, \tilde{\nu})$ is a strictly decreasing function on $w_c$ when $w_c \geq 0$. Also, Lemma 12 shows that:

$$-\|P_A c_y\| \leq w_c \leq \|P_A c_y\|, \tag{212}$$

where

$$0 \leq \cos(\phi) < \|P_A c_y\| < 1. \tag{213}$$

Therefore, when $0 \leq w_c \leq \|P_A c_y\|$, it holds that:

$$\hat{\mathcal{J}}(w_c, \cos(\phi); \kappa_y, \tilde{\nu}) \geq \hat{\mathcal{J}}(\|P_A c_y\|, \cos(\phi); \kappa_y, \tilde{\nu}), \tag{214}$$

where equality is attained if and only if there exists a configuration of $(X, Y)$ such that:

(iii) $c_x = \frac{P_A c_y}{\|P_A c_y\|}$.

Combining Eq. (207), Eq. (211) and Eq. (214), we conclude:

$$\lim_{N \to \infty} \mathcal{L}_{\mathcal{X} \to \mathcal{Y}}(c_x; Y) - \log(N) = \hat{\mathcal{J}}(w_c, \|P_B c_x\|; \kappa_y, \tilde{\nu}) \geq \hat{\mathcal{J}}(\|P_A c_y\|, \cos(\phi); \kappa_y, \tilde{\nu}), \tag{215}$$

and equality is attained if and only if there exists a configuration of $(X, Y)$ such that:

(B4) $c_x \perp \mathbb{C}$.

(B5) $c_x = \frac{P_A c_y}{\|P_A c_y\|}$.

$\square$

### K.3.3. TECHNICAL LEMMAS PART 3

In this subsection, we provide details and proofs of technical lemmas (Lemma 10, Lemma 11, Lemma 12 and Lemma 13) that support the proof of Theorem 3, Theorem S5 and Theorem S6.

**Lemma 10.** $\forall \kappa, \nu, \tau > 0$ and $w_c \in [0, 1]$, a function $\hat{\mathcal{J}}_{t=s}(\cdot; \kappa, \nu) : (0, 1] \to \mathbb{R}$ is defined as:

$$\hat{\mathcal{J}}_{w=w_s}(t; \kappa, \nu) = -\frac{w_s}{\tau} + \log\left(\frac{I_\nu\left(\tilde{M}_{w=w_s}(t)\right)}{\tilde{M}_\kappa(t)^\nu}\right) - \log\left(\frac{I_\nu(\kappa)}{\kappa^\nu}\right)$$
$$= \hat{\mathcal{J}}(w = w_s, t; \kappa, \nu) = \tilde{J}(w = w_s, w = w_s, t; \kappa, \nu), \tag{216}$$

where $\tilde{M}_\kappa(\cdot) : (0, 1] \to \mathbb{R}^+$ is defined as:

$$\tilde{M}_{w=w_s}(t) = \sqrt{\kappa^2 + \frac{2\kappa w_s}{\tau} + \frac{t^2}{\tau^2}} = \tilde{M}_\kappa(w = w_s, t), \tag{217}$$

and $I_\nu$ is the modified Bessel function of the first kind of order $\nu$, which is defined as:

$$I_\nu(m) = \sum_{k=0}^{\infty} \frac{1}{k!\Gamma(\nu + k + 1)} \left(\frac{m}{2}\right)^{2k+\nu}. \tag{218}$$

It holds that, for any fixed $w_s$, $\hat{\mathcal{J}}_{w=w_s}(\cdot)$ is a strictly increasing function on $(0, 1]$.

*Proof.* Let us first decompose the function $\mathcal{J}$. Denote a constant and a function $C_1$ and $G_2(t)$ as:

$$
\begin{aligned}
C_1 &= -\frac{w_s}{\tau}, \\
G_3(m) &= \log\left(I_\nu(m)\right) - \nu\log(m), \\
G_2(t) &= G_3\left(\tilde{M}_{w=w_s}(t)\right) \\
&= \log\left(I_\nu\left(\tilde{M}_{w=w_s}(t)\right)\right) - \nu\log\left(\tilde{M}_{w=w_s}(t)\right).
\end{aligned}
\tag{219}
$$

Denote the function $G(t)$ and the constant $C$ as:

$$
\begin{aligned}
G(t) &= C_1 + G_2(t), \\
C &= -\log\left(\frac{I_\nu(\kappa)}{\kappa^\nu}\right).
\end{aligned}
\tag{220}
$$

Then the function $\hat{\mathcal{J}}_{w=w_s}$ can be written as:

$$
\begin{aligned}
\hat{\mathcal{J}}_{w=w_s}(t;\kappa,\nu) &= -\frac{w_s}{\tau} + \log\left(\frac{I_\nu\left(\tilde{M}_{w=w_s}(t)\right)}{\tilde{M}_{w=w_s}(t)^\nu}\right) - \log\left(\frac{I_\nu(\kappa)}{\kappa^\nu}\right) \\
&= G(t) + C.
\end{aligned}
\tag{221}
$$

Now, we investigate derivatives of $\hat{\mathcal{J}}_{w=w_s}$.

According to Lemma 7, the first derivative of $G_3(m)$ is:

$$
G_3'(m) = \frac{I_{\nu+1}(m)}{I_\nu(m)} \in (0,1).
\tag{222}
$$

The derivative of $\tilde{M}_{w=w_s}$ is:

$$
\begin{aligned}
\tilde{M}_{w=w_s}'(t) &= \frac{d}{dt}\left(\kappa^2 + \frac{2\kappa w_s}{\tau} + \frac{t^2}{\tau^2}\right)^{1/2} \\
&= \frac{1}{2}\left(\kappa^2 + \frac{2\kappa w_s}{\tau} + \frac{t^2}{\tau^2}\right)^{-1/2} \cdot 2\frac{t}{\tau^2} \\
&= \frac{t}{\tau^2}\frac{1}{\tilde{M}_{w=w_s}(t)} \\
&> 0.
\end{aligned}
\tag{223}
$$

Then, the first derivative of $G_2$ is:

$$
\begin{aligned}
G_2'(t) &= G_3'\left(\tilde{M}_{w=w_s}(t)\right)\tilde{M}_{w=w_s}'(t) \\
&= \frac{I_{\nu+1}\left(\tilde{M}_{w=w_s}(t)\right)}{I_\nu\left(\tilde{M}_{w=w_s}(t)\right)}\tilde{M}_{w=w_s}'(t) \\
&= \frac{t}{\tau^2}\frac{1}{\tilde{M}_{w=w_s}(t)}\frac{I_{\nu+1}\left(\tilde{M}_{w=w_s}(t)\right)}{I_\nu\left(\tilde{M}_{w=w_s}(t)\right)} \\
&> 0.
\end{aligned}
\tag{224}
$$

Therefore, we have:

$$
\begin{aligned}
\hat{\mathcal{J}}'_{w=w_s}(t;\kappa,\nu) &= G'(t) = G'_2(t) \\
&= \frac{t}{\tau^2} \frac{1}{\tilde{M}_{w=w_s}(t)} \frac{I_{\nu+1}\left(\tilde{M}_{w=w_s}(t)\right)}{I_\nu\left(\tilde{M}_{w=w_s}(t)\right)} \\
&> 0.
\end{aligned}
\tag{225}
$$

So we can conclude that, for any fixed $w_s$, $\hat{\mathcal{J}}_{w=w_s}(\cdot)$ is a strictly increasing function on $(0,1]$.

$\square$

**Lemma 11.** $\forall \kappa, \nu, \tau > 0$, a function $\hat{\mathcal{J}}(\cdot;\kappa,\nu) : [-1,1] \to \mathbb{R}$ is defined as:

$$
\begin{aligned}
\hat{\mathcal{J}}_{t=w}(w;\kappa,\nu) &= -\frac{w}{\tau} + \log\left(\frac{I_\nu\left(\tilde{M}_{t=w}(w)\right)}{\tilde{M}_{t=w}(w)^\nu}\right) - \log\left(\frac{I_\nu(\kappa)}{\kappa^\nu}\right) \\
&= \hat{\mathcal{J}}(w, t=w;\kappa,\nu) = \tilde{J}(w, w, t=w;\kappa,\nu),
\end{aligned}
\tag{226}
$$

where $\tilde{M}_{t=w}(\cdot) : [-1,1] \to \mathbb{R}^+$ is defined as:

$$
\tilde{M}_{t=w}(w) = \sqrt{\kappa^2 + \frac{2\kappa w}{\tau} + \frac{w^2}{\tau^2}} = |\kappa + \frac{w}{\tau}| = \tilde{M}_\kappa(w, t=w),
\tag{227}
$$

and $I_\nu$ is the modified Bessel function of the first kind of order $\nu$, which is defined as:

$$
I_\nu(m) = \sum_{k=0}^\infty \frac{1}{k!\Gamma(\nu+k+1)} \left(\frac{m}{2}\right)^{2k+\nu}.
\tag{228}
$$

It holds that $\hat{\mathcal{J}}_{t=w}(\cdot)$ is a strictly decreasing function when $w \in [0,1]$.

*Proof.* Let us first decompose the function $\hat{\mathcal{J}}_{t=w}$. Denote the functions $G_1(w)$ and $G_2(w)$ as:

$$
\begin{aligned}
G_1(w) &= -\frac{w}{\tau}, \\
G_3(m) &= \log(I_\nu(m)) - \nu\log(m), \\
G_2(w) &= G_3\left(\tilde{M}_{t=w}(w)\right) \\
&= \log\left(I_\nu\left(\tilde{M}_{t=w}(w)\right)\right) - \nu\log\left(\tilde{M}_{t=w}(w)\right).
\end{aligned}
\tag{229}
$$

Denote the function $G(w)$ and the constant $C$ as:

$$
\begin{aligned}
G(w) &= G_1(w) + G_2(w), \\
C &= -\log\left(\frac{I_\nu(\kappa)}{\kappa^\nu}\right).
\end{aligned}
\tag{230}
$$

Then the function $\hat{\mathcal{J}}_{t=w}$ can be written as:

$$\hat{\mathcal{J}}_{t=w}(w; \kappa, \nu) = -\frac{w}{\tau} + \log\left(\frac{I_\nu\left(\tilde{M}_{t=w}(w)\right)}{\tilde{M}_{t=w}(w)^\nu}\right) - \log\left(\frac{I_\nu(\kappa)}{\kappa^\nu}\right) \tag{231}$$

$$= G(w) + C.$$

Now, we investigate derivatives of $\hat{\mathcal{J}}_{t=w}$.

The first derivative of $G_1$ is:

$$G_1'(w) = -\frac{1}{\tau} < 0. \tag{232}$$

According to Lemma 7, the first derivative of $G_3(m)$ is:

$$G_3'(m) = \frac{I_{\nu+1}(m)}{I_\nu(m)} \in (0, 1). \tag{233}$$

When $w \in [0, 1]$, the derivative of $\tilde{M}_{t=w}$ is:

$$\tilde{M}_{t=w}'(w) = \frac{1}{\tau}. \tag{234}$$

Then, the first derivative of $G_2$ is:

$$
\begin{aligned}
G_2'(w) &= G_3'\left(\tilde{M}_{t=w}(w)\right) \tilde{M}_{t=w}'(w) \\
&= \frac{I_{\nu+1}\left(\tilde{M}_{t=w}(w)\right)}{I_\nu\left(\tilde{M}_{t=w}(w)\right)} \tilde{M}_{t=w}'(w) \\
&= \frac{1}{\tau} \frac{I_{\nu+1}\left(\tilde{M}_{t=w}(w)\right)}{I_\nu\left(\tilde{M}_{t=w}(w)\right)}.
\end{aligned}
\tag{235}
$$

Combining Eq. (232), Eq. (233) and Eq. (235), we have:

$$
\begin{aligned}
\hat{\mathcal{J}}_{t=w}'(w; \kappa, \nu) &= G'(w) \\
&= -\frac{1}{\tau} + \frac{1}{\tau} \frac{I_{\nu+1}\left(\tilde{M}_{t=w}(w)\right)}{I_\nu\left(\tilde{M}_{t=w}(w)\right)} = \frac{1}{\tau}\left(-1 + \frac{I_{\nu+1}\left(\tilde{M}_{t=w}(w)\right)}{I_\nu\left(\tilde{M}_{t=w}(w)\right)}\right) \\
&< 0.
\end{aligned}
\tag{236}
$$

So we can conclude that $\hat{\mathcal{J}}_{t=w}(\cdot)$ is a strictly decreasing function on $[0, 1]$.

$\square$

**Lemma 12.** *Let $h \geq 3$ and $\mathbb{A}, \mathbb{B} \in \mathbb{R}^h$ be two distinct $(h-1)$-dimensional linear subspaces, with $n_A, n_B$ being normal vectors and $P_A, P_B$ being the orthogonal projectors on $\mathbb{A}$ and $\mathbb{B}$, respectively. Denote $\phi = \cos^{-1}\left(\frac{n_A \cdot n_B}{\|n_A\| \cdot \|n_B\|}\right) \in \left(0, \frac{\pi}{2}\right)$ as the angle between $\mathbb{A}$ and $\mathbb{B}$. Let $\mathbb{C} = \mathbb{A} \cap \mathbb{B}$ be an $(h-2)$-dimensional linear subspaces. For each fixed $x \in \mathbb{S}_X = \mathbb{A} \cap \mathbb{S}^{h-1}$, $\forall y \in \mathbb{S}_Y = \mathbb{B} \cap \mathbb{S}^{h-1}$, set $w = x \cdot y$, it holds that:*

$$-\|P_B \cdot x\| \leq w \leq \|P_B \cdot x\|, \tag{237}$$

*and equalities (extreme values) are attained if and only if the following conditions hold:*

*(C1)* $w = \|P_B \cdot x\| \Leftrightarrow y = \frac{P_B \cdot x}{\|P_B \cdot x\|}.$

*(C2)* $w = -\|P_B \cdot x\| \Leftrightarrow y = -\frac{P_B \cdot x}{\|P_B \cdot x\|}.$

*Proof.* **Step 1:** First, let us decompose the embedding space. Define two vectors $e_A$ and $e_B$ such that:

$$
\begin{aligned}
e_A \in \mathbb{S}_X, \quad &\text{and} \quad e_A \perp \mathbb{C}, \\
e_B \in \mathbb{S}_Y, \quad &\text{and} \quad e_B \perp \mathbb{C}.
\end{aligned}
\tag{238}
$$

Let $\mathbb{C}^\perp$ be the 2-dimensional orthogonal complement of $C$, and $\mathbb{C}^\perp$ satisfies:

$$
\begin{aligned}
\mathbb{C}^\perp &= \text{span}\left\{e_A\right\} \oplus \text{span}\left\{e_B\right\}, \\
\mathbb{R}^h &= \mathbb{C} \oplus \mathbb{C}^\perp.
\end{aligned}
\tag{239}
$$

Since $n_A, n_B \in \mathbb{C}^\perp$, $n_A \perp e_A$ and $n_B \perp e_B$, we have:

$$
\langle e_A, e_B \rangle = \pm \langle n_A, n_B \rangle,
\tag{240}
$$

and we choose a pair of $e_A$ and $e_B$ such that:

$$
\langle e_A, e_B \rangle = \langle n_A, n_B \rangle = \cos(\phi) \in (0, 1).
\tag{241}
$$

Therefore, $\forall x \in \mathbb{S}_X = \mathbb{A} \cap \mathbb{S}^{h-1}$ and $\forall y \in \mathbb{S}_Y = \mathbb{B} \cap \mathbb{S}^{h-1}$, $\exists u_A, u_B \in \mathbb{C} \cap \mathbb{S}^{h-1}$, such that $\cos(\theta_A) = x \cdot e_A$ and $\cos(\theta_B) = y \cdot e_B$. And then $x$ and $y$ can be represented as:

$$
\begin{aligned}
x &= \cos(\theta_A) e_A + \sin(\theta_A) u_A, \\
y &= \cos(\theta_B) e_B + \sin(\theta_B) u_B.
\end{aligned}
\tag{242}
$$

Using orthogonality, we have:

$$
\begin{aligned}
P_B \cdot e_A &= \langle e_A, e_B \rangle e_B = \cos(\phi) e_B, \\
P_B \cdot u_A &= u_A,
\end{aligned}
\tag{243}
$$

and

$$
\begin{aligned}
P_A \cdot e_B &= \langle e_A, e_B \rangle e_A = \cos(\phi) e_A, \\
P_A \cdot u_B &= u_B.
\end{aligned}
\tag{244}
$$

Then the projections of $(x_i, y_i)$ are:

$$
\begin{aligned}
P_B \cdot x &= \cos(\theta_A) \cos(\phi) e_B + \sin(\theta_A) u_A, \\
P_A \cdot y &= \cos(\theta_B) \cos(\phi) e_A + \sin(\theta_B) u_B.
\end{aligned}
\tag{245}
$$

**Step 2:** Next, we can investigate the range of $w$.

$$
\begin{aligned}
w &= x \cdot y \\
&= \cos(\theta_A) \cos(\theta_B) e_A e_B + \sin(\theta_A) \sin(\theta_B) u_A u_B \\
&= \cos(\theta_A) \cos(\theta_B) \cos(\phi) + \sin(\theta_A) \sin(\theta_B) u_A u_B.
\end{aligned}
\tag{246}
$$

Since $u_A, u_B \in \mathbb{C}$ and $\|u_A\| = \|u_B\| = 1$, then $\|u_A \cdot u_B\| \le 1$. Denote $f(\cdot)_\pm$ as:

$$f_\pm(\theta_B) = \cos(\theta_A)\cos(\theta_B)\cos(\phi) \pm \sin(\theta_A)\sin(\theta_B), \tag{247}$$

then :

$$f_-(\theta_B) \le w \le f_+(\theta_B). \tag{248}$$

Now, let us check the extreme values of $f_\pm(w)$. First, we find the derivative of $f_\pm(w)$:

$$f'_\pm(\theta_B) = -\cos(\theta_A)\sin(\theta_B)\cos(\phi) \pm \sin(\theta_A)\cos(\theta_B), \tag{249}$$

then:

$$f'_\pm(\theta_B) = 0 \quad \Rightarrow \quad \tan(\theta_B) = \pm \frac{\sin(\theta_A)}{\cos(\theta_A)\cos(\phi)}, \tag{250}$$

and

$$
\begin{aligned}
w &\ge f_-\left(\arctan\left(-\frac{\sin(\theta_A)}{\cos(\theta_A)\cos(\phi)}\right)\right) = -\sqrt{\sin^2(\theta_A) + \cos^2(\theta_A)\cos^2(\phi)}, \\
w &\le f_+\left(\arctan\left(\frac{\sin(\theta_A)}{\cos(\theta_A)\cos(\phi)}\right)\right) = \sqrt{\sin^2(\theta_A) + \cos^2(\theta_A)\cos^2(\phi)}.
\end{aligned}
\tag{251}
$$

Denote:

$$r(x) = \sqrt{\sin^2(\theta_A) + \cos^2(\theta_A)\cos^2(\phi)} \in (0, 1). \tag{252}$$

and therefore:

$$|w| \le r(x) < 1. \tag{253}$$

**Step 3:** Last, we find the optimal condition of $w$. When $\theta_B = \arctan\left(\frac{\sin(\theta_A)}{\cos(\theta_A)\cos(\phi)}\right)$ and $u_A = u_B$, $w$ reaches its maximum. At this time:

$$
\begin{aligned}
\cos(\theta_B) &= \frac{\cos(\theta_A)\cos(\phi)}{r}, \\
\sin(\theta_B) &= \frac{\sin(\theta_A)}{r}.
\end{aligned}
\tag{254}
$$

Plugging Eq. (254) into Eq. (245), we get:

$$
\begin{aligned}
P_B \cdot x &= \cos(\theta_A)\cos(\phi)\, e_B + \sin(\theta_A)u_A \\
&= r\cos(\theta_B)\cos(\phi)\, e_B + r\sin(\theta_B)u_B \\
&= ry,
\end{aligned}
\tag{255}
$$

and

$$\|P_B \cdot x\| = \|ry\| = r. \tag{256}$$

Therefore, $w$ reaches its maximum if and only if the following condition holds:

(C1) $y = \frac{P_B \cdot x}{\|P_B \cdot x\|}$.

When $\theta_B = \arctan\left(-\frac{\sin(\theta_A)}{\cos(\theta_A)\cos(\phi)}\right)$ and $u_A = -u_B$, $w$ reaches its minimum. At this time:

$$\cos(\theta_B) = -\frac{\cos(\theta_A)\cos(\phi)}{r},$$
$$\sin(\theta_B) = \frac{\sin(\theta_A)}{r}. \tag{257}$$

Plugging Eq. (257) into Eq. (245), we get:

$$\begin{aligned} P_B \cdot x &= \cos(\theta_A)\cos(\phi)\, e_B + \sin(\theta_A) u_A \\ &= -r\cos(\theta_B)\cos(\phi)\, e_B - r\sin(\theta_B) u_B \\ &= -ry. \end{aligned} \tag{258}$$

and

$$\|P_B \cdot x\| = \|-ry\| = r. \tag{259}$$

Therefore, $w$ reaches its minimum if and only if the following condition holds:

(C2) $y = -\frac{P_B \cdot x}{\|P_B \cdot x\|}$.

$\square$

**Lemma 13.** *Let $h \geq 3$ and $\mathbb{A}, \mathbb{B} \in \mathbb{R}^h$ be two distinct $(h-1)$-dimensional linear subspaces, with $n_A, n_B$ being normal vectors and $P_A, P_B$ being the orthogonal projectors on $\mathbb{A}$ and $\mathbb{B}$, respectively. Denote $\phi = \cos^{-1}\left(\frac{n_A \cdot n_B}{\|n_A\| \cdot \|n_B\|}\right) \in \left(0, \frac{\pi}{2}\right)$ as the angle between $\mathbb{A}$ and $\mathbb{B}$. Let $\mathbb{C} = \mathbb{A} \cap \mathbb{B}$ be an $(h-2)$-dimensional linear subspaces. For $x \in \mathbb{S}_X = \mathbb{A} \cap \mathbb{S}^{h-1}, y \in \mathbb{S}_Y = \mathbb{B} \cap \mathbb{S}^{h-1}$, the projections of $x$ and $y$ are collinear with the other vector:*

*(i) The orthogonal projection of $x$ on $\mathbb{B}$ is a scalar multiple of $y$:*

$$P_B x = \lambda_x y, \quad \lambda_x \neq 0,$$

*(ii) The orthogonal projection of $y$ on $\mathbb{A}$ is a scalar multiple of $x$:*

$$P_A y = \lambda_y x, \quad \lambda_y \neq 0,$$

*if and only if the following conditions holds:*

*(C3) Either $x \perp \mathbb{C}$ and $y \perp \mathbb{C}$, or $x = \pm y \in \mathbb{C}$.*

*Moreover, in the first case ($x \perp \mathcal{C}$, $y \perp \mathcal{C}$), it holds that:*

$$\langle x, y \rangle = \cos(\phi), \qquad P_B x = (\cos(\phi))\, y, \qquad P_A y = (\cos(\phi))\, x,$$

*while in the second case ($x = \pm y \in \mathcal{C}$), it holds that:*

$$P_B x = x = (\pm 1)\, y, \qquad P_A y = y = (\pm 1)\, x.$$

*Proof.* **Step 1:** First, we need to decompose the embedding space. This step is the same with **Step 1** of Sec. K.3.3. For convenience in reading, we repeat this step here.

Define two vectors $e_A$ and $e_B$ such that:

$$
\begin{aligned}
e_A \in \mathbb{S}_X, \quad \text{and} \quad e_A \perp \mathbb{C}, \\
e_B \in \mathbb{S}_Y, \quad \text{and} \quad e_B \perp \mathbb{C}.
\end{aligned}
\tag{260}
$$

Let $\mathbb{C}^\perp$ be the 2-dimensional orthogonal complement of $C$, and $\mathbb{C}^\perp$ satisfies:

$$
\begin{aligned}
\mathbb{C}^\perp &= \operatorname{span}\{e_A\} \oplus \operatorname{span}\{e_B\}, \\
\mathbb{R}^h &= \mathbb{C} \oplus \mathbb{C}^\perp.
\end{aligned}
\tag{261}
$$

Since $n_A, n_B \in \mathbb{C}^\perp$, $n_A \perp e_A$ and $n_B \perp e_B$, we have:

$$
\langle e_A, e_B \rangle = \pm \langle n_A, n_B \rangle,
\tag{262}
$$

and we choose a pair of $e_A$ and $e_B$ such that:

$$
\langle e_A, e_B \rangle = \langle n_A, n_B \rangle = \cos(\phi) \in (0, 1).
\tag{263}
$$

Therefore, $\forall x \in \mathbb{S}_X = \mathbb{A} \cap \mathbb{S}^{h-1}$ and $\forall y \in \mathbb{S}_Y = \mathbb{B} \cap \mathbb{S}^{h-1}$, $\exists u_A, u_B \in C \cap \mathbb{S}^{h-1}$, such that $\cos(\theta_A) = x \cdot e_A$ and $\cos(\theta_B) = y \cdot e_B$. And then $x$ and $y$ can be represented as:

$$
\begin{aligned}
x &= \cos(\theta_A) e_A + \sin(\theta_A) u_A, \\
y &= \cos(\theta_B) e_B + \sin(\theta_B) u_B.
\end{aligned}
\tag{264}
$$

Using orthogonality, we have:

$$
\begin{aligned}
P_B \cdot e_A &= \langle e_A, e_B \rangle e_B = \cos(\phi) e_B, \\
P_B \cdot u_A &= u_A,
\end{aligned}
\tag{265}
$$

and

$$
\begin{aligned}
P_A \cdot e_B &= \langle e_A, e_B \rangle e_A = \cos(\phi) e_A, \\
P_A \cdot u_B &= u_B.
\end{aligned}
\tag{266}
$$

Then the projections of $(x_i, y_i)$ are:

$$
\begin{aligned}
P_B \cdot x &= \cos(\theta_A) \cos(\phi) e_B + \sin(\theta_A) u_A, \\
P_A \cdot y &= \cos(\theta_B) \cos(\phi) e_A + \sin(\theta_B) u_B.
\end{aligned}
\tag{267}
$$

**Step 2:** $\Rightarrow$ Next, we prove the sufficiency. If conditions (i) and (ii) hold, then:

$$
\begin{aligned}
\cos(\theta_A) \cos(\phi) e_B + \sin(\theta_A) u_A &= \lambda_x \cos(\theta_B) e_B + \lambda_x \sin(\theta_B) u_B, \\
\cos(\theta_B) \cos(\phi) e_A + \sin(\theta_B) u_B &= \lambda_y \cos(\theta_A) e_A + \lambda_y \sin(\theta_A) u_A.
\end{aligned}
\tag{268}
$$

Decompose both equations into $\mathbb{C}$ and $\mathbb{C}^\perp$. In $\mathbb{C}$, we get:

$$\begin{aligned}
\sin(\theta_A)u_A &= \lambda_x \sin(\theta_B)u_B, \\
\sin(\theta_B)u_B &= \lambda_y \sin(\theta_A)u_A.
\end{aligned} \tag{269}$$

and in $\mathbb{C}^\perp$ we get:

$$\begin{aligned}
\cos(\theta_A)\cos(\phi)\,e_B &= \lambda_x \cos(\theta_B)e_B, \\
\cos(\theta_B)\cos(\phi)\,e_A &= \lambda_y \cos(\theta_A)e_A.
\end{aligned} \tag{270}$$

Then it can be concluded from Eq. (269) that:

$$\begin{aligned}
\sin(\theta_A)u_A &= \lambda_x \lambda_y \sin(\theta_A)u_A, \\
\sin(\theta_B)u_B &= \lambda_x \lambda_y \sin(\theta_B)u_B.
\end{aligned} \tag{271}$$

Eq. (271) leads to two scenarios:

(S1) $\lambda_x \lambda_y = 1$.

(S2) $\sin(\theta_A) = \sin(\theta_B) = 0$.

When (S1) holds, multiply two equations in Eq. (270) and we get:

$$\cos(\theta_A)\cos(\theta_B)\cos^2(\phi) = \cos(\theta_A)\cos(\theta_B). \tag{272}$$

And since:

$$0 < \cos^2(\phi) < 1, \tag{273}$$

we can conclude that:

$$\begin{aligned}
\cos(\theta_A) &= \cos(\theta_B) = 0, \\
\sin(\theta_A) &= \sin(\theta_B) = \pm 1.
\end{aligned} \tag{274}$$

Plugging Eq. (274) into Eq. (269), we get:

$$\begin{aligned}
u_A &= \lambda_x u_B, \\
u_B &= \lambda_y u_A.
\end{aligned} \tag{275}$$

Since $\|u_A\| = \|u_B\| = 1$, Eq. (275) $\Rightarrow \lambda_x = \lambda_y = \pm 1 \Rightarrow u_A = \pm u_B$. And according to Eq. (264) and we have:

$$x = \pm y \in \mathbb{C}. \tag{276}$$

We conclude that (S1) $\Rightarrow x = \pm y \in \mathbb{C}$.

When (S2) holds, we have:

$$\begin{aligned}
\sin(\theta_A) &= \sin(\theta_B) = 0, \\
\cos(\theta_A) &= \cos(\theta_B) = \pm 1.
\end{aligned} \tag{277}$$

Plugging Eq. (277) into Eq. (264), we have:

$$
\begin{aligned}
x &= \pm e_A \perp \mathbb{C}, \\
y &= \pm e_B \perp \mathbb{C}.
\end{aligned}
\tag{278}
$$

We conclude that (S2) $\Rightarrow x \perp \mathbb{C}$ and $y \perp \mathbb{C}$.

So the sufficiency is confirmed.

**Step 3:** $\Leftarrow$ Last, we prove the necessity. If $x = \pm y \in \mathbb{C}$, then

$$
\begin{aligned}
\cos(\theta_A) &= \cos(\theta_B) = 0, \\
\sin(\theta_A) &= \sin(\theta_B) = \pm 1.
\end{aligned}
\tag{279}
$$

and

$$
\begin{aligned}
x &= u_A, \\
y &= u_B,
\end{aligned}
\tag{280}
$$

According to Eq. (267) and Eq. (280), we have:

$$
\begin{aligned}
P_B \cdot x &= u_A = x = \pm y, \\
P_A \cdot y &= u_B = y = \pm x.
\end{aligned}
\tag{281}
$$

Let $\lambda_x = \lambda_y = \pm 1$, conditions (i) and (ii) hold.

If $x \perp \mathbb{C}$ and $y \perp \mathbb{C}$, then:

$$
\begin{aligned}
\sin(\theta_A) &= \sin(\theta_B) = 0, \\
\cos(\theta_A) &= \cos(\theta_B) = \pm 1.
\end{aligned}
\tag{282}
$$

and

$$
\begin{aligned}
x &= \pm e_A, \\
y &= \pm e_B.
\end{aligned}
\tag{283}
$$

According to Eq. (267) and Eq. (283), we have:

$$
\begin{aligned}
P_B \cdot x &= \pm \cos\left(\phi\right) e_B = \pm \cos\left(\phi\right) y, \\
P_A \cdot y &= \pm \cos\left(\phi\right) e_A = \pm \cos\left(\phi\right) x.
\end{aligned}
\tag{284}
$$

Let $\lambda_x = \lambda_y = \pm \cos\left(\phi\right)$, conditions (i) and (ii) hold.

Therefore, the necessity is confirmed.

$\square$

## K.4. Details of Theorem 4

In this section, we provide proofs of Theorem 4 that is proposed in Sec. 4.1. We also provide details and proofs of the auxiliary theorems (Theorem S7 and Theorem S8) and the technical lemmas (Lemma 14 and Lemma 15) that support the proof Theorem 4. For convenience in reading, let us recall some related notions and definitions.

- $h, N \in \mathbb{N}$.

- $\mathbb{S}^{h-1} = \left\{ z \in \mathbb{R}^h : \|z\| = 1 \right\}$.

- $\mathbb{A} = \left\{ x \in \mathbb{R}^h : n_A \cdot x = 0 \right\}$ where $n_A$ is the normal vector of $\mathbb{A}$.

- $\mathbb{B} = \left\{ y \in \mathbb{R}^h : n_B \cdot y = 0 \right\}$ where $n_A$ is the normal vector of $\mathbb{B}$.

- $\phi = \cos^{-1}\left( \frac{n_x \cdot n_y}{\|n_x\| \cdot \|n_y\|} \right)$ and $0 < \phi_{\min} \leq \phi < \frac{\pi}{2}$.

- $\mathbb{S}_X = \mathbb{S}^{h-1} \cap \mathbb{A} = \left\{ x \in \mathbb{R}^h : \|x\| = 1, n_A \cdot x = 0 \right\} \cong S^{h-2} \in \mathbb{S}^{h-1}$.

- $\mathbb{S}_Y = \mathbb{S}^{h-1} \cap \mathbb{B} = \left\{ y \in \mathbb{R}^h : \|y\| = 1, n_B \cdot y = 0 \right\} \cong S^{h-2} \in \mathbb{S}^{h-1}$.

- $\mathbb{C} = \mathbb{A} \cap \mathbb{B}$.

- $h_X = h_Y = h - 1$.

- $h_C = h - 2$.

- $P_A$: the projection matrix of $\mathbb{A}$.

- $P_B$: the projection matrix of $\mathbb{B}$.

- $P_C$: the projection matrix of $\mathbb{C}$.

- $e_A = \{z \in \mathbb{S}_X : z \perp \mathbb{C}\}$.

- $e_B = \{z \in \mathbb{S}_Y : z \perp \mathbb{C}\}$.

- $\mathbb{C}^\perp = \operatorname{span}\{e_A\} \oplus \operatorname{span}\{e_B\}$

- $\mathbb{R}^h = \mathbb{C} \oplus \mathbb{C}^\perp$.

- $X = (x_1, \ldots, x_N) \in (\mathbb{S}_X)^N$.

- $Y = (y_1, \ldots, y_N) \in (\mathbb{S}_Y)^N$.

- $\mu_x = \frac{1}{N} \sum_{i=1}^{N} x_i$.

- $\mu_y = \frac{1}{N} \sum_{i=1}^{N} y_i$.

- $c_x = \frac{\mu_x}{\|\mu_x\|}$.

- $c_y = \frac{\mu_y}{\|\mu_y\|}$.

**Definition** (Multimodal Contrastive Loss (MCL Loss)). Let $(X, Y)$ be an $N$-pair configuration, where $X = (x_1, \ldots, x_N) \in (\mathbb{S}^{h-1})^N$ and $Y = (y_1, \ldots, y_N) \in (\mathbb{S}^{h-1})^N$. $\forall \tau > 0$, the multimodal contrastive loss $\mathcal{L}_{\mathrm{MCL}}(\cdot, \cdot) : (\mathbb{S}^{h-1})^N \times (\mathbb{S}^{h-1})^N \to \mathbb{R}$ is defined as:

$$\mathcal{L}_{\mathrm{MCL}} = \frac{1}{N} \sum_{i=1}^{N} \mathcal{L}_{\mathrm{MCL}}^i, \quad \text{where } \mathcal{L}_{\mathrm{MCL}}^i = \mathcal{L}_{\mathcal{X} \to \mathcal{Y}}(x_i; Y) + \mathcal{L}_{\mathcal{Y} \to \mathcal{X}}(y_i; X).$$

Here, $\mathcal{L}_{\mathcal{X} \to \mathcal{Y}}$ is the $\mathcal{X}$-to-$\mathcal{Y}$ alignment and $\mathcal{L}_{\mathcal{Y} \to \mathcal{X}}$ is the $\mathcal{Y}$-to-$\mathcal{X}$ alignment, which are defined respectively as:

$$\mathcal{L}_{\mathcal{X}\to\mathcal{Y}}(x_i; Y) = -\log \frac{\exp\left(x_i \cdot y_i/\tau\right)}{\sum_{j=1}^{N} \exp\left(x_i \cdot y_j/\tau\right)}, \quad \mathcal{L}_{\mathcal{Y}\to\mathcal{X}}(y_i; X) = -\log \frac{\exp\left(x_i \cdot y_i/\tau\right)}{\sum_{j=1}^{N} \exp\left(x_j \cdot y_i/\tau\right)}.$$

**Definition**(Modality Gap) Let $(X, Y)$ be an $N$-pair configuration, where $X = (x_1, \ldots, x_N) \in (\mathbb{S}^{h-1})^N$ and $Y = (y_1, \ldots, y_N) \in (\mathbb{S}^{h-1})^N$. The modality gap between $X$ and $Y$ can be expressed as the angle between the center representations:

$$\Delta_\theta = \cos^{-1}(c_x \cdot c_y).$$

**Definition** (vMF Distribution). $\forall c \in \mathbb{S}^{h-1}$ and $\kappa \geq 0$, the probability density of a random $h$-dimensional unit vector $z \sim \text{vMF}(c, \kappa)$ is given by:

$$f_h(z; c, \kappa) = D_h(\kappa) e^{\kappa c^\top z}, \quad \text{where } D_h(\kappa) = \frac{\kappa^\nu}{(2\pi)^{\nu+1} I_\nu(\kappa)}.$$

Here, $\nu = h/2 - 1$, and $I_\nu(\cdot) : \mathbb{R} \to \mathbb{R}$ is the modified Bessel function of the first kind of order $\nu$, which is defined as:

$$I_\nu(x) = \sum_{k=0}^{\infty} \frac{1}{k! \Gamma(\nu + k + 1)} \left(\frac{x}{2}\right)^{2k+\nu}.$$

**Definition** (Function $\tilde{M}$). $\forall \kappa, \tau > 0$, a function $\tilde{M}_\kappa(\cdot, \cdot) : [-1, 1] \times [0, 1] \to \mathbb{R}_0^+$ is defined as:

$$\tilde{M}_\kappa(w, t) = \sqrt{\kappa^2 + \frac{2\kappa w}{\tau} + \frac{t^2}{\tau^2}}.$$

**Definition** (Function $\tilde{\mathcal{J}}$). $\forall \kappa, \nu, \tau > 0$, $\tilde{\mathcal{J}}(\cdot, \cdot, \cdot; \kappa, \nu) : [-1, 1] \times [-1, 1] \times [0, 1] \to \mathbb{R}$ is defined as:

$$\tilde{\mathcal{J}}(w_1, w_2, t; \kappa, \nu) = -\frac{w_1}{\tau} + \log\left(\frac{I_\nu\left(\tilde{M}_\kappa(w_2, t)\right)}{\tilde{M}_\kappa(w_2, t)^\nu}\right) - \log\left(\frac{I_\nu(\kappa)}{\kappa^\nu}\right).$$

**Definition** (Function $M$). $\forall \kappa, \tau > 0$, a function $M_\kappa(\cdot) : [-1, 1] \to \mathbb{R}_0^+$ is defined as:

$$M_\kappa(w) = \sqrt{\kappa^2 + \frac{2\kappa w}{\tau} + \frac{1}{\tau^2}}$$
$$= \tilde{M}_\kappa(w, 1).$$

**Definition** (Function $\mathcal{J}$). $\forall \kappa, \nu, \tau > 0$, a function $\mathcal{J}(\cdot; \kappa, \nu) : [-1, 1] \to \mathbb{R}$ is defined as:

$$\mathcal{J}(w; \kappa, \nu) = -\frac{w}{\tau} + \log\left(\frac{I_\nu(M_\kappa(w))}{M_\kappa(w)^\nu}\right) - \log\left(\frac{I_\nu(\kappa)}{\kappa^\nu}\right)$$
$$= \tilde{\mathcal{J}}(w, w, 1; \kappa, \nu).$$

**Definition** (Function $\tilde{M}$). $\forall \kappa, \tau > 0$, a function $\tilde{M}_\kappa(\cdot, \cdot) : [-1, 1] \times [0, 1] \to \mathbb{R}_0^+$ is defined as:

$$\tilde{M}_\kappa(w, t) = \tilde{M}_\kappa(w, t).$$

**Definition** (Function $\hat{\mathcal{J}}$). $\forall \kappa, \nu, \tau > 0$, a function $\hat{\mathcal{J}}(\cdot, \cdot; \kappa, \nu) : [-1, 1] \times [0, 1] \to \mathbb{R}$ is defined as:

$$\hat{\mathcal{J}}(w, t; \kappa, \nu) = -\frac{w}{\tau} + \log\left(\frac{I_\nu\left(\tilde{M}_\kappa(w, t)\right)}{\tilde{M}_\kappa(w, t)^\nu}\right) - \log\left(\frac{I_\nu(\kappa)}{\kappa^\nu}\right)$$

$$= \tilde{\mathcal{J}}(w, w, t; \kappa, \nu).$$

### K.4.1. PROOF OF THEOREM 4

In this subsection, we provide the proof of Theorem 4. For convenience in reading, we first restate Theorem 4 here.

The Platonic Representation Hypothesis (Huh et al., 2024) suggests that contrastive learners are optimized by representations of $X$ and $Y$ whose intra-modal kernels (i.e., pairwise similarities) align. Building on this idea, we define the kernel alignment as *Intra-Modal Isometry*.

**Definition 4**[Intra-Modal Isometry (IMS)] Let $(X, Y)$ be an $N$-pair configuration in $\mathbb{R}^h$, we say $(X, Y)$ achieves Intra-Modal Isometry if and only if $\forall i, j \in [N], i \neq j, x_i \cdot x_j = y_i \cdot y_j$.

The Intra-Modal Isometry assumption implies that $\forall i \in [N], x_i \cdot c_x = y_i \cdot c_y$, and thus $\kappa_x = \kappa_y$. We view this as a necessary condition for the COR.

**Theorem 4.** [Restate] Let $(X, Y)$ be an $N$-pair configuration, where $X = (x_1, \ldots, x_N) \in (\mathbb{S}_X \setminus \mathbb{C})^N$ are $iid$ samples from $\mu_x = \text{vMF}(c_x, \kappa_x)$, and $Y = (y_1, \ldots, y_N) \in (\mathbb{S}_Y \setminus \mathbb{C})^N$ are $iid$ samples from $\mu_y = \text{vMF}(c_y, \kappa_y)$. Let $\tilde{\nu} = (h-1)/2 - 1$. Denote $\Delta_\theta = \cos^{-1}(c_x \cdot c_y)$ and assume $c_x, c_y \perp \mathbb{C}$ with $c_x \cdot c_y > 0$. Suppose $(X, Y)$ achieves Intra-Modal Isometry. Then $\forall i \in [N]$, denote $\theta_i^c = \cos^{-1}(x_i \cdot c_x) = \cos^{-1}(y_i \cdot c_y)$, and $\kappa = \kappa_x = \kappa_y$. Let $\theta_i^c \in (0, \frac{\pi}{2})$ and $\kappa > 0$, it holds that:

$$\lim_{N \to \infty} \mathcal{L}_{\text{MCL}}^{i \neq c} - 2\log(N)$$
$$= \tilde{\mathcal{J}}\left(\cos(\Delta_\theta), \cos(\theta_i^c), \|P_B x_i\|; \kappa, \tilde{\nu}\right) + \tilde{\mathcal{J}}\left(\cos(\Delta_\theta), \cos(\theta_i^c), \|P_A y_i\|; \kappa, \tilde{\nu}\right)$$
$$\geq 2\tilde{\mathcal{J}}\left(\cos^2(\theta_i^c)\cos(\phi_{\min}) + \sin^2(\theta_i^c), \cos(\theta_i^c), \sqrt{\cos^2(\theta_i^c)\cos^2(\phi_{\min}) + \sin^2(\theta_i^c)}; \kappa, \tilde{\nu}\right),$$

where equality is attained if and only if there exists a configuration of $(X, Y)$ such that:

(A6) $P_C x_i = P_C y_i$.

(A7) $\Delta_\theta = \cos^{-1}(c_x \cdot c_y) = \phi_{\min}$.

*Proof.* According to Theorem S7, the convergent function of $\lim_{N \to \infty} \mathcal{L}_{\text{MCL}}^{i \neq c} - 2\log(N)$ is:

$$\lim_{N \to \infty} \mathcal{L}_{\text{MCL}}^{i \neq c} - 2\log(N) = \lim_{N \to \infty}\left(\mathcal{L}_{\mathcal{X} \to \mathcal{Y}}(x_{i \neq c}; Y) - \log(N) + \mathcal{L}_{\mathcal{Y} \to \mathcal{X}}(y_{i \neq c}; X) - \log(N)\right)$$
$$= \tilde{\mathcal{J}}\left(w_i, w_i^c, \|P_B x_i\|; \kappa_y, \tilde{\nu}\right) + \tilde{\mathcal{J}}\left(w_i, w_i^c, \|P_A y_i\|; \kappa_x, \tilde{\nu}\right) \tag{285}$$
$$= 2\tilde{\mathcal{J}}\left(w_i, w_i^c, t; \kappa, \tilde{\nu}\right),$$

where

$$w_i = \cos^2(\theta_i^c)\cos(\Delta_\theta) + (\theta_i^c)(P_C \cdot x_i) \cdot (P_C \cdot y_i),$$
$$w_i^c = \cos(\theta_i^c), \tag{286}$$
$$t = \sqrt{\cos^2(\theta_i^c)\cos^2(\Delta_\theta) + \sin^2(\theta_i^c)}.$$

And Theorem S8 shows the lower bound of the convergent function is:

$$2\tilde{\mathcal{J}}\left(w_i, w_i^c, t; \kappa, \tilde{\nu}\right) \geq 2\tilde{\mathcal{J}}\left(w_{i,\min}, w_i^c, t_{\min}; \kappa, \tilde{\nu}\right), \tag{287}$$

where

$$w_{i,\min} = \cos^2\left(\theta_i^c\right)\cos\left(\phi_{\min}\right) + \sin^2\left(\theta_i^c\right),$$
$$t_{\min} = \sqrt{\cos^2\left(\theta_i^c\right)\cos^2\left(\phi_{\min}\right) + \sin^2\left(\theta_i^c\right)}, \tag{288}$$

and equality is attained if and only if there exists a configuration of $(X, Y)$ such that:

(i) $P_C \cdot x_i = P_C \cdot y_i$.

(ii) $\Delta_\theta = \phi_{\min}$.

Combining Eq. (285) and Eq. (288), we conclude that:

$$\lim_{N\to\infty} \mathcal{L}_{\mathrm{MCL}}^{i\neq c} - 2\log(N)$$
$$= \tilde{\mathcal{J}}\left(\cos\left(\Delta_\theta\right), \cos\left(\theta_i^c\right), \|P_B x_i\|; \kappa, \tilde{\nu}\right) + \tilde{\mathcal{J}}\left(\cos\left(\Delta_\theta\right), \cos\left(\theta_i^c\right), \|P_A y_i\|; \kappa, \tilde{\nu}\right) \tag{289}$$
$$\geq 2\tilde{\mathcal{J}}\left(\cos^2\left(\theta_i^c\right)\cos\left(\phi_{\min}\right) + \sin^2\left(\theta_i^c\right), \cos\left(\theta_i^c\right), \sqrt{\cos^2\left(\theta_i^c\right)\cos^2\left(\phi_{\min}\right) + \sin^2\left(\theta_i^c\right)}; \kappa, \tilde{\nu}\right),$$

where equality is attained if and only if there exists a configuration of $(X, Y)$ such that:

(A6) $P_C x_i = P_C y_i$.

(A7) $\Delta_\theta = \cos^{-1}\left(c_x \cdot c_y\right) = \phi_{\min}$.

$\square$

### K.4.2. AUXILIARY THEOREMS PART 4

In this subsection, we provide details and proofs of the auxiliary theorems (Theorem S5 and Theorem S7) that support the proof of Theorem 4.

**Theorem S7.** Let $(X, Y)$ be an $N$-pair configuration, where $X = (x_1, \ldots, x_N) \in (\mathbb{S}_X \setminus \mathbb{C})^N$ are *iid* samples from $\mu_x = \mathrm{vMF}(c_x, \kappa_x)$, and $Y = (y_1, \ldots, y_N) \in (\mathbb{S}_Y \setminus \mathbb{C})^N$ are *iid* samples from $\mu_y = \mathrm{vMF}(c_y, \kappa_y)$. Let $\tilde{\nu} = (h-1)/2 - 1$. Denote $\Delta_\theta = \cos^{-1}\left(c_x \cdot c_y\right)$ and assume $c_x, c_y \perp \mathbb{C}$ with $c_x \cdot c_y > 0$. Suppose $(X, Y)$ achieves Intra-Modal Isometry. Then $\forall i \in [N]$, denote $\theta_i^c = \cos^{-1}\left(x_i \cdot c_x\right) = \cos^{-1}\left(y_i \cdot c_y\right)$, and $\kappa = \kappa_x = \kappa_y$. Let $\kappa > 0$, it holds that:

$$\lim_{N\to\infty} \mathcal{L}_{\mathrm{MCL}}^{i\neq c} - 2\log(N) = \lim_{N\to\infty}\left(\mathcal{L}_{\mathcal{X}\to\mathcal{Y}}(x_{i\neq c}; Y) - \log(N) + \mathcal{L}_{\mathcal{Y}\to\mathcal{X}}(y_{i\neq c}; X) - \log(N)\right)$$
$$= \tilde{\mathcal{J}}\left(w_i, w_i^c, \|P_B x_i\|; \kappa_y, \tilde{\nu}\right) + \tilde{\mathcal{J}}\left(w_i, w_i^c, \|P_A y_i\|; \kappa_x, \tilde{\nu}\right) \tag{290}$$
$$= 2\tilde{\mathcal{J}}\left(w_i, w_i^c, t; \kappa, \tilde{\nu}\right),$$

where

$$w_i = \cos^2\left(\theta_i^c\right)\cos\left(\Delta_\theta\right) + \left(\theta_i^c\right)\left(P_C \cdot x_i\right)\cdot\left(P_C \cdot y_i\right),$$
$$w_i^c = \cos\left(\theta_i^c\right),$$
$$t = \sqrt{\cos^2\left(\theta_i^c\right)\cos^2\left(\Delta_\theta\right) + \sin^2\left(\theta_i^c\right)}. \tag{291}$$

*Proof.* **Step 1**: We first decompose $\lim_{N\to\infty} \mathcal{L}_{\mathrm{MCL}}^{i\neq c} - 2\log(N)$ into two parts:

$$\lim_{N \to \infty} \mathcal{L}_{\text{MCL}}^{i \neq c} - 2 \log(N) = \lim_{N \to \infty} \mathcal{L}_{\mathcal{X} \to \mathcal{Y}}(x_{i \neq c}; Y) - \log(N)$$
$$+ \lim_{N \to \infty} \mathcal{L}_{\mathcal{Y} \to \mathcal{X}}(y_{i \neq c}; X) - \log(N). \tag{292}$$

The convergent function of $\mathcal{L}_{\mathcal{X} \to \mathcal{Y}}(x_{i \neq c}; Y)$ as $N \to \infty$. $\forall i \in [N], i \neq c, x_i \in X$, denote $w_i = x_i \cdot y_i$, $w_{x_i, c_y} = x_i \cdot c_y$ and $w_{y_i, c_x} = y_i \cdot c_x$. $\forall \kappa_y > 0$, as prove in Theorem S5:

$$\lim_{N \to \infty} \mathcal{L}_{\mathcal{X} \to \mathcal{Y}}(x_i; Y) - \log(N) = \lim_{N \to \infty} - \log \frac{\exp(x_i \cdot y_i / \tau)}{\sum_{j=1}^{N} \exp(x_i \cdot y_j / \tau)} - \log(N)$$
$$= -\frac{w_i}{\tau} + \log \left( \frac{I_{\tilde{\nu}} \left( \tilde{M}_{\kappa_y} \left( w_{x_i, c_y}, \|P_B x_i\| \right) \right)}{\tilde{M}_{\kappa_y} \left( w_{x_i, c_y}, \|P_B x_i\| \right)^{\tilde{\nu}}} \right) - \log \left( \frac{I_{\tilde{\nu}}(\kappa_y)}{\kappa_y^{\tilde{\nu}}} \right) \tag{293}$$
$$= \tilde{\mathcal{J}} \left( w_i, w_{x_i, c_y}, \|P_B c_x\| ; \kappa_y, \tilde{\nu} \right),$$

where $\forall \kappa, \tau > 0$, $\tilde{\mathcal{J}}(\cdot, \cdot, \cdot; \kappa, \tilde{\nu})$ is a function on $[-1, 1] \times [-1, 1] \times [0, 1]$ and $\tilde{M}_\kappa(\cdot, \cdot) : [-1, 1] \times [0, 1] \to \mathbb{R}_0^+$ is defined as:

$$\tilde{M}_\kappa(w, t) = \sqrt{\kappa^2 + \frac{2\kappa w}{\tau} + \frac{t^2}{\tau^2}}, \tag{294}$$

and $I_\nu$ is the modified Bessel function of the first kind of order $\nu$, which is defined as:

$$I_\nu(m) = \sum_{k=0}^{\infty} \frac{1}{k! \Gamma(\nu + k + 1)} \left( \frac{m}{2} \right)^{2k + \nu}. \tag{295}$$

When $(X, Y)$ achieves Intra-Modal Isometry, we have $w_{x_i, c_y} = x_i \cdot c_x = y_i \cdot c_x = w_{y_i, c_x}$ Denote $w_i^c = w_{x_i, c_y} = w_{y_i, c_x} = \cos(\theta_i^c)$. This implies $\kappa_x = \kappa_y = \kappa$.

Then, Eq. (293) can be re-written as:

$$\lim_{N \to \infty} \mathcal{L}_{\mathcal{X} \to \mathcal{Y}}(x_i; Y) - \log(N) = -\frac{w_i}{\tau} + \log \left( \frac{I_{\tilde{\nu}} \left( \tilde{M}_\kappa \left( w_i^c, \|P_B x_i\| \right) \right)}{\tilde{M}_\kappa \left( w_i^c, \|P_B x_i\| \right)^{\tilde{\nu}}} \right) - \log \left( \frac{I_{\tilde{\nu}}(\kappa)}{\kappa^{\tilde{\nu}}} \right)$$
$$= \tilde{\mathcal{J}} \left( w_i, w_i^c, \|P_B c_x\| ; \kappa, \tilde{\nu} \right). \tag{296}$$

Similarly, the convergent function of $\mathcal{L}_{\mathcal{Y} \to \mathcal{X}}(y_{i \neq c}; X)$ as $N \to \infty$ can be written as:

$$\lim_{N \to \infty} \mathcal{L}_{\mathcal{Y} \to \mathcal{X}}(y_i; X) - \log(N) = \lim_{N \to \infty} - \log \frac{\exp(x_i \cdot y_i / \tau)}{\sum_{j=1}^{N} \exp(x_i \cdot y_j / \tau)} - \log(N)$$
$$= -\frac{w_i}{\tau} + \log \left( \frac{I_{\tilde{\nu}} \left( \tilde{M}_{\kappa_x} \left( w_{y_i, c_x}, \|P_A y_i\| \right) \right)}{\tilde{M}_{\kappa_x} \left( w_{y_i, c_x}, \|P_A y_i\| \right)^{\tilde{\nu}}} \right) - \log \left( \frac{I_{\tilde{\nu}}(\kappa_x)}{\kappa_x^{\tilde{\nu}}} \right)$$
$$= -\frac{w_i}{\tau} + \log \left( \frac{I_{\tilde{\nu}} \left( \tilde{M}_\kappa \left( w_i^c, \|P_A y_i\| \right) \right)}{\tilde{M}_\kappa \left( w_i^c, \|P_A y_i\| \right)^{\tilde{\nu}}} \right) - \log \left( \frac{I_{\tilde{\nu}}(\kappa)}{\kappa^{\tilde{\nu}}} \right)$$
$$= \tilde{\mathcal{J}} \left( w_i, w_i^c, \|P_A y_i\| ; \kappa, \tilde{\nu} \right). \tag{297}$$

**Step 2** Now, let us decompose the embedding space. Define two vectors $e_A$ and $e_B$ such that:

$$e_A \in \mathbb{S}_X, \quad \text{and} \quad e_A \perp \mathbb{C},$$
$$e_B \in \mathbb{S}_Y, \quad \text{and} \quad e_B \perp \mathbb{C}.$$
$$(298)$$

Let $\mathbb{C}^\perp$ be the 2-dimensional orthogonal complement of $C$, and $\mathbb{C}^\perp$ satisfies:

$$\mathbb{C}^\perp = \text{span}\{e_A\} \oplus \text{span}\{e_B\},$$
$$\mathbb{R}^h = \mathbb{C} \oplus \mathbb{C}^\perp.$$
$$(299)$$

Since $n_A, n_B \in \mathbb{C}^\perp$, $n_A \perp e_A$ and $n_B \perp e_B$, we have:

$$\langle e_A, e_B \rangle = \pm \langle n_A, n_B \rangle,$$
$$(300)$$

and we choose a pair of $e_A$ and $e_B$ such that:

$$\langle e_A, e_B \rangle = \langle n_A, n_B \rangle = \cos(\phi) \in (0, 1).$$
$$(301)$$

Denote $\theta_i = \cos^{-1}(w_i)$. When $c_x, c_y \perp \mathbb{C}$, $\Delta_\theta = \phi$. And without loss of generality, we can set the coordinate as:

$$n_A = (\sin\left(\frac{\Delta_\theta}{2}\right), -\cos\left(\frac{\Delta_\theta}{2}\right), 0, 0, \cdots, 0),$$
$$n_B = (-\sin\left(\frac{\Delta_\theta}{2}\right), -\cos\left(\frac{\Delta_\theta}{2}\right), 0, 0, \cdots, 0),$$
$$c_x = e_A = (\cos\left(\frac{\Delta_\theta}{2}\right), \sin\left(\frac{\Delta_\theta}{2}\right), 0, 0, \cdots, 0),$$
$$c_y = e_B = (\cos\left(\frac{\Delta_\theta}{2}\right), -\sin\left(\frac{\Delta_\theta}{2}\right), 0, 0, \cdots, 0),$$
$$\mathbb{C} = \text{span}\{e_3\} \oplus \text{span}\{e_3\} \oplus \cdots, \oplus \text{span}\{e_h\}.$$
$$(302)$$

Therefore, $\forall x_i \in \mathbb{S}_X = \mathbb{A} \cap \mathbb{S}^{h-1}$ and $\forall y_i \in \mathbb{S}_Y = \mathbb{B} \cap \mathbb{S}^{h-1}$, $\exists u_i^x, u_i^y \in \mathbb{C} \cap \mathbb{S}^{h-1}$, such that:

$$x_i = \cos(\theta_i^c) e_A + \sin(\theta_i^c) u_i^x = \cos(\theta_i^c) c_x + \sin(\theta_i^c) u_i^x,$$
$$y_i = \cos(\theta_i^c) e_B + \sin(\theta_i^c) u_i^y = \cos(\theta_i^c) c_y + \sin(\theta_i^c) u_i^y.$$
$$(303)$$

Using orthogonality, we have:

$$P_B \cdot e_A = \langle e_A, e_B \rangle e_B = \cos(\Delta_\theta) e_B,$$
$$P_B \cdot u_i^x = u_i^x,$$
$$(304)$$

and

$$P_A \cdot e_B = \langle e_A, e_B \rangle e_A = \cos(\Delta_\theta) e_A,$$
$$P_A \cdot u_i^y = u_i^y,$$
$$(305)$$

and

$$P_C \cdot e_A = P_C \cdot e_B = 0,$$
$$P_C \cdot u_i^x = u_i^x,$$
$$P_C \cdot u_i^y = u_i^y.$$
$$(306)$$

Then the projections of $(x_i, y_i)$ are:

$$P_B \cdot x_i = \cos(\theta_i^c) \cos(\Delta_\theta) e_B + \sin(\theta_i^c) u_i^x = \cos(\theta_i^c) \cos(\Delta_\theta) c_y + \sin(\theta_i^c) u_i^x,$$
$$P_A \cdot y_i = \cos(\theta_i^c) \cos(\Delta_\theta) e_A + \sin(\theta_i^c) u_i^y = \cos(\theta_i^c) \cos(\Delta_\theta) c_x + \sin(\theta_i^c) u_i^y,$$

(307)

and

$$P_C \cdot x_i = \sin(\theta_i^c) u_i^x,$$
$$P_C \cdot y_i = \sin(\theta_i^c) u_i^y.$$

(308)

Therefore, we get:

$$
\begin{aligned}
w_i = x_i \cdot y_i &= \cos^2(\theta_i^c) c_x \cdot c_y + \sin^2(\theta_i^c) u_i^x \cdot u_i^y \\
&= \cos^2(\theta_i^c) \cos(\Delta_\theta) + \sin^2(\theta_i^c) u_i^x \cdot u_i^y \\
&= \cos^2(\theta_i^c) \cos(\Delta_\theta) + (P_C \cdot x_i) \cdot (P_C \cdot y_i),
\end{aligned}
$$

(309)

$$
\begin{aligned}
\|P_B x_i\| &= \sqrt{\cos^2(\theta_i^c) \cos^2(\Delta_\theta) c_y \cdot c_y + 2\cos(\theta_i^c)\cos(\Delta_\theta)\sin(\theta_i^c) c_y \cdot u_i^x + \sin^2(\theta_i^c) u_i^x \cdot u_i^x} \\
&= \sqrt{\cos^2(\theta_i^c)\cos^2(\Delta_\theta) + 2\cos(\theta_i^c)\cos(\Delta_\theta)\sin(\theta_i^c) c_y \cdot u_i^x + \sin^2(\theta_i^c)} \\
&= \sqrt{\cos^2(\theta_i^c)\cos^2(\Delta_\theta) + \sin^2(\theta_i^c)},
\end{aligned}
$$

(310)

and

$$
\begin{aligned}
\|P_A y_i\| &= \sqrt{\cos^2(\theta_i^c)\cos^2(\Delta_\theta) c_x \cdot c_x + 2\cos(\theta_i^c)\cos(\Delta_\theta)\sin(\theta_i^c) c_x \cdot u_i^y + \sin^2(\theta_i^c) u_i^y \cdot u_i^y} \\
&= \sqrt{\cos^2(\theta_i^c)\cos^2(\Delta_\theta) + 2\cos(\theta_i^c)\cos(\Delta_\theta)\sin(\theta_i^c) c_y \cdot u_i^y + \sin^2(\theta_i^c)} \\
&= \sqrt{\cos^2(\theta_i^c)\cos^2(\Delta_\theta) + \sin^2(\theta_i^c)}, \\
&= \|P_B x_i\|.
\end{aligned}
$$

(311)

Let $t = \|P_B x_i\| = \|P_A y_i\|$. Plugging Eq. (309), Eq. (310) and Eq. (311) into Eq. (292), Eq. (296) and Eq. (297), we conclude that:

$$
\begin{aligned}
\lim_{N \to \infty} \mathcal{L}_{\text{MCL}}^{i \neq c} - 2\log(N) &= \lim_{N \to \infty} \mathcal{L}_{\mathcal{X} \to \mathcal{Y}}(x_{i \neq c}; Y) - \log(N) \\
&\quad + \lim_{N \to \infty} \mathcal{L}_{\mathcal{Y} \to \mathcal{X}}(y_{i \neq c}; X) - \log(N) \\
&= \tilde{\mathcal{J}}(w_i, w_i^c, \|P_B x_i\|; \kappa, \tilde{\nu}) + \tilde{\mathcal{J}}(w_i, w_i^c, \|P_A y_i\|; \kappa, \tilde{\nu}) \\
&= 2\tilde{\mathcal{J}}(w_i, w_i^c, t; \kappa, \tilde{\nu}),
\end{aligned}
$$

(312)

where

$$
\begin{aligned}
w_i &= x_i \cdot y_i = \cos^2(\theta_i^c)\cos(\Delta_\theta) + (P_C \cdot x_i) \cdot (P_C \cdot y_i), \\
w_i^c &= x_i \cdot c_y = y_i \cdot c_x = \cos(\theta_i^c), \\
t &= \sqrt{\cos^2(\theta_i^c)\cos^2(\Delta_\theta) + \sin^2(\theta_i^c)}.
\end{aligned}
$$

(313)

$\square$

**Theorem S8.** Let $(X, Y)$ be an $N$-pair configuration, where $X = (x_1, \ldots, x_N) \in (\mathbb{S}_X \setminus \mathbb{C})^N$ are $iid$ samples from $\mu_x = \text{vMF}(c_x, \kappa_x)$, and $Y = (y_1, \ldots, y_N) \in (\mathbb{S}_Y \setminus \mathbb{C})^N$ are $iid$ samples from $\mu_y = \text{vMF}(c_y, \kappa_y)$. Let $\tilde{\nu} = (h-1)/2 - 1$. Denote $\Delta_\theta = \cos^{-1}(c_x \cdot c_y)$ and assume $c_x, c_y \perp \mathbb{C}$ with $c_x \cdot c_y > 0$. $\forall i \in [N]$, suppose $\theta_i^c = \cos^{-1}(x_i \cdot c_x) = \cos^{-1}(y_i \cdot c_y) \in (0, \frac{\pi}{2})$ and $\kappa > 0$, it holds that:

$$\tilde{\mathcal{J}}\left(w_i, w_i^c, t; \kappa, \tilde{\nu}\right) \geq \tilde{\mathcal{J}}\left(w_{i,\min}, w_i^c, t_{\min}; \kappa, \tilde{\nu}\right), \tag{314}$$

where

$$
\begin{aligned}
w_i &= \cos^2\left(\theta_i^c\right) \cos\left(\Delta_\theta\right) + (P_C \cdot x_i) \cdot (P_C \cdot y_i), \\
w_i^c &= \cos\left(\theta_i^c\right), \\
t &= \sqrt{\cos^2\left(\theta_i^c\right) \cos^2\left(\Delta_\theta\right) + \sin^2\left(\theta_i^c\right)}, \\
w_{i,\min} &= \cos^2\left(\theta_i^c\right) \cos\left(\phi_{\min}\right) + \sin^2\left(\theta_i^c\right), \\
t_{\min} &= \sqrt{\cos^2\left(\theta_i^c\right) \cos^2\left(\phi_{\min}\right) + \sin^2\left(\theta_i^c\right)},
\end{aligned}
\tag{315}
$$

and equality is attained if and only if there exists a configuration of $(X, Y)$ such that:

(B6) $P_C \cdot x_i = P_C \cdot y_i$.

(B7) $\Delta_\theta = \phi_{\min}$.

*Proof.* **Step 1**: Similarly to the proof of Theorem S6 in Sec. K.3.2, we start the proof by finding the convergent function of $\lim_{N \to \infty} \mathcal{L}_{\text{MCL}}^{i \neq c} - 2\log(N)$ as $N \to \infty$. Let $w_i =$

As proven in Theorem S7:

$$
\begin{aligned}
\lim_{N \to \infty} \mathcal{L}_{\text{MCL}}^{i \neq c} - 2\log(N) &= \lim_{N \to \infty} \left(\mathcal{L}_{\mathcal{X} \to \mathcal{Y}}(x_{i \neq c}; Y) - \log(N) + \mathcal{L}_{\mathcal{Y} \to \mathcal{X}}(y_{i \neq c}; X) - \log(N)\right) \\
&= \tilde{\mathcal{J}}\left(w_i, w_i^c, \|P_B x_i\|; \kappa, \tilde{\nu}\right) + \tilde{\mathcal{J}}\left(w_i, w_i^c, \|P_A y_i\|; \kappa, \tilde{\nu}\right) \\
&= 2\tilde{\mathcal{J}}\left(w_i, w_i^c, t; \kappa, \tilde{\nu}\right).
\end{aligned}
\tag{316}
$$

$\forall \kappa, \nu, \tau > 0$, $\tilde{\mathcal{J}}(\cdot, \cdot, \cdot; \kappa, \nu) : [-1, 1] \times [-1, 1] \times [0, 1] \to \mathbb{R}$ is defined as:

$$\tilde{\mathcal{J}}\left(w_1, w_2, t; \kappa, \nu\right) = -\frac{w_1}{\tau} + \log\left(\frac{I_\nu\left(\tilde{M}_\kappa(w_2, t)\right)}{\tilde{M}_\kappa(w_2, t)^\nu}\right) - \log\left(\frac{I_\nu(\kappa)}{\kappa^\nu}\right), \tag{317}$$

and $\tilde{M}_\kappa(\cdot, \cdot) : [-1, 1] \times [0, 1] \to \mathbb{R}_0^+$ is defined as:

$$\tilde{M}_\kappa(w, t) = \sqrt{\kappa^2 + \frac{2\kappa w}{\tau} + \frac{t^2}{\tau^2}}. \tag{318}$$

and $I_\nu$ is the modified Bessel function of the first kind of order $\nu$, which is defined as:

$$I_\nu(m) = \sum_{k=0}^{\infty} \frac{1}{k! \Gamma(\nu + k + 1)} \left(\frac{m}{2}\right)^{2k+\nu}, \tag{319}$$

and

$$w_i = \cos^2\left(\theta_i^c\right)\cos\left(\Delta_\theta\right) + \sin^2\left(\theta_i^c\right)\left(P_C \cdot x_i\right) \cdot \left(P_C \cdot y_i\right),$$
$$w_i^c = \cos\left(\theta_i^c\right), \tag{320}$$
$$t = \sqrt{\cos^2\left(\theta_i^c\right)\cos^2\left(\Delta_\theta\right) + \sin^2\left(\theta_i^c\right)}.$$

**Step 2:**

According to the Cauchy-Schwarz inequality and Eq. (308):

$$\left(P_C \cdot x_i\right) \cdot \left(P_C \cdot y_i\right) \leq \sin^2\left(\theta_i^c\right), \tag{321}$$

where equality is attained if and only if there exists a configuration of $(X, Y)$ such that:

(B6) $P_C \cdot x_i = P_C \cdot y_i$.

And therefore:

$$w_i = \cos^2\left(\theta_i^c\right)\cos\left(\Delta_\theta\right) + \left(P_C \cdot x_i\right) \cdot \left(P_C \cdot y_i\right)$$
$$\leq \cos^2\left(\theta_i^c\right)\cos\left(\Delta_\theta\right) + \leq \sin^2\left(\theta_i^c\right), \tag{322}$$

and then $\tilde{\mathcal{J}}\left(w_i, w_i^c, t; \kappa, \tilde{\nu}\right)$ in Eq. (316) can be bounded below by:

$$\tilde{\mathcal{J}}\left(w_i, w_i^c, t; \kappa, \tilde{\nu}\right) \geq \tilde{\mathcal{J}}\big(\cos^2\left(\theta_i^c\right)\cos\left(\Delta_\theta\right) + \sin^2\left(\theta_i^c\right),$$
$$\cos\left(\theta_i^c\right), \tag{323}$$
$$\sqrt{\cos^2\left(\theta_i^c\right)\cos^2\left(\Delta_\theta\right) + \sin^2\left(\theta_i^c\right)}; \kappa, \tilde{\nu}\big).$$

Here, for any given non-center pair $(x_i, y_i)_{i \neq c}$, $\theta_i^c$ is fixed, then the RHS of Eq. (323) becomes a function of $\cos\left(\Delta_\theta\right)$.

Denote:

$$f_1\left(\cos\left(\Delta_\theta\right)\right) := \cos^2\left(\theta_i^c\right)\cos\left(\Delta_\theta\right) + \sin^2\left(\theta_i^c\right),$$
$$f_2\left(\cos\left(\Delta_\theta\right)\right) := \sqrt{\cos^2\left(\theta_i^c\right)\cos^2\left(\Delta_\theta\right) + \sin^2\left(\theta_i^c\right)}, \tag{324}$$

then the Eq. (316) can be re-written as:

$$\tilde{\mathcal{J}}\left(w_i, w_i^c, t; \kappa, \tilde{\nu}\right) \geq \tilde{\mathcal{J}}\left(f_1\left(\cos\left(\Delta_\theta\right)\right), \cos\left(\theta_i^c\right), f_2\left(\cos\left(\Delta_\theta\right)\right); \kappa, \tilde{\nu}\right). \tag{325}$$

According to Lemma 14, $\tilde{\mathcal{J}}\left(f_1\left(\cos\left(\Delta_\theta\right)\right), \cos\left(\theta_i^c\right), f_2\left(\cos\left(\Delta_\theta\right)\right); \kappa, \tilde{\nu}\right)$ is a decreasing function of $\cos\left(\Delta_\theta\right)$ when $\theta_i^c \in [0, \frac{\pi}{2}]$, we have:

$$\tilde{\mathcal{J}}\left(f_1\left(\cos\left(\Delta_\theta\right)\right), \cos\left(\theta_i^c\right), f_2\left(\cos\left(\Delta_\theta\right)\right)\right) \geq \tilde{\mathcal{J}}\left(f_1\left(\cos\left(\phi_{\min}\right)\right), \cos\left(\theta_i^c\right), f_2\left(\cos\left(\phi_{\min}\right)\right)\right). \tag{326}$$

where equality is attained if and only if there exists a configuration of $(X, Y)$ such that:

(B7) $\Delta_\theta = \phi_{\min}$.

Combining Eq. (321) and Eq. (326), we conclude that:

$$\tilde{\mathcal{J}}\left(w_i, w_i^c, t; \kappa, \tilde{\nu}\right) \geq \tilde{\mathcal{J}}\left(w_{i,\min}, w_i^c, t_{\min}; \kappa, \tilde{\nu}\right), \tag{327}$$

where

$$
\begin{aligned}
w_i &= \cos^2\left(\theta_i^c\right)\cos\left(\Delta_\theta\right) + \left(P_C \cdot x_i\right) \cdot \left(P_C \cdot y_i\right), \\
w_i^c &= \cos\left(\theta_i^c\right), \\
t &= \sqrt{\cos^2\left(\theta_i^c\right)\cos^2\left(\Delta_\theta\right) + \sin^2\left(\theta_i^c\right)}, \\
w_{i,\min} &= \cos^2\left(\theta_i^c\right)\cos\left(\phi_{\min}\right) + \sin^2\left(\theta_i^c\right), \\
t_{\min} &= \sqrt{\cos^2\left(\theta_i^c\right)\cos^2\left(\phi_{\min}\right) + \sin^2\left(\theta_i^c\right)}.
\end{aligned}
\tag{328}
$$

and equality is attained if and only if there exists a configuration of $(X, Y)$ such that:

(B6) $P_C \cdot x_i = P_C \cdot y_i$.

(B7) $\Delta_\theta = \phi_{\min}$.

$\square$

### K.4.3. PROOFS COROLLARY 1 AND 2

In this subsection, we provide the proofs of Corollary 1 and Corollary 2. Note that these corollaries all follow the conditions described in Theorem 3 and Theorem 4. For convenience in reading, we restate Corollary 1 and 2 before the proofs.

**Corollary 1.** $\forall i \in [N], i \neq c$, if $x_i \in \mathbb{S}_X, y_i \in \mathbb{S}_Y, x_i, y_i \notin \mathbb{C}, P_C x_i = P_C y_i$ and $\phi > 0$, it holds that:

(A8) $(x_i, y_i)_{i \neq c}$ can be mis-matched.

*Proof.* $\forall (x_i, y_i)_{i \neq c}$, let $w_i = x_i \cdot y_i$. We prove this corollary by showing that when the projections of $(x_i, y_i)$ are optimized to align, $w_i$ is not maximized and thus $(x_i, y_i)$ can be mismatched.

According to Lemma 12, for a given $x_i$, $w_i$ is maximized if and only if:

(i) $y_i = \frac{P_B \cdot x_i}{\|P_B \cdot x_i\|}$.

Similarly, for a given $y_i$, $w_i$ is maximized if and only if:

(ii) $x_i = \frac{P_A \cdot y_i}{\|P_A \cdot y_i\|}$.

According to Lemma 13, when $\phi > 0$, $x_i, y_i \not\perp \mathbb{C}$ and $x_i, y_i \notin \mathbb{C}$, Conditions (i) and (ii) cannot be satisfied simultaneously. Moreover, when the projections of $(x_i, y_i)$ are optimized to align, i.e., $P_C x_i = P_C y_i$, neither condition (i) or (ii) can be satisfied:

$$
\begin{aligned}
y_i &\neq \frac{P_B \cdot x_i}{\|P_B \cdot x_i\|}, \\
x_i &\neq \frac{P_A \cdot y_i}{\|P_A \cdot y_i\|}.
\end{aligned}
\tag{329}
$$

Therefore, $(x_i, y_i)_{i \neq c}$ can be mis-matched

$\square$

**Corollary 2.** $\forall i \in [N], x_i \in \mathbb{S}_X^{h_x}$ and $y_i \in \mathbb{S}_Y^{h_y}$, if $P_C x_i = P_C y_i$, then the following holds:

(A9) $\frac{P_C x_i}{\|P_C x_i\|} = \frac{P_C y_i}{\|P_C y_i\|}$

*Proof.* Denote:

$$
\begin{aligned}
x_i^{**} &= \frac{P_C x_i}{\|P_C x_i\|}, \\
y_i^{**} &= \frac{P_C y_i}{\|P_C y_i\|}.
\end{aligned}
\tag{330}
$$

Since $P_C x_i = P_C y_i$, then $x_i^{**} = y_i^{**}$.

$\square$

### K.4.4. TECHNICAL LEMMAS PART 4

In this subsection, we provide details and proofs of technical lemmas (Lemma 14 and Lemma 15) that support the proof of Theorem 4, Theorem S7 and Theorem S8.

**Lemma 14.** $\forall \kappa, \nu, \tau > 0$, a function $\bar{\mathcal{J}}(\cdot; \kappa, \nu) : (0, 1] \to \mathbb{R}$ is defined as:

$$
\bar{\mathcal{J}}(w_c; \kappa, \nu) = \tilde{\mathcal{J}}\left(f_1(w_c), \cos(\theta_i^c), f_2(w_c); \kappa, \tilde{\nu}\right),
\tag{331}
$$

where $f_1(\cdot) : (0, 1] \to \mathbb{R}_0^+$ and $f_2(\cdot) : [0, 1] \to \mathbb{R}_0^+$ are defined as:

$$
\begin{aligned}
f_1(w_c) &:= \cos^2(\theta_i^c) w_c + \sin^2(\theta_i^c), \\
f_2(w_c) &:= \sqrt{\cos^2(\theta_i^c) w_c^2 + \sin^2(\theta_i^c)}.
\end{aligned}
\tag{332}
$$

and $\tilde{\mathcal{J}}(\cdot, \cdot, \cdot; \kappa, \nu) : [-1, 1] \times [-1, 1] \times [0, 1] \to \mathbb{R}$ is defined as:

$$
\tilde{\mathcal{J}}(w_1, w_2, t; \kappa, \nu) = -\frac{w_1}{\tau} + \log\left(\frac{I_\nu\left(\tilde{M}_\kappa(w_2, t)\right)}{\tilde{M}_\kappa(w_2, t)^\nu}\right) - \log\left(\frac{I_\nu(\kappa)}{\kappa^\nu}\right),
\tag{333}
$$

and $\tilde{M}_\kappa(\cdot, \cdot) : [-1, 1] \times [0, 1] \to \mathbb{R}_0^+$ is defined as:

$$
\tilde{M}_\kappa(w, t) = \sqrt{\kappa^2 + \frac{2\kappa w}{\tau} + \frac{t^2}{\tau^2}},
\tag{334}
$$

and $I_\nu$ is the modified Bessel function of the first kind of order $\nu$, which is defined as:

$$
I_\nu(m) = \sum_{k=0}^{\infty} \frac{1}{k!\Gamma(\nu + k + 1)} \left(\frac{m}{2}\right)^{2k+\nu},
\tag{335}
$$

It holds that, for any fixed $\theta_i^c \in [0, \frac{\pi}{2}]$, $\bar{\mathcal{J}}(\cdot)$ is a strictly decreasing function on $(0, 1]$.

*Proof.* Let us first decompose the function $\mathcal{J}$. Denote a constant and a function $C_1$ and $G_2(t)$ as:

$$G_1(w_c) = -\frac{\cos^2(\theta_i^c) w_c}{\tau},$$
$$G_3(m) = \log(I_\nu(m)) - \nu \log(m),$$
$$G_2(w_c) = G_3\left(\tilde{M}_\kappa(\cos(\theta_i^c), f_2(w_c))\right)$$
$$= \log\left(I_\nu\left(\tilde{M}_\kappa(\cos(\theta_i^c), f_2(w_c))\right)\right) - \nu \log\left(\tilde{M}_\kappa(\cos(\theta_i^c), f_2(w_c))\right). \tag{336}$$

Denote the function $G(w_c)$ and the constant $C$ as:

$$G(w_c) = G_2(w_c) + G_2(w_c),$$
$$C = -\log\left(\frac{I_\nu(\kappa)}{\kappa^\nu}\right). \tag{337}$$

Then the function $\bar{\mathcal{J}}$ can be written as:

$$\bar{\mathcal{J}}(w_c; \kappa, \nu) = -\frac{\cos^2(\theta_i^c) w_c}{\tau} + \log\left(\frac{I_\nu\left(\tilde{M}_\kappa(\cos(\theta_i^c), f_2(w_c))\right)}{\tilde{M}_\kappa(\cos(\theta_i^c), f_2(w_c))^\nu}\right) - \log\left(\frac{I_\nu(\kappa)}{\kappa^\nu}\right) \tag{338}$$
$$= G(w_c) + C.$$

Now, we investigate derivatives of $G(w_c)$.

The first derivative of $G_1$ is:

$$G_1'(w_c) = -\frac{\cos^2(\theta_i^c)}{\tau} < 0. \tag{339}$$

According to Lemma 7, the first derivative of $G_3(m)$ is:

$$G_3'(m) = \frac{I_{\nu+1}(m)}{I_\nu(m)} \in (0, 1). \tag{340}$$

The derivative of $\tilde{M}_\kappa$ with respect to is $f_2^2(w_c)$:

$$\tilde{M}_\kappa'(\cos(\theta_i^c), f_2(w_c)) = \frac{\partial}{\partial f_2^2(w_c)} \tilde{M}_\kappa(\cos(\theta_i^c), f_2(w_c))$$
$$= \frac{\partial}{\partial f_2^2(w_c)} \left(\kappa^2 + \frac{2\kappa \cos(\theta_i^c)}{\tau} + \frac{f_2^2(w_c)}{\tau^2}\right)^{1/2}$$
$$= \frac{1}{2}\left(\kappa^2 + \frac{2\kappa \cos(\theta_i^c)}{\tau} + \frac{f_2^2(w_c)}{\tau^2}\right)^{-1/2} \cdot \frac{1}{\tau^2} \tag{341}$$
$$= \frac{1}{2\tau^2} \frac{1}{\tilde{M}_\kappa(\cos(\theta_i^c), f_2(w_c))}$$
$$> 0.$$

The derivative of $f_2^2$ is:

$$f_2^{2\prime}(w_c) = \frac{d}{dw_c}\left(\cos^2(\theta_i^c) w_c^2 + \sin^2(\theta_i^c)\right)$$
$$= 2\cos^2(\theta_i^c) w_c \tag{342}$$
$$\geq 0.$$

Let $m = \tilde{M}_\kappa \left( \cos \left( \theta_i^c \right), f_2 \left( w_c \right) \right)$. Then, the first derivative of $G_2$ is:

$$
\begin{aligned}
G_2' \left( w_c \right) &= G_3' \left( m \right) \tilde{M}_\kappa' \left( \cos \left( \theta_i^c \right), f_2 \left( w_c \right) \right) f_2^{2\prime} \left( w_c \right) \\
&= \frac{I_{\nu+1} \left( m \right)}{I_\nu \left( m \right)} \frac{1}{2\tau^2 m} 2 \cos^2 \left( \theta_i^c \right) w_c \\
&= \frac{\cos^2 \left( \theta_i^c \right) w_c}{\tau^2} \frac{1}{m} \frac{I_{\nu+1} \left( m \right)}{I_\nu \left( m \right)} \\
&> 0.
\end{aligned}
\tag{343}
$$

Combining Eq. (339) and Eq. (343), we have:

$$
\begin{aligned}
\bar{\mathcal{J}}' \left( w_c; \kappa, \nu \right) = G' \left( w_c \right) &= G_1' \left( t \right) + G_2' \left( t \right) \\
&= \frac{\cos^2 \left( \theta_i^c \right)}{\tau} \left( -1 + \frac{w_c}{\tau} \frac{1}{m} \frac{I_{\nu+1} \left( m \right)}{I_\nu \left( m \right)} \right).
\end{aligned}
\tag{344}
$$

Since $0 < w_c < 1$, then:

$$
\begin{aligned}
0 \leq w_c^2 \leq 1 &\Leftrightarrow \sin^2 \left( \theta_i^c \right) \geq \sin^2 \left( \theta_i^c \right) w_c^2 \\
&\Leftrightarrow \sin^2 \left( \theta_i^c \right) \geq w_c^2 - \cos^2 \left( \theta_i^c \right) w_c^2 \\
&\Leftrightarrow \cos^2 \left( \theta_i^c \right) w_c^2 + \sin^2 \left( \theta_i^c \right) \geq w_c^2 \\
&\Leftrightarrow f_2^2 (w_c) \geq w_c^2.
\end{aligned}
\tag{345}
$$

Therefore, consider $\theta_i^c \in [0, \frac{\pi}{2}]$, we have:

$$
\begin{aligned}
m^2 &= \tilde{M}_\kappa^2 \left( \cos \left( \theta_i^c \right), f_2 \left( w_c \right) \right) \\
&= \kappa^2 + \frac{2\kappa \cos \left( \theta_i^c \right)}{\tau} + \frac{f_2^2 (w_c)}{\tau^2} \\
&\geq \kappa^2 + \frac{2\kappa \cos \left( \theta_i^c \right)}{\tau} + \frac{w_c^2}{\tau^2} \\
&\geq \frac{w_c^2}{\tau^2} \\
&\geq 0,
\end{aligned}
\tag{346}
$$

which implies:

$$
m \geq \frac{w_c}{\tau} \Leftrightarrow \frac{w_c}{\tau} \frac{1}{m} \leq 1.
\tag{347}
$$

Plugging Eq. (340) and Eq. (347) into Eq. (344), we have:

$$
\begin{aligned}
\bar{\mathcal{J}} \left( w_c; \kappa, \nu \right) &= \frac{\cos^2 \left( \theta_i^c \right)}{\tau} \left( -1 + \frac{w_c}{\tau} \frac{1}{m} \frac{I_{\nu+1} \left( m \right)}{I_\nu \left( m \right)} \right) \\
&< 0.
\end{aligned}
\tag{348}
$$

So we can conclude that, for any fixed $\theta_i^c \in [0, \frac{\pi}{2}]$, $\bar{\mathcal{J}} \left( \cdot \right)$ is a strictly decreasing function on $(0, 1]$. $\qquad \square$

**Lemma 15.** *Let $X$ be an $N$-point configuration, where $X = (x_1, \ldots, x_N) \in (\mathbb{S}^{h-1})^N$ are iid samples from $\mu = \text{vMF}(c, \kappa)$. When $\kappa$ is sufficiently large, $\forall i, j \in [K], i \neq j$, it holds that:*

$$
P(x_i \cdot x_j \geq 0) \approx 1.
\tag{349}
$$

*Proof.* Let $X \sim \text{vMF}(c, \kappa)$ on $\mathbb{S}^{h-1}$ and set $U = c^\top X = \cos \Theta \in [-1, 1]$. Then:

$$P(X \cdot c \geq 0) = \frac{\int_0^1 e^{\kappa u} \left(1 - u^2\right)^{\frac{p-3}{2}} du}{\int_{-1}^1 e^{\kappa u} \left(1 - u^2\right)^{\frac{p-3}{2}} du}. \tag{350}$$

Using standard integral representations of the modified Bessel and modified Struve functions,

$$
\begin{aligned}
I_\nu(z) &= \frac{(z/2)^\nu}{\sqrt{\pi}\Gamma\left(\nu + \frac{1}{2}\right)} \int_{-1}^1 e^{zt} \left(1 - t^2\right)^{\nu - \frac{1}{2}} dt, \\
\nu(z) &= \frac{(z/2)^\nu}{\sqrt{\pi}\Gamma\left(\nu + \frac{1}{2}\right)} \int_0^1 2\sinh(zt) \left(1 - t^2\right)^{\nu - \frac{1}{2}} dt,
\end{aligned}
\tag{351}
$$

with $\nu = h/2 - 1$, the ratio simplifies to the neat closed form

$$P(X \cdot c \geq 0) = \frac{1}{2} \left(1 + \frac{L_\nu(\kappa)}{I_\nu}\right) \tag{352}$$

where $L_\nu$ the modified Struve function. And we list numerical values of this probability:

- $h = 128$:

| $\kappa$ | 1 | 5 | 10 | 20 | 30 | 50 | 100 | 200 |
|---|---|---|---|---|---|---|---|---|
| P | 0.5353 | 0.6710 | 0.8117 | 0.9609 | 0.9956 | 1.0000 | 1.0000 | 1.0000 |

- $h = 512$:

| $\kappa$ | 1 | 5 | 10 | 20 | 30 | 50 | 100 | 200 |
|---|---|---|---|---|---|---|---|---|
| P | 0.5176 | 0.5875 | 0.6708 | 0.8116 | 0.9075 | 0.9863 | 1.0000 | 1.0000 |

- $h = 1024$:

| $\kappa$ | 1 | 5 | 10 | 20 | 30 | 50 | 100 | 200 |
|---|---|---|---|---|---|---|---|---|
| P | 0.5125 | 0.5621 | 0.6227 | 0.7340 | 0.8258 | 0.9409 | 0.9991 | 1.0000 |

$\square$

