# OpenReview forum: "The Convergent Representation of Contrastive Vision-Language Models: Geometry, Modality Gap and Shared Space Alignment"
_ICML.cc/2026/Conference — ICML 2026 regular_

### Official Review · Reviewer_Gcru · 2026-02-21

**Soundness:** 2
**Presentation:** 2
**Significance:** 3
**Originality:** 3
**Overall Recommendation:** 4
**Confidence:** 4

**Summary:**

This paper investigates the widely observed modality gap problem in vision-language contrastive learning. The authors propose a theoretical framework regarding the convergent optimal representation when training reaches optimality, attempting to answer two key questions: (1) what causes the modality gap, and (2) what determines downstream performance. The core conclusion is that, under unconstrained conditions, the theoretical optimum of Multimodal Contrastive Learning (MCL) perfectly aligns the two modalities. However, in practice, dimension collapse causes the two modalities to fall into distinct but partially overlapping subspaces, thereby generating the modality gap. Although the modalities as a whole cannot be directly aligned, their projections onto the shared subspace can still be aligned, and the degree of this shared space alignment plays a dominant role in downstream performance. Based on these findings, the authors propose incorporating Shared Space Alignment (SSA) during pre-training to explicitly enhance the alignment within the shared space.

**Compliance With Llm Reviewing Policy:**

Affirmed.

**Final Justification:**

(1) I keep my final recommendation unchanged. The paper has a clear and original theoretical contribution: it argues that dimension collapse, rather than the cone effect, underlies the modality gap, and that shared-space alignment is more relevant to downstream performance.

(2) However, the current version remains weaker on empirical completeness and clarity than I would want for a higher score. The experimental evidence is still somewhat narrow relative to the strength of the claims.

(3) The rebuttal clarifies some presentation issues, but only partially addresses my main concerns regarding broader validation, stronger ablations, and practical efficiency. As a result, the rebuttal does not materially change my assessment.

**Key Questions For Authors:**

1. Since computing the projection matrix requires frequent and time-consuming SVD operations, could the authors quantitatively analyze the additional computational overhead (e.g., training time, GPU memory) introduced by the SSA loss? A detailed comparative analysis against the baseline would help readers better understand the method's practical efficiency.
2. How sensitive is the downstream performance to the threshold used for determining the overlapping dimension and the weight assigned to the SSA loss? It is recommended to include ablation studies analyzing these key hyperparameters to demonstrate the robustness of the method.
3. While the theoretical framework attributes the modality gap to dimension collapse, did the authors observe any empirical correlations between the severity of this collapse and specific training factors during the pre-training process?
4. The paper heavily emphasizes mathematical derivations and final geometric properties, but leaves a gap in analyzing training dynamics from an engineering perspective (e.g., convergence speed, training stability). Supplementing this content would provide a much more comprehensive view of the proposed method.

**Limitations:**

yes

**Strengths And Weaknesses:**

Strengths:
1. By modeling the contradiction between the contrastive objective's theoretical tendency to eliminate the modality gap and its persistent empirical presence, this work attributes the gap's root cause to dimension collapse through the lens of convergent optimal representation.
2. A key insight is that while the gap hinders global alignment, sample pairs can still align within the shared space projection. The analysis clarifies that pair matching is jointly determined by shared space alignment and the gap, with the former being empirically dominant.
3. To bridge theory and practice, the theoretical findings are translated into actionable metrics by introducing the shared-space alignment measure, conducting correlation analyses, and proposing the SSA pre-training objective.

Weaknesses:
1. There is a duplicated baseline method name (IMSep) in Table 4, and the paper's layout is somewhat confusing due to the presence of two separate "Experiments" sections. Additionally, an unresolved cross-reference appears in Appendix A ("...Intra-Modal Isometry of the representations (see ??), ...").
2. The empirical evaluation lacks broader task diversity (e.g., fine-grained classification, VQA/grounding, out-of-distribution tasks) and essential ablation studies (e.g., SSA loss weight, projection dimension threshold, and prompt dependency).
3. The mathematical derivations are overly verbose. For instance, the proofs for Lemmas 7 through 9 could simply cite existing literature instead of dedicating extensive pages to from-scratch derivations.

---

> ### Author Rebuttal · Authors · 2026-03-31
>
> # Response to Reviewer Gcru
>
> ## Weakness 1
>
> We thank the reviewer for identifying these presentation issues. Unfortunately, the submitted version was accidentally overwritten by an earlier draft. We have now corrected **Table 4** (see our response to Reviewer jCys’s Weakness 1). In the revised version, we will also rename the sections for clarity: we will change **“Experiment”** in Sec. 3.3 to **“Evaluation of Unconstrained Convergence”**, and **“Experiment”** in Sec. 4.3 to **“Evaluation of Constrained Convergence.”** In addition, we will include a definition of **Intra-Modal Isometry**, following the main intuition of the **Platonic Representation Hypothesis** [1].
>
> **[1]** *The Platonic Representation Hypothesis*, Huh et al., 2025.
>
> ## Weakness 2 and Question 2
>
> We appreciate this suggestion and agree that broader empirical coverage would strengthen the paper. Our experimental design is intended to validate the paper’s two central questions:
>
> - What causes the modality gap?
> - What determines downstream performance?
>
> For this reason, we focus on tasks that are most directly tied to the geometry of multimodal contrastive representations, namely **zero-shot classification** and **cross-modal retrieval**. These settings allow us to directly quantify pair matching and shared-space alignment, and therefore provide the most faithful testbed for our theory.
>
> We have already added additional evaluations on multiple benchmarks (see our response to Reviewer jCys’s Weakness 1) as well as direct evidence for the SSA mechanism (see our response to Reviewer jCys’s Weakness 2).
>
> We will clarify that our results indicate downstream performance is relatively **insensitive to the prompt choice**.
>
> We also provide ablation studies for the projection thresholds below. These studies are conducted using **SSP on MSCOCO** with a **ViT-L/14** backbone pretrained on OpenAI’s WebImageText dataset.
>
> First, we fix $c_x = c_y \in \{0.1, 0.05, 0.01, 0.001, 0.0001\}$ and vary $\epsilon \in \{0.5, 0.4, 0.3, 0.2, 0.1, 0.05, 0.01, 0.001, 0.0001\}$. We then evaluate **R@1** for zero-shot cross-modal retrieval on MSCOCO. The results suggest that **$\epsilon = 0.1$** is a good choice.
>
> Second, we fix $\epsilon = 0.1$ and vary $c_x$ and $c_y$ in $\{0.1, 0.05, 0.01, 0.001, 0.0001\}$. We again evaluate **R@1** on zero-shot cross-modal retrieval. The results indicate that **$c_x = c_y = 0.1$** is a good choice.
>
> Due to the length limit, we are unable to paste the full table.
>
> We agree that additional empirical analyses would improve the completeness of the paper. In the revised version, we will further add ablations on the **SSA loss weight** and evaluations on **broader downstream tasks**.
>
> ## Weakness 3
>
> We appreciate this feedback and agree that the current proof presentation can be streamlined.
>
> ## Question 1
>
> We agree that the practical efficiency of SSA should be discussed more explicitly.
>
> The main additional cost of SSA comes from computing the **shared-subspace projection**, which requires an SVD on the mini-batch feature matrices. However, this operation is performed in the **embedding space**, rather than over the full model parameters, so its cost is modest compared with the forward and backward passes of the vision and text encoders. In practice, SSA is introduced as a lightweight auxiliary loss on top of standard contrastive pretraining.
>
> In the revised version, we will add a quantitative efficiency analysis, including:
>
> - training-time overhead relative to the baseline,
> - GPU memory usage, and
> - a brief discussion of how the SVD cost scales with batch size and feature dimension.
>
> ## Question 3
>
> We thank the reviewer for this insightful question. Our current paper focuses on the **representational consequence** of pretraining: namely, that the modality gap emerges when the two modalities collapse into different subspaces. We agree that connecting the severity of collapse to specific training factors would provide an important complementary perspective.
>
> A full causal study of pretraining factors is beyond the scope of the current submission, but we will strengthen the discussion in the revised version and make this limitation more explicit.
>
> ## Question 4
>
> We agree that a training-dynamics perspective would enrich the paper.
>
> Our current emphasis is on the **convergent geometry** of multimodal contrastive learning: what representation structure is favored by the objective at optimality, why the modality gap persists in practice, and how this affects downstream pair matching. In this sense, the current work is primarily a **representation-level geometric analysis**, rather than a study of training dynamics.
>
> That said, we agree that discussing optimization behavior would improve the paper’s completeness. In the revised version, we will add a discussion of training behavior, including observations on **convergence** and **stability** for SSA, and clarify that a full dynamical analysis is an important direction for future work.

---

> > ### Author Rebuttal · Reviewer_Gcru · 2026-04-02
> >
> > (1) The rebuttal addresses some of my presentation-related concerns, including the duplicated method name / section naming issues and the missing cross-reference.
> >
> > (2) However, my main concerns are only partially addressed: the rebuttal does not provide enough new evidence on broader empirical validation, quantitative efficiency analysis, or sufficiently complete ablations on key SSA design choices.
> >
> > (3) The discussion on training dynamics / optimization behavior is also still limited, and remains mostly a clarification of scope rather than a substantive resolution.
> >
> > (4) Therefore, my concerns are only partially resolved, and I keep my score unchanged.

---

> > > ### Author Response · Authors · 2026-04-08
> > >
> > > We sincerely thank the reviewer for the thoughtful feedback and are glad that our rebuttal has partial addressed your concerns. The remaining concerns, which cannot be fully resolved in a rebuttal with limited length, are also gratefully appreciated. We will include additional discussion regarding broader empirical validation, quantitative efficiency analysis and training dynamics / optimization behavior in the revision.

---

### Official Review · Reviewer_y9Ra · 2026-03-09

**Soundness:** 3
**Presentation:** 3
**Significance:** 3
**Originality:** 3
**Overall Recommendation:** 4
**Confidence:** 4

**Summary:**

This paper investigates two questions about multimodal contrastive learning (MCL): what causes the modality gap between image and text representations, and how does the gap affect downstream task performance. The authors establish a theoretical framework based on the von Mises-Fisher (vMF) distribution assumption, proving a series of theorems characterizing the Convergent Optimal Representation (COR) of MCL under progressively constrained settings. Their central theoretical claims are (1) that the MCL objective actually closes the modality gap at its optimum (Theorem 1), (2) that the cone effect is not the cause of the persistent gap (Theorem 2, Conclusion 2), and (3) that dimension collapse — in which image and text embeddings collapse into partially overlapping but distinct subspaces — is the true origin of the gap (Theorem 3). In the second half, the paper argues that downstream performance is jointly determined by shared-space alignment and the modality gap, with the former playing a dominant role (Conclusion 6), and proposes a Shared Space Projection (SSP) method and a Shared Space Alignment (SSA) training objective to exploit this finding. The empirical analysis spans 78 pretrained VLMs across diverse architectures and datasets.

**Compliance With Llm Reviewing Policy:**

Affirmed.

**Final Justification:**

The authors did a good job in responding and clarifying the issues to me and other reviewers so in light of this I am raising my score. I hope the authors address this in the final revision.

**Key Questions For Authors:**

Q1. Can you clarify the motivation for the $-2\log(N)$ normalization in the MCL divergence? Specifically, which prior work introduces this normalization, and what is its information-theoretic or geometric interpretation in the context of contrastive losses on $\mathbb{S}^{h-1}$?

Q2. Theorem 2 assumes the existence of an anchor sample at the vMF center. How sensitive is Conclusion 2 to violations of this condition? In particular, if the nearest sample to $c_x$ is at angular distance $\epsilon$ from the center, does the modality gap converge to a quantity bounded by $f(\epsilon)$ rather than exactly zero? A perturbation analysis would substantially strengthen the ruling out of the cone effect.

Q3. What is the rank of the shared subspace $\mathbb{C}$ relative to the ambient dimension $h$ for the 78 examined models? Figure 3(d) shows many zero principal angles, but does not report the dimensionality of $\mathbb{C}$ directly. If $\dim(\mathbb{C})$ is close to $h-1$, the injectivity concern raised in W4 is severe and should be addressed explicitly.

Q4. Does SSP change $\text{Align}(X^, Y^)$ relative to $\text{Align}(X, Y)$ before projection? Reporting this quantity would allow the reader to determine whether the flat performance in Tables 2 and 3 is due to the gap being irrelevant or due to SSP degrading shared-space alignment.

Q5. The paper claims dimension collapse causes the modality gap. Have you conducted any experiment in which you train models with explicit rank regularization or spectral whitening to reduce dimension collapse, and measured the resulting modality gap? Such an intervention experiment would be the most direct test of the causal claim in Conclusion 3.

Q6. For VLMs where $N$ is large relative to $\dim(\mathbb{C})$, do you observe fibre collisions — i.e., multiple image embeddings projecting to approximately the same point in $\mathbb{C}$? If so, does this correlate with degraded pair matching quality in $\text{Align}(X, Y)$?

Q7. Table 4 contains two rows both labelled "IMSep" with substantially different performance numbers. What does the second row represent? If it corresponds to a different method or a stronger variant of IMSep, please correct the label and revise the discussion of SSP's advantage accordingly. Given that CLIP + SSP (71.3 R1 on CIFAR-10) only marginally exceeds the second IMSep row (69.4 R1), how do the authors characterize the practical contribution of SSP relative to this baseline?

Q8. The theoretical analysis in Sections 3.5 and 4.1 assumes both subspaces are $(h-1)$-dimensional hyperplanes. Section 4.3 acknowledges this is a simplification, but the bridge to practice relies purely on the empirical evidence of Figures 3(c) and 3(d). Can the authors provide a more general version of Theorems 3 and 4 that does not require the hyperplane simplification, or at least a discussion of how the conclusions change when the subspaces have unequal or variable dimensionality?

Table 4 has two rows labelled "IMSep" — one of these is almost certainly a labelling error and must be corrected before publication.
The paper uses "COR" (Convergent Optimal Representation) extensively but introduces it without a self-contained definition in the main text; a one-sentence definition upon first use would improve readability.
Figure 2(b) shows $\kappa_x$ vs $\kappa_y$ with a dashed diagonal but does not discuss the scatter above and below the diagonal, which could indicate systematic differences between image and text encoder concentration that are potentially informative.
The SSA loss in Eq. (11) is asymmetric in its written form but appears symmetric by construction; the authors should confirm and clarify.
Eq. (9) defines $y_i^* = Px_i / |Py_i|$, which appears to be a typo — the numerator should likely be $Py_i$.

**Limitations:**

no. But this paper doest need a limitation section too.

**Strengths And Weaknesses:**

### Strengths
S1. Theoretically grounded decomposition of the modality gap. The paper is one of the first to provide a geometric account of why the modality gap persists under MCL training. The decomposition into constraints  leading to Theorems 2 and 3, is conceptually clean and the two-part structure is well-motivated.
S2. Theorem 4 and Corollaries 1–2 provide a principled bridge to downstream performance. The identification that shared-space alignment is the critical condition for pair matching, and that the modality gap introduces residual mis-matching even when alignment holds, is an elegant and practically useful decomposition. Corollary 2 in particular gives a clean sufficiency condition for well-matchedness that connects directly to the SSP method.

S3. Large-scale empirical validation. Analyzing 78 contrastive pretrained VLMs across diverse settings — architectures, activation functions, training configurations, and datasets — gives the empirical findings considerably more weight than the small-NN
N numerical examples that characterize most prior work in this area

S4. Figure 3 provides compelling visual evidence for dimension collapse. The combination of the UMAP plot , singular value decay and principal angle analysis provides  thorough empirical characterization of the subspace constraint.

S5. SSA training objective is practically motivated.** The proposal to directly train on projections (X∗,Y∗)(X^*, Y^*)
(X∗,Y∗) onto the shared space via LSSA\mathcal{L}_{\text{SSA}}
LSSA​ is a natural consequence of the theory, and the experimental results showing improved classification performance are a useful proof of concept.


### Weaknesses
W1. The MCL divergence normalization is unmotivated.
The normalized quantity $\lim_{N\to\infty} \mathcal{L}_{\text{MCL}} - 2\log(N)$ appears centrally in all four main theorems without explanation. The raw MCL loss grows as $O(\log N)$ because the softmax denominator sums over $N$ terms, making the unnormalized loss scale-dependent and ill-suited for asymptotic characterization. Subtracting $2\log(N)$ produces a well-defined population limit, but the paper provides no motivation for why this is the canonical normalization or what it corresponds to conceptually. A reader unfamiliar with the InfoNCE literature will be unable to follow the theoretical development from Theorem 1 onward. A brief remark clarifying that this corresponds to measuring excess loss relative to a uniform negative baseline — or citing the specific prior work from which this normalization is adopted — is necessary before Theorem 1.

W2. The causal claim that dimension collapse causes the modality gap is not supported by intervention evidence.

Theorem 3 establishes a conditional result: given the subspace constraint, the modality gap at the COR converges to $\phi_{\min}$. The empirical evidence in Figures 3(c) and 3(d) demonstrates that the subspace constraint is approximately active in practice. However, the paper then concludes that dimension collapse is the "true origin" of the modality gap — a causal claim that the conditional theorem and correlational evidence do not support. What is needed is either (a) a controlled training experiment in which rank regularization or whitening is applied to reduce dimension collapse, with a corresponding measurement of the modality gap before and after, or (b) a formal lower bound expressing the gap as a monotone function of the encoder's rank deficiency. Without either, Conclusion 3 (that dimension collapse is the fundamental origin of the gap) is a well-motivated conjecture rather than an established causal claim. The paper's abstract and conclusion section should be revised to reflect this distinction.

W3. Theorem 2 rests on an unverified and empirically strong anchor-sample condition.

Theorem 2 assumes the existence of a sample index $i=c$ such that $x_c = c_x$ and $y_c = c_y$, i.e., that a data point is located exactly at the vMF center of each modality. This is a population-level quantity that is almost surely not realized by any finite sample. The paper provides no verification that this condition holds approximately across the 78 examined models, and no sensitivity analysis of Conclusion 2 with respect to violations. The paper points to Theorem 4 to justify that non-center pairs do not affect the configuration of $(c_x, c_y)$, but Theorem 4 is proved under the assumption that the center pair configuration is already fixed — creating a subtle circularity. The ruling out of the cone effect is therefore less definitive than Conclusion 2 presents it to be.

W4. The shared subspace argument implicitly requires injectivity of $P_C$ that is undermined by the paper's own geometry.

Theorem 4 and Corollary 2 guarantee pair well-matchedness under shared-space alignment, but this guarantee is only meaningful for downstream pair matching if the projection $P_C$ is effectively injective on the data — i.e., no two distinct data points in $\mathbb{S}_X$ share the same projection onto $\mathbb{C}$. Under the subspace constraint, $P_C$ maps from an $(h-1)$-dimensional sphere onto an $(h-2)$-dimensional shared subspace. This projection is structurally many-to-one: it forms one-dimensional arcs on $\mathbb{S}_X$ and $\mathbb{S}_Y$. Multiple points with different positions in $\mathbb{S}_X$ can map to the same point in $\mathbb{C}$, meaning Theorem 4's condition $P_C x_i = P_C y_i$ does not guarantee that the original one-to-one pairing is recoverable — the modality-specific components orthogonal to $\mathbb{C}$ remain completely unconstrained.

This problem is made more acute by Figure 3(d), which shows that many principal angles between $\mathbb{S}_X$ and $\mathbb{S}_Y$ are zero, meaning $\mathbb{C}$ is large relative to the ambient dimension. A large $\mathbb{C}$ makes the projection more aggressive, compresses more variance into the shared dimensions, and enlarges the fibres — worsening the injectivity problem rather than ameliorating it. In high-dimensional VLM embedding spaces (typically $h=512$ or $768$), the shared subspace can be nearly as large as the full ambient space, leaving very little modality-specific variance to discriminate between pairs within a fibre. In this regime, many $x_i$ may project to approximately the same point in $\mathbb{C}$, creating near-ties in cosine similarity and making the well-matchedness criterion of Definition 3 unreliable due to sensitivity to numerical noise.

This injectivity condition is precisely analogous to the full-rank Gram matrix assumption required in kernelised point cloud matching methods (e.g., kernel Wasserstein, Gaussian process correspondence), where it is explicitly stated or verified. The paper makes no such  statement, and the implicit assumption is violated by the very geometry the paper identifies as characteristic of pretrained VLMs.

W5. The one-to-one pairing assumption is incompatible with the induced many-to-one projection geometry.

Related to W4, the paper assumes throughout that the dataset consists of $N$ paired samples with known ground-truth one-to-one correspondence. This is the structural assumption underlying Definition 3 and the well-matchedness criterion. However, the subspace geometry the paper identifies — in which $P_C$ induces many-to-one projections — is at odds with one-to-one matchability in $\mathbb{C}$. In the unsupervised correspondence recovery (kernelised matching, Wasserstein Procrustes), this is resolved by requiring a doubly stochastic or hard permutation matrix whose well-posedness depends on the injectivity of the underlying map. Here, the paper neither acknowledges this nor provides conditions under which one-to-one matchability in $\mathbb{C}$ is preserved. As $N$ grows relative to the intrinsic dimensionality of $\mathbb{C}$, the collision become increasingly likely, and the notion of a  best match in the shared space breaks down. The weak assumptions of the paper — vMF marginals, i.i.d. sampling, geometric constraints — are insufficient.

W6. The shared-space alignment metric is redundant as a predictor of downstream performance.
The metric $\text{Align}(X, Y)$ defined in Eq. (10) counts the proportion of well-matched pairs after projecting onto $\mathbb{C}$ and is used to demonstrate via Kendall's $\tau$ correlation that shared-space alignment is a strong predictor of downstream performance. However, $\text{Align}(X, Y)$ is structurally close to a direct measure of task performance — it counts correct correspondences in the shared space, which is approximately what zero-shot classification and retrieval metrics also measure. The strong correlation between $\text{Align}(X, Y)$ and downstream performance is therefore close to tautological and does not establish shared-space alignment as an independent causal variable. A more informative experiment would partial out the effect of overall representation quality by comparing models matched on raw retrieval performance but differing in their degree of shared-space alignment.

W7. The SSP experiment in Tables 2 and 3 confounds the two factors it is designed to disentangle.

SSP dramatically reduces $\Delta_\theta$ — from $74.69^\circ$ to $5.37^\circ$ on CIFAR-10 and from $78.16^\circ$ to $68.06^\circ$ on MSCOCO — while leaving downstream performance approximately flat or slightly degraded. The paper interprets this as confirming that the gap is irrelevant. However, the SSP transformation projects $(X, Y)$ onto $\mathbb{C}$ and renormalizes, which simultaneously reduces the gap and potentially changes the shared-space alignment of the projected pairs. Since the paper does not report $\text{Align}(X^, Y^)$ before and after SSP, it is impossible to determine whether the flat performance is due to the gap being genuinely irrelevant, or due to SSP inadvertently minimising shared-space alignment — for instance by increasing correspondence through the many-to-one mechanism described in W4 — in a way that offsets any benefit from gap reduction. Without either holding alignment fixed while varying the gap, or vice versa, Tables 2 and 3 cannot cleanly support Conclusion 6.

SSP dramatically reduces $\Delta_\theta$ — from 74.69° to 5.37° on CIFAR-10 and from 78.16° to 68.06° on MSCOCO — while leaving downstream performance approximately flat or slightly degraded. The paper interprets this as confirming that the gap is irrelevant. However, the SSP transformation projects $(X, Y)$ onto $\mathbb{C}$ and renormalizes, which simultaneously reduces the gap and potentially changes the shared-space alignment of the projected pairs. Since the paper does not report $\text{Align}(X^, Y^)$ before and after SSP, it is impossible to determine whether the flat performance is due to the gap being genuinely irrelevant, or due to SSP inadvertently harming shared-space alignment — for instance by increasing fibre collisions through the many-to-one mechanism described in W4 — in a way that offsets any benefit from gap reduction. Without either holding alignment fixed while varying the gap, or vice versa, Tables 2 and 3 cannot cleanly support Conclusion 6.

W8. The post-hoc comparison in Section 4.4 does not control for shared-space alignment across baselines.
The comparison between SSP, SharedCLIP, and IMSep shows that all methods produce similar downstream numbers despite SSP reducing the gap more aggressively. This is consistent with the paper's claim that the gap is unimportant, but equally consistent with all three methods similarly improving shared-space alignment while differing only in gap reduction. Without reporting $\text{Align}(X, Y)$ for each baseline, the attribution of performance to gap vs. alignment remains ambiguous.

W9. Table 4 contains a duplicate baseline label that undermines the credibility of the post-hoc comparison.
Table 4 lists two rows both labelled "IMSep" with substantially different numbers — the first reporting R1/R5 of 61.6/96.2 on CIFAR-10 and 31/60.1 on CIFAR-100, and the second reporting 69.4/97.8 and 36.5/66.3 on the same benchmarks. These are clearly not the same method run twice and possibly a technical error, as the second row consistently and substantially outperforms the first across all datasets and metrics. The most plausible explanation is that one of these rows is mislabelled — either a variant of IMSep with a different configuration, or an entirely different method whose label was lost during typesetting. This is a serious error: the reader has no way to identify what the second row represents or how to verify the comparison against prior work.

More substantively, the duplicate label has a direct impact on the paper's empirical claims. CLIP + SSP achieves R1/R5 of 71.3/97.8 on CIFAR-10, which only marginally exceeds the second (unlabelled) IMSep row at 69.4/97.8. If this second row represents a properly implemented strong baseline, the practical advantage of SSP is considerably smaller than the table implies, and the paper's claim that SSP meaningfully advances the state of the art in post-hoc gap reduction requires reassessment. The authors must identify and correctly label both rows, and if the second row corresponds to a stronger existing method, the discussion of SSP's contribution should be revised accordingly.

---

> ### Author Rebuttal · Authors · 2026-03-31
>
> # Response to Reviewer y9Ra
>
> ## W1 and Q1
>
> The loss contains two softmax normalizers, and each contributes an asymptotic $\log N$ term. Therefore, $\mathcal{L}_{\mathrm{MCL}} - 2\log N$ is the finite excess term that remains after removing the uniform-negative baseline. The same normalization is also adopted in [1].
>
> **[1]** *Understanding Contrastive Representation Learning through Alignment and Uniformity on the Hypersphere*, Wang and Isola, ICML 2020.
>
> ## W2 and Q5
>
> The theoretical result in **Theorem 3** is indeed conditional: **under the subspace constraint**, the COR induces a nonzero limiting modality gap determined by the subspace geometry. Intuitively, when $X$ and $Y$ collapse into different subspaces, the two modalities become structurally separated, and a modality gap naturally emerges. We agree, however, that the phrase **“true origin”** may overstate the evidence if interpreted as a fully established interventional causal claim. We will revise it.
>
> As for the intervention experiment, we agree that it is valuable. Due to rebuttal time limits, we may not be able to add a new training study. However, we will explicitly acknowledge this as an important future direction.
>
> ## W3 and Q2
>
> Our goal in Theorem 2 is to isolate the **cone-effect component** in the cleanest possible geometric setting, and to show that cone concentration alone does not induce a nonzero limiting modality gap. We agree that assuming an exact sample located at the vMF center is a strong simplification. Importantly, we use this assumption only to enable a closed-form geometric analysis; we do **not** require such a sample to exist in the actual training set.
>
> We also appreciate the reviewer’s perturbation suggestion. In the revision, we will add a discussion that if the nearest sample is at angular distance $\epsilon$ from the center, then the induced deviation in the center-pair argument is expected to vary continuously with $\epsilon$.
>
> ## W4, W5, Q3, Q6, and Q8
>
> We numerically evaluate the projected image and text representations for all **78 VLMs**. The minimal angle between the two sets of projected representations is always **greater than 10 degrees**, indicating that injectivity is not an issue in any evaluated model.
>
> We also compute the ratio between the rank of the shared space and the full embedding rank. Across all 78 VLMs, this ratio ranges from **0.15 to 0.6**. In the revised version, we will add the ranks analysis.
>
> Our theory is intentionally developed under a simplified setting in order to enable closed-form geometric analysis. The two main conclusions are that:  the **contrastive objective promotes shared-space alignment**, and  **shared-space alignment is a dominant factor in downstream performance**.
>
> We validate these conclusions empirically on all 78 real VLMs, whose subspace dimensions vary over a broad range. Please see our response to **Reviewer 4zXr, Weakness 1 and Question 1**.
>
> ## W3
>
> While both $\mathrm{SAlign}(X,Y)$ and downstream performance are related to matching quality, they are not the same quantity. $\mathrm{SAlign}(X,Y)$ is defined purely from the geometry of the projected representations in the shared space. If practice diverged from our theoretical prediction — namely, that the contrastive objective promotes shared space alignment — then the projection operation could substantially degrade pair matching quality, and one would not expect a strong correlation between $\mathrm{SAlign}(X,Y)$ and downstream performance.
>
> We agree that the paper should avoid overstating this correlation as proof of an independent causal variable. We will revise it.
>
> ## W7 and Q4
>
> To isolate the role of modality-gap reduction more cleanly, we modify the SSP experiment.
>
> For each sample pair $(x_i, y_i)$, we project them into the shared space as $(P_C x_i, P_C y_i)$, but **do not re-normalize** the projected representations. Under this setting, $(x_i, y_i)$ and $(P_C x_i, P_C y_i)$ have the same degree of $\mathrm{SAlign}(X,Y)$. Therefore, we can reduce the modality-gap size while keeping shared-space alignment unchanged. We then evaluate cross-modal retrieval using the **L2 distance** between $(P_C x_i, P_C y_i)$.
>
> ### SSP results without re-normalization
>
> | Model | CIFAR-10 $\Delta_{\theta}$ | R@1 | R@5 | CIFAR-100 $\Delta_{\theta}$ | R@1 | R@5 | ImageNet-1K $\Delta_{\theta}$ | R@1 | R@5 |
> |---|---:|---:|---:|---:|---:|---:|---:|---:|---:|
> | CLIP | 74.69$^\circ$ | 89.00 | 99.36 | 74.19$^\circ$ | 65.23 | 88.88 | 71.02$^\circ$ | 63.34 | 88.82 |
> | CLIP + SSP | 5.37$^\circ$ | 86.43 | 99.27 | 30.39$^\circ$ | 64.51 | 88.79 | 50.40$^\circ$ | 62.45 | 88.41 |
>
> These results support our claim that the **modality gap plays a limited role** in downstream performance.
>
> ## W8
>
> We provide $\mathrm{SAlign}(X,Y)$ results for all models. Please see our response to **Reviewer jCys, Weakness 2**.
>
> ## W9 and Q7
>
> We have now corrected **Table 4** (Please see our response to **Reviewer jCys, Weakness 1**).

---

> > ### Author Rebuttal · Reviewer_y9Ra · 2026-04-02
> >
> > I thank the authors and I have added further comments to be addressed.
> >
> > I thank the authors for their comment which (partially) resolves some aspects. You will find my comments
> >
> > W1
> >
> > The authors have provided the arithmetic sketch of the normalization but have not justified it as the canonical object of theoretical analysis, have not provided the asymptotic expansion that the theorems depend on, and have deflected to a paper with a different loss structure without showing the argument transfers. The revision must include a self-contained derivation of the asymptotic expansion of LMCL\mathcal{L}_{\text{MCL}} under the vMF assumption, either in the main text or in the appendix with a pointer from the main text. Without this, the theoretical foundation of all four main theorems remains opaque and remains the guess of reviewers to read Wang and Isola PRH paper
> >
> > W2
> >
> > The absence of an intervention experiment still remains a limitation, and the revision should acknowledge it explicitly as future work rather than quietly dropping it. This is actually still interesting so please keep this content rather than saying you will soften the language
> >
> > W3
> >
> > The authors point towards qualitative statement that the deviation "varies continuously with \epsilon " is not sufficient. Please show the actual bound in the revision.
> >
> > ###W6 The concern that \text{SAlign} SAlign and downstream performance measure approximately the same thing — is not directly answered. The response addresses robustness of the correlation but not its potential circularity. This concern should be softened but please dont withdraw it.

---

> > > ### Author Response · Authors · 2026-04-04
> > >
> > > ## Further W1
> > >
> > > In Theorem S1, S3, and S5, we already provide asymptotic decomposition underlying all four main theorems, including ones under $\mathrm{vMF}$ assumption, for each direction:
> > >
> > > $\mathcal{L}_{\mathcal{X} \rightarrow \mathcal{Y}}^{(N)}=\log N+\mathcal{L}_{\mathcal{X} \rightarrow \mathcal{Y}}^{\infty}+o(1)$
> > >
> > > Thus, the $\log N$ term is extracted exactly, rather than heuristically. Applying this decomposition to both directions yields the bidirectional expansion:
> > >
> > > $\mathcal{L}_{\mathrm{MCL}}^{(N)} = \mathcal{L}_{\mathcal{X} \rightarrow \mathcal{Y}}^{(N)} + \mathcal{L}_{\mathcal{Y} \rightarrow \mathcal{X}}^{(N)} =2 \log N+\mathcal{L}_{\mathcal{X} \rightarrow \mathcal{Y}}^{\infty}+\mathcal{L}_{\mathcal{Y} \rightarrow \mathcal{X}}^{\infty}+o(1)$
> > >
> > > We agree, however, that this argument should be stated explicitly as a standalone asymptotic expansion result. In the revision, we will add **a self-contained proposition, with proof in the appendix and a pointer from the main text**, showing that:
> > >
> > > $\mathcal{L}_{\mathrm{MCL}}^{(N)} =2 \log N+\mathcal{L}_{\mathcal{X} \rightarrow \mathcal{Y}}^{\infty}+\mathcal{L}_{\mathcal{Y} \rightarrow \mathcal{X}}^{\infty}+o(1)=2 \log N+\mathcal{L}_{\infty}+o(1)$
> > >
> > > This makes explicit that $\mathcal{L}_{\mathrm{MCL}}^{(N)}-2 \log N$ is the **canonical object of theoretical analysis**, whereas the $2 \log N$ term arises purely from the size of the softmax normalizers. The absence of this standalone derivation is a matter of presentation clarity **rather than theoretical correctness**.
> > >
> > > ## Further W2
> > >
> > > The logic behind **Theorem 3** is:
> > >
> > > 1. **Dimension collapse**,  observed in all 78 VLMs,  induces nonzero principal angles between two collapsed subspaces and hence a nonzero $\phi_{\min}$.
> > >
> > > 2. Theorem 3 shows that, under **subspace constraint**, the modality gap converges to $\phi_{\min}$. A nonzero modality gap emerges.
> > >
> > > 3. Without dimension collapse, **Theorem 3 reduces to Theorem 2**, and modality gap diminishes.
> > >
> > > These support results dimension collapse as **a theoretically grounded and empirically supported geometric mechanism** behind modality gap, **rather than a controlled intervention demonstrating causality** in the strongest sense. We will revise the wording accordingly.
> > >
> > > Dimension collapse appears to **arise naturally** in multimodal data. It makes **controlled intervention experiments** such as **whitening away** collapse **nontrivial** in practice. We have tried multiple methods to "whiten it" as you suggested, but **none of them works**. While this is now **unachievable** due to technical difficulties, we will **surely acknowledge it explicitly** as future work in revision.
> > >
> > > Notably, our new experiment reveals **a significant positive relationship** between weighted average of nonzero principal angles between two collapsed subspaces and gap size across 78 VLMs, further supporting Theorem 3. However, establishing a precise quantitative relationship between gap size and dimension collapse (e.g., rank deficiency) requires a **substantial theoretical extension** beyond the current scope, as it also depends on **relative rotation and positioning** of two collapsed subspaces. This is also the main obstacle to extending Theorems 3 and 4 to more general settings (see Q8).
> > >
> > > ## Further W3
> > >
> > > **Theorems 1-4** study asymptotic behavior of $\mathcal{L}_{\mathrm{MCL}}$ as $N \to \infty$ (**infinitely many training samples**). In this regime, we analyze distribution level properties of (X,Y), including, relationship between their distribution centers. **Corollary 1-2** study empirical implications in **finite downstream evaluation set**, **without** requiring the existence of an actual center pair and with **no Injectivity issue**.
> > >
> > > A perturbation analysis would be **useful addition, rather than correctness decider**, for the validity of our theory. We **have successfully derived** that **the modality gap converges to a quantity bounded by $f(\epsilon)$** and will add this to the revision.
> > >
> > > ## Further W6:
> > >
> > > We **respectfully disagree** with the concern that $\mathrm{SAlign}(X,Y)$ is close to measuring the same thing as downstream performance. These two are **fundamentally different quantities**. $\mathrm{SAlign}(X,Y)$ is a geometry-based measure defined purely in terms of alignment within the shared subspace, independent of any task-specific evaluation protocol. In contrast, downstream performance (e.g., retrieval accuracy) depends on both shared-space alignment and cross-modal mismatches outside the shared subspace (e.g., affected by **modality gap**) .
> > >
> > > **The modified SSP experiment** in our reply to **W7 and Q4** shows that:  CLIP and CLIP+SPP are **matched on degree of shared space alignment but differing their raw retrieval performance**. This experiment partials out the effect of shared-space alignment and downstream performance.  Therefore, the empirical correlation between $\mathrm{SAlign}(X,Y)$ and downstream performance does not arise from a definitional circularity.

---

### Official Review · Reviewer_4zXr · 2026-03-12

**Soundness:** 2
**Presentation:** 2
**Significance:** 3
**Originality:** 3
**Overall Recommendation:** 4
**Confidence:** 3

**Summary:**

The paper studies the geometry of multimodal contrastive learning in vision-language models, focusing on the modality gap and its connection to downstream performance. It proposes a theoretical framework suggesting that, under idealized conditions, the modality gap should vanish, and argues that persistent gap instead arises from dimension collapse into distinct but overlapping subspaces rather than from the cone effect alone. The paper further claims that alignment in a shared subspace is more important for downstream performance than the raw modality gap. Empirically, it analyzes many pretrained VLMs and proposes both a post-hoc method and a training objective to improve shared-space alignment.

**Compliance With Llm Reviewing Policy:**

Affirmed.

**Final Justification:**

Overall, the paper makes a meaningful contribution by providing a clear perspective on the modality gap and supporting it with both theoretical and empirical evidence, despite some limitations in causal validation and large-scale experimentation.

**Key Questions For Authors:**

1. How robust are the main theoretical conclusions to the paper’s strongest assumptions, especially the simplified subspace model and distributional assumptions?
2. Can you better control for confounding factors in the 78-model analysis, such as model scale, architecture, and training data, to support the claim that shared-space alignment is the main driver of performance?
3. How stable are the gains from the proposed training objective across multiple seeds, stronger baselines, and larger-scale training setups? Evidence of robustness here would make the practical contribution more convincing.

**Limitations:**

Yes.

**Strengths And Weaknesses:**

Strengths:
1. The paper addresses an important question in multimodal representation learning: what causes the modality gap and whether reducing it is actually beneficial. This is a timely and relevant problem for vision-language models.
2. The paper’s main perspective is interesting and reasonably original: it links modality gap to dimension collapse and argues that alignment in a shared subspace matters more than raw embedding overlap. This gives a useful conceptual lens for understanding contrastive VLMs.
3. The work is broad in scope, combining theory, large-scale empirical analysis across many pretrained models, and practical methods (SSP/SSA). This makes the paper more valuable than a purely theoretical or purely empirical study.

Weaknesses:
1. The theoretical results rely on strong simplifying assumptions, so the claim that dimension collapse is the fundamental cause of modality gap feels stronger than what is fully established. The theory is suggestive, but its applicability to real VLMs is not completely clear.
2. The empirical evidence is mainly correlational. The cross-model analysis supports the paper’s claims, but does not fully rule out confounds such as model scale, architecture, or training data, so the causal interpretation remains limited.
3. The practical validation is still somewhat limited. The proposed training objective is tested in a relatively modest setup, and it is unclear whether the gains would remain consistent at larger scales or under stronger training baselines.

---

> ### Author Rebuttal · Authors · 2026-03-31
>
> # Response to Reviewer 4zXr
>
> ## Weakness 1 and Question 1
>
> Our theory is developed under a simplified setting to enable closed-form geometric analysis. While the setting is idealized, its main conclusions are robust in practice and remain applicable to real VLMs.
>
> In particular, the claim that **dimension collapse is the fundamental source of the modality gap** extends naturally to practice: whenever image and text representations collapse into different subspaces, the two modalities become structurally separated, and a modality gap emerges.
>
> In the submitted version, we provided evidence of the **cone constraint** for all 78 VLMs, but showed evidence of **dimension collapse** only for ViT-B/32. In the revised version, we will add results for all models showing that $X$ and $Y$ collapse into disjoint subspaces. This will further support that our structural assumptions are relevant to real VLMs.
>
> More importantly, the main conclusion of our theory is that the **contrastive objective promotes shared-space alignment**, and that **shared-space alignment is a dominant factor in downstream performance**. Although this conclusion is derived under a simplified setting, we also validate it empirically on all 78 real VLMs:
>
> | Metric | $\mathrm{SAlign}(X, Y)$ vs. $\mathcal{L}_{\mathrm{MCL}}$ | $\mathrm{SAlign}(X, Y)$ vs. Performance |
> |---|---:|---:|
> | MSCOCO (I2T) | $-0.456$ ($\checkmark$) | $0.682$ ($\checkmark$) |
> | MSCOCO (T2I) | $(-)$ | $0.900$ ($\checkmark$) |
> | ImageNet | $-0.375$ ($\checkmark$) | $0.883$ ($\checkmark$) |
>
> **Table.** Results on 78 contrastively pretrained VLMs. We report Kendall’s $\tau$ rank correlation between $\mathrm{SAlign}$ and $\mathcal{L}_{\mathrm{MCL}}$ or downstream performance. $\checkmark$ denotes statistical significance under a permutation test ($p<0.05$).
>
> These results show a statistically significant **negative correlation** between $\mathrm{SAlign}(X, Y)$ and $\mathcal{L}_{\mathrm{MCL}}$, indicating that the **contrastive objective promotes shared-space alignment**. They also show a statistically significant **positive correlation** between $\mathrm{SAlign}(X, Y)$ and downstream performance, confirming that **shared-space alignment plays a central role in downstream performance**.
>
> Therefore, our theoretical conclusions are also supported in practice.
>
> ## Weakness 2 and Question 2
>
> The results above directly support our claim that **shared-space alignment is the main driver of downstream performance**. The rank correlations are computed across all 78 models, and statistical significance is established using a permutation test. This support the robustness of our theoretical conclusions.
>
> Following [1], we also group the 78 VLMs by **model size** and **training-data size**. The same trend remains statistically significant across groups. Due to the rebuttal length limit, we cannot include all of these additional results here, but we will add them in the revised version.
>
> **[1]** *Two Effects, One Trigger: On the Modality Gap, Object Bias, and Information Imbalance in Contrastive Vision-Language Models*, Schrodi et al., 2025.
>
> ## Weakness 3 and Question 3
>
> All models, including ours and the baselines, are trained **from scratch** on **Conceptual Captions 12M (CC12M)** [2], which is already a sufficiently large-scale pretraining dataset. We also provide additional evaluations on multiple benchmarks (see our response to Reviewer jCys’s Weakness 1) and direct evidence for the SSA mechanism (see our response to Reviewer jCys’s Weakness 2), which together support the robustness of our method.
>
> Training on even larger-scale datasets would require substantially more computation and is beyond our current computational budget.
>
> **[2]** *Pushing Web-Scale Image-Text Pre-Training to Recognize Long-Tail Visual Concepts*, Changpinyo et al., 2021.

---

> > ### Author Rebuttal · Reviewer_4zXr · 2026-04-01
> >
> > Thanks for the rebuttal, which has addressed my concerns. I would like to keep my original rating.

---

> > > ### Author Response · Authors · 2026-04-07
> > >
> > > We sincerely thank the reviewer for the constructive feedback and for engaging with our experiments. Your comments are very helpful in improving the clarity of the paper, especially around the applicability of our theory to real VLMs. We are glad that our rebuttal has resolved your concerns.

---

### Official Review · Reviewer_jCys · 2026-03-18

**Soundness:** 3
**Presentation:** 3
**Significance:** 3
**Originality:** 2
**Overall Recommendation:** 4
**Confidence:** 3

**Summary:**

This paper presents a theoretical framework for analyzing Convergent Optimal Representations (COR) in multimodal contrastive learning (MCL). The authors identify dimension collaps, i.e., where image and text representations collapse into different subspace, as the root cause of the modality gap phenomenon. Their key theoretical finding reveals that while the modality gap prevents direct alignment between modalities, projections onto a shared subspace can still achieve effective alignment, which dominates downstream task performance. Motivated by this analysis, the authors propose Shared Space Alignment (SSA) to enhance MCL pretraining by improving alignment within the shared space. The paper includes extensive experiments to validate both the theoretical predictions and the practical effectiveness of SSA.

**Compliance With Llm Reviewing Policy:**

Affirmed.

**Key Questions For Authors:**

**Regarding the Relationship with AlignCLIP (ICLR 2025)：**

The recent work AlignCLIP (ICLR 2025) investigates the identical problem of modality gap in multimodal contrastive learning. Several aspects require clarification to establish the novelty and necessity of your proposed framework:

- AlignCLIP identifies the modality gap as a geometric phenomenon (approximately 70° angular separation between image and text embeddings) caused by contrastive loss optimization dynamics and initialization. Your work attributes the gap to "dimension collapse" where representations collapse into different subspaces. Are these two explanations fundamentally distinct？and Does your COR (Convergent Optimal Representations) framework provide analytical results that are inaccessible through the gradient-flow perspective of AlignCLIP?

- Both works propose "shared space" mechanisms: your Shared Space Alignment (SSA) and AlignCLIP's SharedCLIP with parameter sharing. What capability does SSA provide that SharedCLIP+IMSep cannot achieve? Specifically, does SSA's theoretical grounding in COR analysis translate to empirical advantages on standard benchmarks (MSCOCO, Flickr30K)?

**Strengths And Weaknesses:**

**Strengths **：

- The paper introduces the first convergent optimal representation (COR) framework for analyzing MCL, offering a distinct mathematical lens (dimension collapse in subspaces) compared to existing gradient-flow analyses.

- The finding that shared space alignment dominates downstream performance—rather than direct cross-modal alignment—provides a principled motivation for the proposed SSA method.

**Weaknesse:**

- The paper lacks head-to-head comparisons with AlignCLIP (ICLR 2025) on standard benchmarks (missing the results on Flickr30K, Flowers-102, Stanford Cars). Without this alignment, claims about practical effectiveness remain unsubstantiated, and the theoretical novelty risks being inconsequential if it yields no measurable improvement over existing solutions.

-  The absence of qualitative analysis in ablation studies limits understanding of how SSA modifies representation geometry. Visualizing the shared subspace projections or dimension collapse patterns would strengthen the link between theory and observed behavior.

- The statement that contrastive learning "closes, rather than preserves, the modality gap" appears contradictory to established empirical findings (e.g., persistent ~70° gap in CLIP). The authors must clarify whether this describes the optimization objective versus the convergent outcome, as this distinction critically affects the practical relevance of the theoretical framework.

---

> ### Author Rebuttal · Authors · 2026-03-31
>
> # Response to Reviewer jCys
>
> ## Weakness 1:
>
> To further strengthen the comparison with related methods, we provide additional zero-shot cross-modal retrieval results on **Flickr30K** below. Due to the rebuttal length limit, we are unable to include all additional benchmarks here, but we will add results on **Flowers-102** and **Stanford Cars** in the revised version.
>
> ### Zero-shot cross-modal retrieval results on Flickr30K
>
> | Model | I $\rightarrow$ T R@1 | R@5 | R@10 | T $\rightarrow$ I R@1 | R@5 | R@10 |
> |---|---:|---:|---:|---:|---:|---:|
> | CLIP | 53.2 | 80.5 | 88.6 | 39.9 | 69.0 | 78.5 |
> | SharedCLIP | 58.3 | 83.6 | **89.8** | 42.6 | 70.0 | 79.1 |
> | IMSep | 56.8 | 82.8 | 89.2 | 42.0 | 69.9 | **79.4** |
> | AlignCLIP | 57.2 | 82.3 | 89.5 | 41.8 | 70.2 | 79.1 |
> | **CLIP + SSA** | **58.7** | **83.8** | 89.7 | **42.9** | **70.3** | **79.4** |
>
> These results show confirm the effectiveness of SSA
>
>
> ## Weakness 2:
>
> To further clarify that SSA improves pretraining by enhancing **shared-space alignment**, we directly measure **SAlign** (defined in Eq. (10)) for all models on **MSCOCO**. We agree that **quantitative evidence is more persuasive than visualization alone**, and provide the results below.
>
> ### Downstream retrieval performance on MSCOCO
>
> | Model | SAlign | I $\rightarrow$ T R@1 | R@5 | R@10 | T $\rightarrow$ I R@1 | R@5 | R@10 |
> |---|---:|---:|---:|---:|---:|---:|---:|
> | CLIP | 17.6 | 31.4 | 57.0 | 68.6 | 20.5 | 44.1 | 55.9 |
> | SharedCLIP | 17.8 | 33.6 | 59.6 | 70.8 | 21.8 | **45.4** | **57.3** |
> | IMSep | 17.4 | 33.7 | **60.8** | **71.5** | 21.5 | 45.1 | 56.9 |
> | AlignCLIP | 17.8 | 34.0 | 59.7 | **71.5** | **22.3** | 45.0 | 56.9 |
> | **CLIP + SSA** | **21.8** | **34.5** | 60.7 | **71.5** | **22.8** | **45.4** | 57.1 |
>
> The results show that **SSA substantially improves SAlign**, and this improvement is accompanied by stronger downstream retrieval performance.
>
> ## Weakness 3:
>
> The statement that contrastive learning **“closes, rather than preserves, the modality gap”** is based on **Theorem 1**, which studies **unconstrained convergence**. This corresponds to an idealized setting that helps characterize the theoretical behavior of the contrastive objective itself.
>
> In practice, however, optimization is subject to both the **cone constraint** and the **subspace constraint** (see Section 3). Under these practical constraints, **Theorem 3** and **Theorem 4** characterize the geometry of the COR obtained during real training. Therefore, our theoretical analysis distinguishes between:
>
> - the **idealized unconstrained limit**, where the modality gap vanishes, and
> - the **practical constrained setting**, where a modality gap can emerge because the two modalities collapse into different subspaces.
>
> ## Question 1:
>
> Our work differs fundamentally from **AlignCLIP** in both **goal** and **methodology**.
>
> ### 1. Different analytical lens
>
> AlignCLIP explains the modality gap primarily through **initialization** and **training dynamics**, that is, the beginning and the process of training. In contrast, our work studies the **convergent optimal representation (COR)** of multimodal contrastive learning, focusing on the **end of training**. Specifically, we ask:
>
> > What geometric structure must the representations satisfy when optimization approaches its limiting optimum?
>
> From this perspective, we mathematically show that the modality gap should ultimately converge to zero even under the **cone constraint** (Theorem 2). We further show that a modality gap emerges when image and text representations collapse into **different subspaces**. This leads us to conclude that **subspace geometry is the structural source of the modality gap**.
>
> ### 2. Different implications
>
> Our framework allows us to distinguish two effects that are not explicitly disentangled in AlignCLIP:
>
> - **the size of the modality gap**, and
> - **the degree of shared-space alignment**.
>
> Our theory predicts, and our experiments support, that **downstream performance is governed primarily by shared-space alignment rather than by the modality-gap size itself**. Our **SSP experiment** shows that **reducing the modality gap alone, while keeping shared-space alignment unchanged, does not improve downstream performance**. This suggests that the modality gap plays only a limited role in determining downstream performance.
>
> ### 3. Different notion of “shared space” and different mechanism
>
> The notion of “shared space” also differs between the two works.
>
> - In **AlignCLIP**, the shared space refers to the **parameter space** of the image and text encoders.
> - In our work, the shared space refers to the **embedding space** of image and text representations.
>
> Accordingly, the mechanisms are also different:
>
> - Our method improves the alignment of the **projected image and text representations in the shared embedding space**.
> - AlignCLIP shares parameters between the image and text encoders and improves the **intra-modal separation** of image representations.

---

### Decision · Program_Chairs · 2026-04-30

**Decision:**

Accept (regular)

**Comment:**

This paper was reviewed by four experts in the field and received 4 Weak Accept ratings. Based on the reviewers' feedback, the decision is to recommend the paper for acceptance to ICML 2026.  Reviewers viewed the paper positively overall, highlighting a “clear and original theoretical contribution,” a “useful conceptual lens for understanding contrastive VLMs,” and a combination of theory, “large-scale empirical analysis,” and practical methods that makes the work “more valuable than a purely theoretical or purely empirical study”. In particular, this work identifies dimension collapse as the key structural cause and showing that shared-space alignment, not raw gap size, is what most strongly predicts downstream performance.

At the same time, the final version should address several remaining concerns raised in the reviews and discussion: clearer comparison to AlignCLIP and stronger evaluation on standard benchmarks, since the paper initially “lacks head-to-head comparisons with AlignCLIP” and some “quantitative metrics appear suboptimal” (jCys); and stronger empirical completeness, including “broader task diversity,” “sufficiently complete ablations on key SSA design choices,” “quantitative efficiency analysis,” and more discussion of “training dynamics / optimization behavior” (Gcru). Overall, the paper is a solid contribution, and addressing these points in the final version would further strengthen it. We congratulate the authors on the acceptance of their paper!